# Fast local warming is the main driver of recent deoxygenation in the northern Arabian Sea

Zouhair Lachkar[1], Michael Mehari[1], Muchamad Al Azhar[1,4], Marina Lévy[2], and Shafer Smith[1,3]

[1]Center for Prototype Climate Modeling, New York University Abu Dhabi, Abu Dhabi, UAE
[2]Sorbonne Université (CNRS/IRD/MNHN), LOCEAN-IPSL, Paris, France
[3]Courant Institute of Mathematical Sciences, New York University, New York, USA
[4]Plymouth Marine Laboratory, Plymouth, UK

**Correspondence:** Zouhair Lachkar (zouhair.lachkar@nyu.edu)

**Abstract.**

The Arabian Sea (AS) hosts one of the most intense oxygen minimum zones (OMZs) in the world. Observations suggest a decline of $O_2$ in the northern AS over the recent decades accompanied by an intensification of the suboxic conditions there. Over the same period, the local sea-surface temperature has risen significantly, particularly over the Arabian Gulf (also known as Persian Gulf, hereafter the Gulf), while summer monsoon winds may have intensified. Here, we simulate the evolution of dissolved oxygen in the AS from 1982 through 2010 and explore its controlling factors, with a focus on changing atmospheric conditions. To this end, we use a set of eddy-resolving hindcast simulations forced with winds and heat and freshwater fluxes from an atmospheric reanalysis. We find a significant deoxygenation in the northern AS with $O_2$ inventories north of 20°N dropping by over 6% decade$^{-1}$ between 100 and 1000 m. These changes cause an expansion of the OMZ volume north of 20°N at a rate of 0.6% decade$^{-1}$ as well as an increase in the volume of suboxia and the rate of denitrification by 14% decade$^{-1}$ and 15% decade$^{-1}$, respectively. We also show that strong interannual and decadal variability modulate dissolved oxygen in the northern AS with most of the $O_2$ decline taking place in the 1980s and 1990s. Using a set of sensitivity simulations we demonstrate that deoxygenation in the northern AS is essentially caused by reduced ventilation induced by the recent fast warming of the sea surface, including in the Gulf, with a contribution from concomitant summer monsoon wind intensification. This is because, on the one hand, surface warming enhances vertical stratification and increases Gulf water buoyancy, thus inhibiting vertical mixing and ventilation of the thermocline. On the other hand, summer monsoon wind intensification causes a rise of the thermocline depth in the northern AS that lowers $O_2$ levels in the upper ocean. Our findings confirm that the AS OMZ is strongly sensitive to upper-ocean warming and concurrent changes in the Indian monsoon winds. Finally, our results also demonstrate that changes in the local climatic forcing play a key role in regional dissolved oxygen changes and hence need to be properly represented in global models to reduce uncertainties in future projections of deoxygenation.

## 1 Introduction

Rising ocean temperatures decrease $O_2$ solubility in seawater, increase respiration-driven oxygen consumption and enhance vertical stratification, thus reducing interior ocean ventilation (Oschlies et al., 2018). These changes collectively cause the

ocean to lose oxygen as it warms up, a process termed ocean deoxygenation. Observational and modeling evidence suggest that the majority of the observed oxygen decline is caused by changes in ocean ventilation and biogeochemistry (Bindoff et al., 2019). Over the last five to six decades, the global ocean oxygen content has dropped by 2% (Ito et al., 2017; Schmidtko et al., 2017), a tendency predicted to accelerate in the future in response to ocean warming. In the upper 1000 m, a growing consensus

points towards a loss of $O_2$ of 0.5-3.3% between 1970-2010 (Bindoff et al., 2019). However, the analysis of local time series suggests much stronger trends at particular sites (Whitney et al., 2007; Bograd et al., 2015). Ocean deoxygenation can cause the expansion of naturally-occurring low-$O_2$ water bodies known as Oxygen Minimum Zones (OMZs) (Stramma et al., 2008; Breitburg et al., 2018; Bindoff et al., 2019). This can increase the frequency and intensity of hypoxic events, stressing sensitive organisms and causing loss of marine biodiversity and shifts in the food web structure (Rabalais et al., 2002; Vaquer-Sunyer

and Duarte, 2008; Laffoley and Baxter, 2019).

The Arabian Sea (AS) hosts one of the world's largest and most extreme OMZs, with suboxia ($O_2 < 4$ mmol m$^{-3}$) prevailing across most of the intermediate ocean (from 150 down to 1,250 m) in its northern and northeastern parts, turning it into a global hotspot of denitrification (Bange et al., 2005; Codispoti et al., 2001). Because of the challenges associated with data sparsity, the previously documented $O_2$ changes in the AS depict a complex picture with no consistent trends across the entire

region (Laffoley and Baxter, 2019; Bindoff et al., 2019). Yet, there is mounting evidence for a decline of $O_2$ concentrations in the northern AS over the last few decades. For example, the global analysis of historical oxygen observations by Ito et al. (2017) reveals a moderate drop of $O_2$ levels in the subsurface of the AS between 1960 and 2010. The analysis of available $O_2$ observations by Schmidtko et al. (2017) similarly indicates a decline of oxygen in the northern and western AS as well as along the west coast of India between 1960 and 2010. In an analysis of over 2000 $O_2$ profiles collected during 53 oceanographic

expeditions that took place between 1960 and 2008 off the coast of Oman, Piontkovski and Al-Oufi (2015) documented a decline of $O_2$ in the upper 300 m in the northern and northwestern AS that they attributed to increased thermal stratification and a shoaling of the oxycline between the 1960s and 2000s. Banse et al. (2014) analyzed discrete historical $O_2$ measurements collected in the central and southern AS in the 150-500 m layer between 1959 and 2004. They found no clear systematic trend across the entire AS, although $O_2$ was found to decline in most of the central AS and slightly increase in the southern AS

(between 8-12°N). Because of the sparsity of observations, these trends were generally based on a small number of samples and not always statistically significant. Authors in this study also analyzed trends in subsurface nitrite ($NO_2^-$) concentrations, typically used as a proxy of the presence of suboxia and denitrification at depth. They found both positive and negative trends in different locations with a larger number of profiles indicating an increase in nitrite concentrations over time, suggesting an overall intensification of the OMZ over the study period. do Rosário Gomes et al. (2014) reported a radical shift in the

winter bloom dominant phytoplankton species from diatoms to large dinoflagellate, Noctiluca scintillans, which they linked to a decline of $O_2$ concentrations in the region. More recently, using sea glider data and historical observations, Queste et al. (2018) showed an intensification of the suboxic conditions at depth in the Gulf of Oman over the last three decades. Although there has been little work done on documenting potential deoxygenation in the Arabian marginal seas (i.e., the Red Sea and the Gulf), preliminary observations suggest ongoing deoxygenation in the Gulf with recent emergence of summertime hypoxia

documented there (Al-Ansari et al., 2015; Al-Yamani and Naqvi, 2019).

In addition to changes in $O_2$, the AS has undergone major environmental changes over the recent decades that may intensify in the future. In particular, the AS has experienced a strong warming throughout most of the twentieth century that has accelerated since the early 1990s (Kumar et al., 2009; Gopika et al., 2020). The warming has been particularly fast in the two Arabian marginal seas (i.e., the Red Sea and the Gulf) over the last three decades with warming rates reaching up to $0.17\pm0.07°C$ decade$^{-1}$ and $0.6\pm0.3°C$ decade$^{-1}$ in the two semi-enclosed seas, respectively (Chaidez et al., 2017; Strong et al., 2011; Al-Rashidi et al., 2009). The AS warming has been linked to important physical and biogeochemical changes. For instance, using satellite observations and a set of historical simulations of the northern Indian Ocean, Roxy et al. (2016) found a drop of summer productivity by up to 20% between 1950 and 2005 in the western AS that they linked to surface warming and increased stratification. Goes et al. (2020) reported an increase in winter stratification associated with a weakening of winter convective mixing in the northern AS. These authors also presented observational evidence of a strong loss of inorganic nitrogen in the AS over the recent decades that they linked to enhanced denitrification driven by OMZ intensification. These changes have been suggested to favor winter blooms of the mixotroph Noctiluca scintillans at the expense of diatoms (Goes et al., 2020). The Gulf warming has also been associated with important ecological and biogeochemical changes such as recent frequent mass coral bleaching events (Burt et al., 2019) as well as a potential reduction of the ventilation of the AS OMZ (Lachkar et al., 2019). Finally, important changes in the summer monsoon winds have also been reported. For instance, Sandeep and Ajayamohan (2015) described a poleward shift in the monsoon low level jet stream over the recent decades that is expected to intensify in the future. This would lead to an intensification (resp. weakening) of summer monsoon winds off the coast of Oman (resp. Somalia). Praveen et al. (2016) suggested the Oman coast is to experience a future increase in summer upwelling accompanied with an enhancement of summer productivity whereas deCastro et al. (2016) predicted a future strengthening of the Somali coastal upwelling based on an ensemble of global and regional model simulations for the 21st century. Yet, other studies reported a decline in the intensity of summer monsoon winds. For instance, Swapna et al. (2017) suggested a weakening of the summer monsoon circulation over the recent decades. The analysis of a selected set of CMIP5 models by Sooraj et al. (2015) also suggests a future weakening of the Asian Summer monsoon circulation south of $10°N$. However, this study also projects an intensification of the summer monsoon circulation north of $10°N$ associated with a northward shift of the monsoon circulation in agreement with previous CMIP5 model analyses.

While these environmental perturbations may have contributed to the observed $O_2$ decline in the northern AS, their potential interactions and the mechanisms through which they act to modulate $O_2$ remain poorly understood. In particular, the respective roles of recent surface warming on the one hand and summer monsoon wind intensification on the other hand are yet to be quantified. Moreover, the potential contribution of the recent fast warming of the Gulf to the declining $O_2$ levels in the northern AS has not yet been investigated. Here we reconstruct the trends in $O_2$ over the period between 1982 and 2010 using a high-resolution hindcast simulation of the Indian Ocean and examine their physical and biogeochemical drivers using a set of sensitivity experiments. We show that recent deoxygenation in the northern AS has been caused essentially by surface warming, with a significant contribution from the Gulf warming, bringing about a reduction in the ventilation of the subsurface and intermediate layers. We also show that summer monsoon intensification enhanced oxygenation of the upper ocean south

of 20°N but has contributed to deoxygenation in the northern AS. These changes are likely to have important ecological and biogeochemical consequences.

## 2 Methods

### 2.1 Models

The circulation model is the Regional Ocean Modeling System (ROMS)-AGRIF version (http://www.croco-ocean.org) configured for the Indian Ocean. It uses the free-surface, hydrostatic, primitive equations in a rotating environment and has a terrain-following vertical coordinate system (Shchepetkin and McWilliams, 2005). To limit diapycnal mixing errors, the diffusive component of the rotated, split, upstream-biased, 3$^{rd}$ order (RSUP3) tracer advection scheme is rotated along geopotential surfaces (Marchesiello et al., 2009). The non-local K-Profile Parameterization (KPP) scheme is used to parametrize subgrid
vertical mixing (Large et al., 1994). The model domain covers the full Indian Ocean from 31°S to 31°N and 30°E to 120°E with a 1/10° horizontal resolution and 32 sigma-coordinate vertical layers with enhanced resolution near the surface. The biogeochemical model is a nitrogen-based nutrient-phytoplankton-zooplankton-detritus (NPZD) model (Gruber et al., 2006). It is based on a system of ordinary differential equations representing the time-evolution of seven state variables: two nutrients (nitrate and ammonium), one phytoplankton class, one zooplankton class, two classes of detritus (small and large sizes) and a
dynamic chlorophyll-to-carbon ratio. The model has a module describing the cycling of oxygen as well as a parameterization of water-column and benthic denitrification (Lachkar et al., 2016).

### 2.2 Experimental design

  The hindcast simulation is forced with ECMWF ERA-Interim 6-hourly heat fluxes, air temperature, pressure, humidity, precipitation and winds over the period from January 1982 to December 2010. Sea surface temperature (SST) is restored to
AVHRR-Pathfinder and Aqua-Modis observations and surface salinity is restored to the Simple Ocean Data Assimilation (SODA) reanalysis data using kinematic heat and freshwater flux corrections proposed by Barnier et al. (1995). Initial and lateral boundary conditions for temperature, salinity, currents and sea surface height are computed from the SODA reanalysis (Carton and Giese, 2008). The initial and lateral boundary conditions for nitrate and oxygen are extracted from the World Ocean Atlas (WOA) 2013 (Garcia et al., 2013a, b). Finally, we used a monthly climatological runoff and annual river nutrient
discharge from major rivers in the northern Indian Ocean (including the Indus and Narmada rivers flowing into the Arabian Sea) derived from available observations and a global hydrological model (Ramesh et al., 1995; Dai and Trenberth, 2002; Krishna et al., 2016). We restrict our analysis to the AS region extending from 3.5°S to 30°N in latitude and from 32°E to 76°E in longitude (Fig S1, Supporting Information (SI)).

    The model is spun-up using climatological forcing during 58 years and is then run for four 29-year (1982-2010) repeating
forcing cycles, with the first three cycles used as a part of the spin-up period (i.e., the total duration of the spin-up phase is 145 years) and the forth cycle used for analysis (similar to the forcing protocol used in the Ocean Model Intercomparison

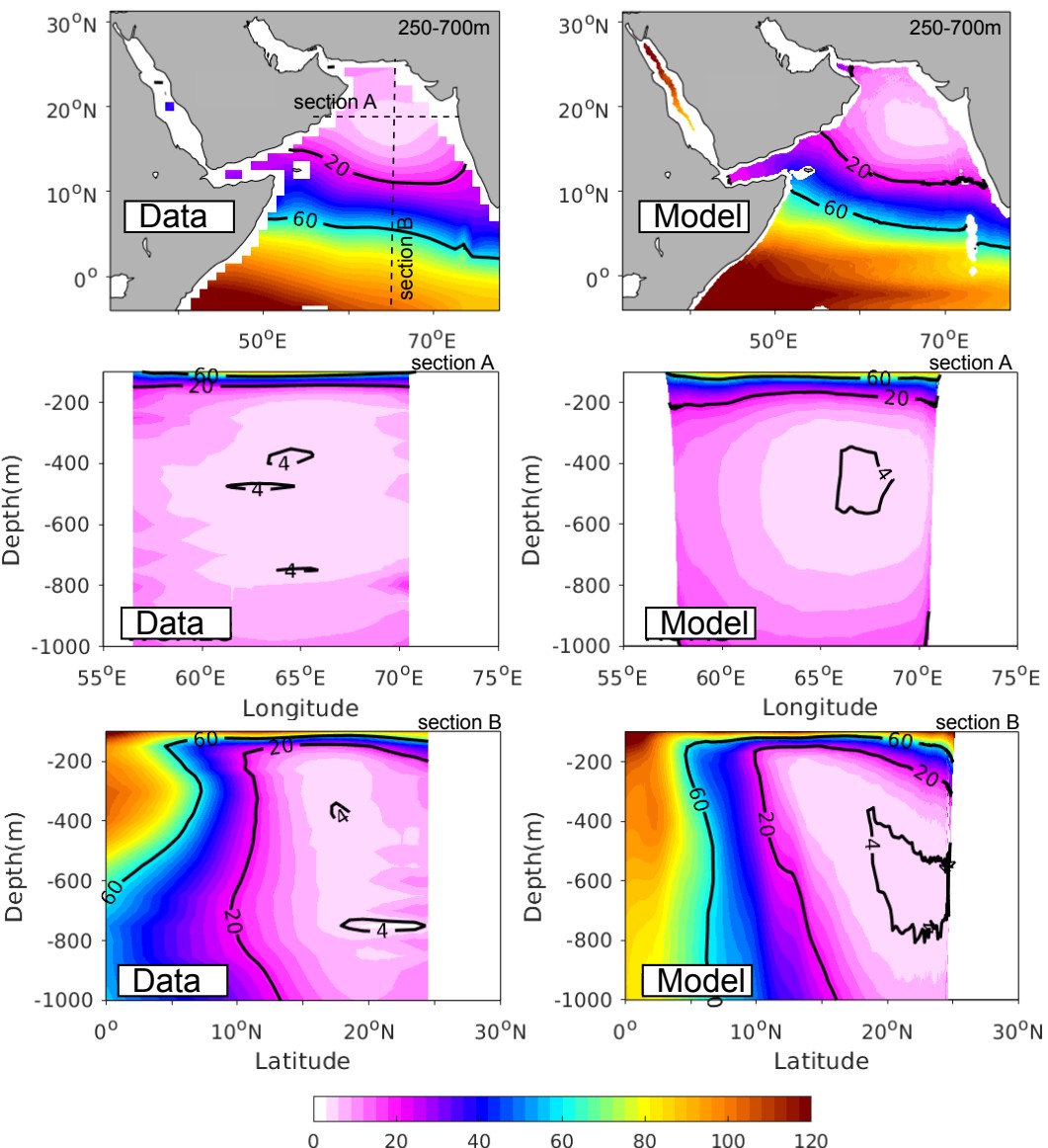

**Figure 1. Evaluation of simulated oxygen** Annual mean $O_2$ (in mmol m$^{-3}$) (top) averaged between 250 m and 700 m and along (middle) zonal section A at 18°N and (bottom) meridional section B at 65°E as simulated in the model (right) and from the WOA-2018 dataset (left).

Project, Griffies et al., 2016). The climatological run is extended for an additional 29 years to quantify the artificial trends purely driven by the model drift and contrast them to trends estimated in our hindcast run. The analysis of model drift indicates a very negligible drift in salinity by the end of the climatological forcing period (Fig S2, SI). For $O_2$, the model drift decreases but remains positive (ocean oxygenation) by the end of the climatological forcing spin-up period, suggesting that the estimated

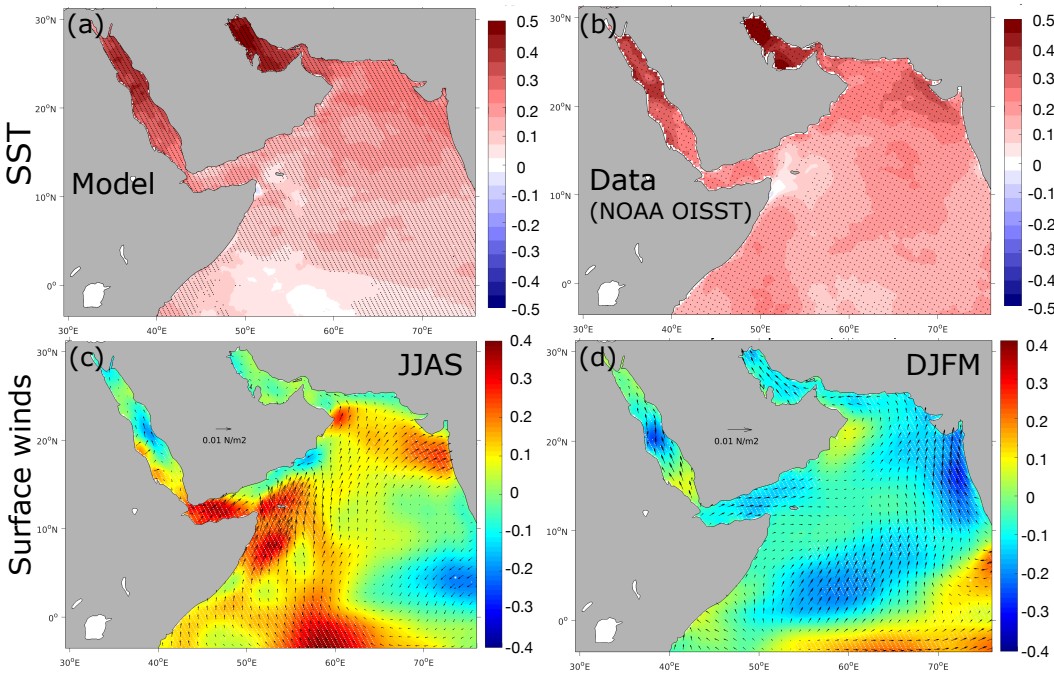

**Figure 2. Warming and surface wind trends.** (a-b) Linear trends in sea surface temperature (in $^{\circ}$C decade$^{-1}$) as simulated by the model (a) and from the NOAA OISST data product. (c-d) trends in ERA-Interim surface winds in summer (c) and winter (d). Color shading indicates trends in wind speed (in m s$^{-1}$ decade$^{-1}$) whereas arrows show trends in wind stress vector. Statistically significant trends at 95% confidence interval following a Mann-Kendall (MK) test are represented by stippling.

deoxygenation rates extracted from our hindcast simulation are rather conservative (Figs S3-S5, SI). A detailed description of the model drift is presented in the SI.

## 2.3 Model evaluation

We use available in-situ and satellite-based observations to evaluate the performance of the model in capturing the spatial patterns as well as the seasonal variability in key physical and biogeochemical properties. We evaluate the mean state by contrasting a model climatology computed over the (1982-2010) study period to available observation-based climatologies. The details of this evaluation are presented in the SI. Despite a few identified biases, the model exhibits reasonable skill in reproducing the mean seasonal climatological state for both physical (i.e., temperature, salinity, currents) and biogeochemical properties (i.e., nitrate, chlorophyll; see Figs S6-S10, SI). Furthermore, the model reproduces fairly well the size, intensity and vertical structure of the AS oxygen minimum zone (Fig 1).

We also evaluate the model performance in reproducing observed long term changes. To this end, we evaluate the model simulated long-term evolution of temperature and salinity at multiple depths as well as the changes in the upper ocean vertical stratification and surface chlorophyll-a concentration. More specifically, we contrast simulated trends in SST to trends based

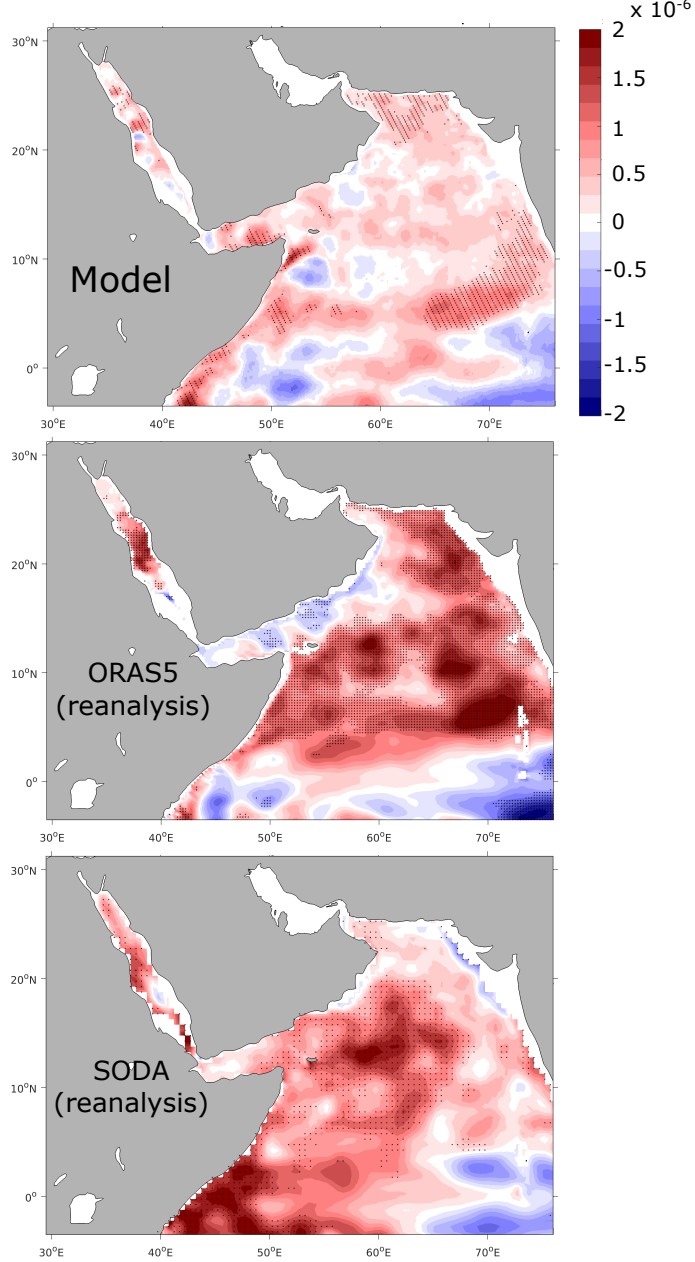

**Figure 3. Trends in upper ocean vertical stratification** Linear trends in static stability $E = \frac{-1}{\sigma}\frac{\partial \sigma}{\partial z}$ (with $\sigma$ the density of the water and z the depth) at $100\,\mathrm{m}$ (in $\mathrm{m}^{-1}\,\mathrm{decade}^{-1}$) between 1982 and 2010 as simulated in ROMS (top) and from ORAS5 (middle) and SODA (bottom) reanalyses. Statistically significant trends at 95% confidence interval following a Mann-Kendall (MK) test are represented by stippling.

on four observational SST products: AVHRR (used to force the model), ERA5 (Merchant et al., 2014), HadlSST (Rayner et al., 2003) and the NOAA OISST blended product (Huang et al., 2021). This comparison reveals that the model simulated warming

agrees relatively well with that from the different SST products, despite differences in the magnitude of warming, with ERA5 (resp. HadlSST) displaying the strongest (resp. weakest) rates of warming (Fig 2 and Fig S11, SI). Furthermore, we also contrast the interannual anomalies in temperature and salinity in the northern AS in the model to those from WOD2018 observations at different depths (Fig S12 and Fig S13, SI). This comparison reveals that in the northern AS the simulated and observed

trends in upper ocean temperature agree well. Indeed, the linear trends in temperature amount to $+0.11°C$ per decade at the surface (resp. $+0.12°C$ per decade at 100m) in the model and $+0.11°C$ per decade at the surface (resp. $+0.14°C$ per decade at 100m) in the WOD2018 dataset (all trends are statistically significant at both depths ($p<0.01$)). In contrast, temperature trends at 200 m are very weak and statistically non-significant in both the model and WOD2018. For salinity, both the model and the WOD2018 observations suggest a slight increase in the upper ocean salinity over the study period (Fig S13, SI). Yet, the highly

sparse observational coverage (most of the observations coming from the last decade of the simulation) precludes extracting meaningful trends from the data to validate the simulated salinity long-term changes (Fig S14, SI). Finally, we also evaluate the evolution of vertical stratification and static stability in the AS in the model. As salinity observations are very sparse over the region during the study period we contrast simulated vertical stratification to that from reanalysis products such as SODA and ORAS5 (Fig 3 and Fig S15, SI). This comparison reveals that overall the simulated increase in vertical stratification in the

AS is comparable to similar trends derived from the ORAS5 and SODA reanalyses, although with local differences in their magnitude and regional patterns. For instance, our model underestimates the magnitude of stratification increase for most of the AS domain relative to ORAS5 and in the central and western AS in comparison to SODA. For biological variables, we only evaluate interannual variability in surface chlorophyll as $O_2$ (and $NO_3^-$) observations are extremely limited in the area during the study period (Fig S14, SI). As satellite chlorophyll data is available only from september 1997, we contrast simulated

chlorophyll to observations over the common period from september 1997 to the end of the year 2010. The 13-year period is too short to extract meaningful long-term trends. Yet, this comparison is still useful as it reveals a decent agreement between the model and observations over the study period with a moderate correlation of 0.48 between the modeled and observed interannual anomalies in the northern AS (Fig S16, SI). In summary, the model reproduces the observed warming trends in the upper ocean of the northern AS and captures relatively well the observed large-scale changes in vertical stratification in the

region. Finally, the model also shows a decent skill at capturing the observed interannual variability in surface chlorophyll.

## 2.4  Trends and oxygen diagnostics

To investigate long-term changes in $O_2$ time series and other environmental properties (e.g., SST, winds, stratification, etc), data was deseasonalized by removing monthly climatologies from the original time series. Furthermore, to identify significant trends in $O_2$ and other environmental factors we used the non-parametric Mann-Kendall (MK) test (Mann, 1945; Kendall, 1948)

that does not assume normality of data distribution and hence is less sensitive to outliers and skewed distributions. Linear trends were computed from the slope of the least squares regression line. To separate oxygen trends driven by changes in solubility (thermal effect) from those caused by changes in ocean ventilation and respiration of organic matter, we decompose the oxygen anomaly $\Delta O_2$ as:

$$\Delta O_2 = \Delta O_2{}^{sat} - \Delta AOU$$

where $O_2{}^{sat}$ is the oxygen saturation concentration (in mmol m$^{-3}$) computed at 1 atm pressure from temperature and salinity following Garcia and Gordon (1992). It corresponds to the maximum $O_2$ concentration in seawater at equilibrium, for a given temperature and salinity; AOU (apparent oxygen utilization) is the difference between oxygen saturation $O_2{}^{sat}$ and the actual $O_2$ concentration (AOU=$O_2{}^{sat}$-$O_2$). It is a measure of $O_2$ utilization through biological activity since the water parcel was last at equilibrium in contact with the atmosphere. Therefore it is sensitive to both biological productivity and circulation (ventilation).

## 2.5 Quantification of the effects of changes in atmospheric forcing

The analysis of long-term trends in atmospheric forcing reveals a widespread warming of the sea surface by between 0.5 and 1°C in the northern AS and by up to 1.5°C in the Gulf and in the northern part of the Red Sea, between 1982 and 2010 (Fig 2). In addition to surface warming, surface winds have undergone important changes with an intensification of upwelling favorable winds off Somalia and Oman, in particular during the summer monsoon season (Fig 2 and Fig S17, SI). To quantify the contributions of surface warming and wind changes to deoxygenation in the AS, we performed four additional sensitivity experiments. In the first simulation, $S_{hclim}$, all atmospheric and lateral boundary conditions were set to vary interannually like in the control run except the heat fluxes that were extracted from a normal year (neutral with respect to major climate variability modes), 1986, and repeated every year (i.e., climatological heat fluxes across the domain). This approach allows us to filter out interannual variability while maintaining the high-frequency variability in the forcing (e.g., Large and Yeager, 2004; Stewart et al., 2020). In the second sensitivity run, $S_{hclim\_AG}$, the heat fluxes were similarly extracted from the year 1986 and repeated annually, but only over the Gulf region (i.e., climatological heat fluxes over the Gulf only). In a third simulation $S_{wclim\_JJAS}$ all atmospheric and lateral boundary conditions were set to vary interannually like in the control run except summer monsoon winds that were extracted from the year 1986 and repeated every year (i.e., climatological summer winds across the domain). Finally, in a forth sensitivity simulation $S_{wclim\_DJFM}$ all atmospheric and lateral boundary conditions were set to vary interannually like in the control run except winter monsoon winds that were extracted from the year 1986 and repeated every year (i.e., climatological winter winds across the domain).

## 3 Results

### 3.1 Deoxygenation trends in the Arabian Sea

The analysis of oxygen trends between 1982 and 2010 shows a decline of $O_2$ concentrations in a large portion of the AS between 100 and 1000 m (Fig. 4). $O_2$ drops locally by more than 10% per decade in the northern AS where the suboxic core of the OMZ is located (Fig. 4). A vertical transect at 65°E indicates that most of the $O_2$ decline is concentrated north of 20°N between 100 and 300 m (Fig. 4). This deoxygenation results in a significant intensification of the OMZ over the three decade study period, with the volume of suboxic ($O_2$ < 4 mmol m$^{-3}$) water increasing by nearly 14% per decade north of 20°N and

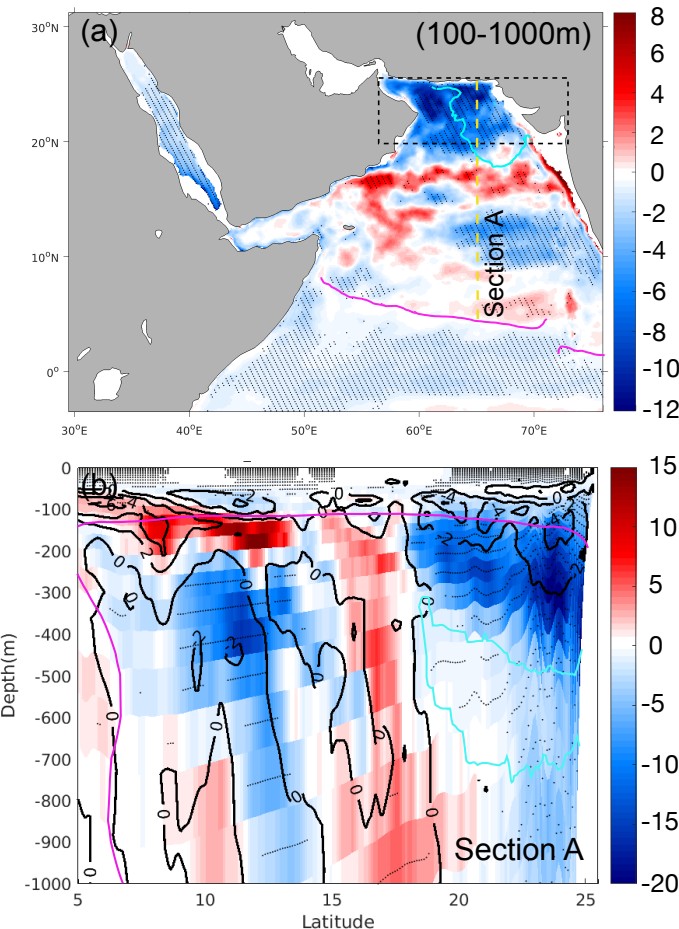

**Figure 4. Deoxygenation rates in the AS between 1982 and 2010.** (a) Trends in $O_2$ inventories (in % per decade) in the 100-1000 m layer. The black dashed line rectangle indicates the location of the northern AS box. The yellow dashed line indicates section A. (b) trends (color shading; in % per decade) and changes between the first five years [1982-1986] and the last five years [2006-2010] (contour lines; in $mmol\,m^{-3}$ per decade) in $O_2$ in the upper 1000 m along section A at 65°E (in % per decade). Statistically significant trends at 95% confidence interval following a Mann-Kendall (MK) test are represented by hatching (a) and stippling (b). The purple and cyan lines indicate the average positions of the hypoxic ($O_2 < 60$ $mmol\,m^{-3}$) and suboxic ($O_2 < 4$ $mmol\,m^{-3}$) boundaries, respectively, at (a) 500 m and (b) along section A. Relative trends (in % per decade) were obtained by dividing the absolute trends by the local mean $O_2$ inventory (a) or concentration (b).

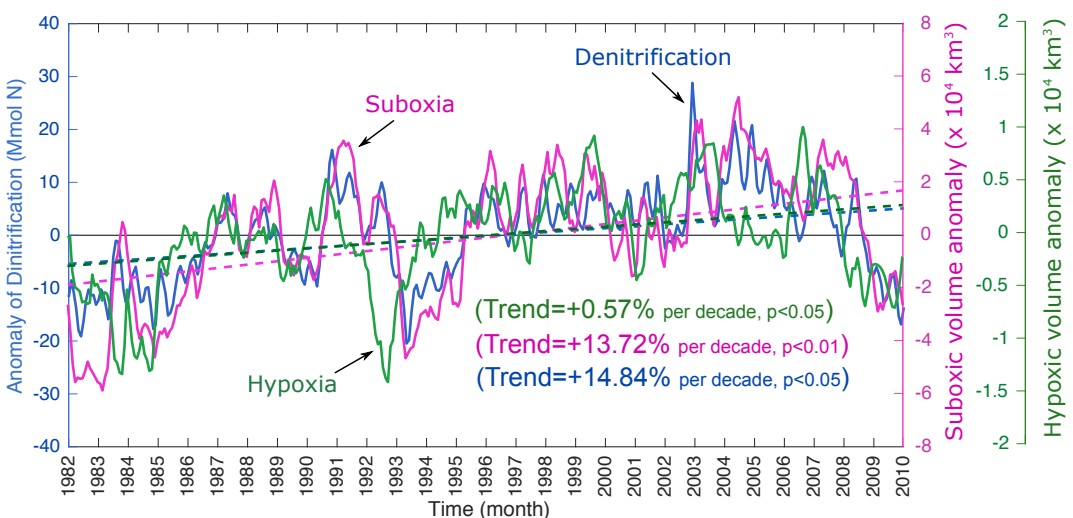

**Figure 5. Changes in Oxygen Minimum Zone and denitrification in northern AS.** Interannual anomalies in the volume of suboxia (purple), hypoxia (green) and water column denitrification (blue) in the upper 1000 m north of 20°N over the study period. The dashed lines indicate the trend lines. The location of the northern AS box is shown in Fig 4a.

by around 10% per decade when considering the entire Arabian Sea domain (Fig. 5 and Fig S18). This causes an amplification of denitrification in the region with an increase in denitrification rates by around 15% per decade over the same period (Fig. 5). The evolution of the suboxic volume and denitrification also reveals strong interannual and decadal fluctuations with a strong rate of intensification of the OMZ during the 1980s and 1990s and a relative stabilization in the last decade of the simulation.

Indeed, the changes in the suboxic volume and denitrification are largest between the early 1980s and early 2000s (Fig. 5). Despite deoxygenation trends dominating in the northern and southern AS, local oxygenation patches are simulated in the central AS (Fig. 4). This results in the net hypoxic volume ($O_2 < 60$ mmol m$^{-3}$) that defines the OMZ volume to change little over the study period (Fig S18). When considering the northern Arabian Sea only, the hypoxic volume shows a statistically significant increase, yet at a modest rate of 1.7% over the three decades of the simulation (Fig. 5). To put these numbers in

a broader context, observations suggest that the volume of the world ocean OMZs has expanded in a range of 3-8% between 1970 and 2010 (Bindoff et al., 2019). The stronger trends simulated for the suboxic volume relative to the hypoxic volume is also consistent with previous studies. For instance, it has been suggested that the volume of anoxic waters in the global ocean has quadrupled since 1960 (Schmidtko et al., 2017). Moreover, using numerical simulations, Deutsch et al. (2011) have shown that the amplitude of variations in the volume of low oxygen conditions in the global ocean increases between 1959 and 2005

from 10% for strong hypoxia ($O_2 < 40$ mmol m$^{-3}$) to nearly 100% for suboxia. Next, we focus on the northern AS (north of 20°N) where simulated deoxygenation and its impacts on the OMZ are the most prominent.

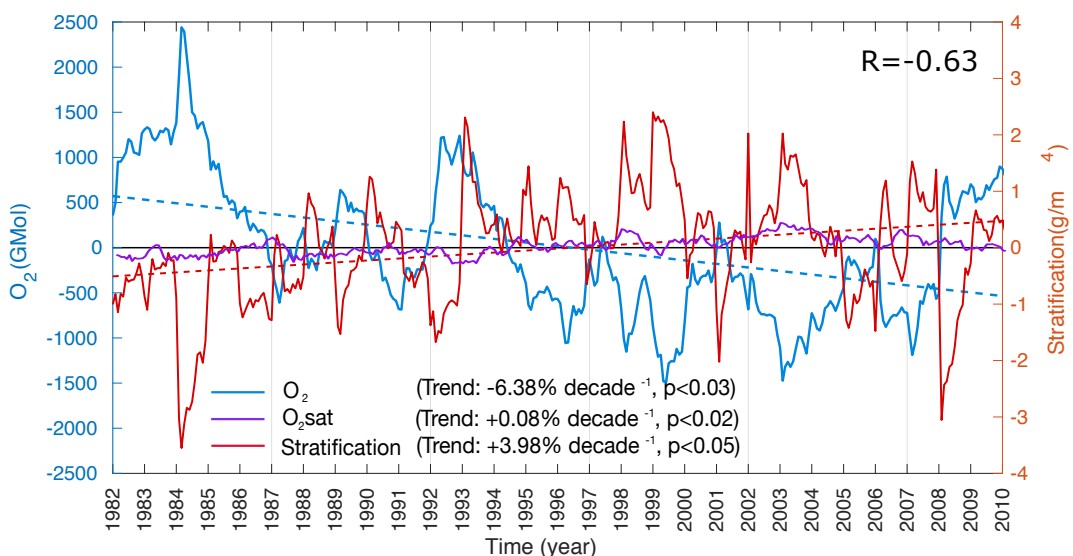

**Figure 6. Drivers of northern AS deoxygenation between 1982 and 2010.** $O_2$ content (blue) and $O_2$ saturation (purple) anomalies between 100 and 1000 m and anomalies of vertical stratification at 100 m (red) in the northern AS box as a function of time. The blue and red dashed lines indicate the trend lines in $O_2$ inventory and vertical stratification at 100 m (estimated as the local vertical density gradient at 100 m). Note the strong anticorrelation between $O_2$ anomalies and stratification anomalies (R=-0.63). The location of the northern AS box is shown in Fig 4a.

## 3.2   Drivers of northern AS ocean deoxygenation

As oxygen equilibrates relatively rapidly at the air-sea interface relative to ocean circulation timescales, observed surface oxygen concentrations are generally close to their saturation levels ($O_2^{sat}$). In the ocean interior, oxygen is depleted because of biological respiration. Therefore, changes in dissolved oxygen concentrations in the ocean interior express either changes

in oxygen saturation levels (solubility effect) or changes in the accumulated oxygen deficits associated with respiration. The latter term, measured by AOU (AOU=$O_2^{sat}$-$O_2$), depends on both biological activity and ventilation. Here, we explore how these different components contribute to the simulated oxygen changes. In the northern AS (north of 20°N), oxygen inventory drops by over 6% per decade between 100 and 1000 m. (Fig. 6). We found this trend to be statistically significant at 95% confidence interval. The deoxygenation trend seems to emerge mostly from changes in the apparent oxygen utilization (AOU)

as $O_2^{sat}$ shows no similar decline and even slightly increases during the study period (Fig. 6). This is in contrast to the surface ocean (top 30 m) where the drop in $O_2^{sat}$ explains the majority (over 70%) of the simulated $O_2$ decline (Fig S19, SI). To further explore the drivers of ocean deoxygenation in the northern AS, we performed an $O_2$ budget analysis north of 20°N in the 100-1000 m layer (Fig. 7). To this end, we quantified in the same layer the cumulative $O_2$ anomalies associated with the transport and the biology sources-minus-sinks terms (see the details of the $O_2$ mass balance equation in the SI). This analysis

indicates that most of $O_2$ decline in the northern region is associated with a drop in ventilation (transport), with $O_2$ anomalies

driven by biological consumption showing no such a drop over the study period (Fig.7). The ventilation anomalies appear to also dominate the strong interannual and decadal variability in $O_2$ (Fig. 7). The reduction in ventilation in the northern AS is particularly strong in the 1980s and 1990s. In contrast, no such decline can be seen in the last decade of the simulation despite an important modulation by interannual fluctuations. Splitting the ventilation contribution into vertical mixing and advection

5   driven parts reveals that the decline in $O_2$ is primarily caused by a reduction in vertical mixing (Fig 7). Next, we explore how changes in atmospheric forcing may have contributed to the ventilation reduction evidenced by the $O_2$ budget analysis.

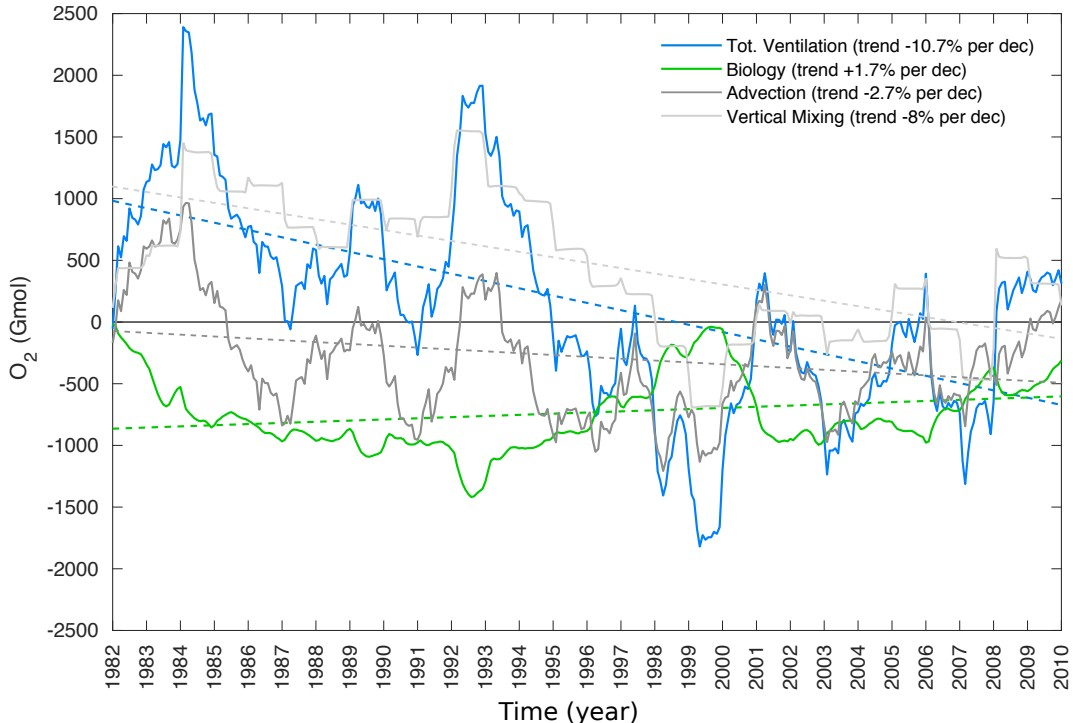

**Figure 7. Role of ventilation and biology in northern AS deoxygenation.** Cumulative $O_2$ anomalies associated with ventilation (blue) and biological (green) changes between 100 and 1000 m in the northern AS (location indicated in Fig 4a). The contributions of advection and subgrid vertical mixing are shown in dark grey and light grey, respectively. The dashed lines indicate the trend lines associated with the different sources.

### 3.3   Impact of changes in atmospheric forcing

To explore the impacts of changes in atmospheric forcing on the oxygen distribution in the Arabian Sea, we contrast oxygen trends in the control run to trends in the different sensitivity experiments. More concretely, the effect of surface warming on

10   oxygen is quantified by subtracting the trends in the no-warming simulation $S_{hclim}$ from those in the control run. Similarly, subtracting the trends in the no-Gulf-warming simulation $S_{hclim\_AG}$ from trends in the control run allows us to measure the relative importance of the fast warming of the Gulf on the deoxygenation in the northern AS. Finally, by subtracting oxygen

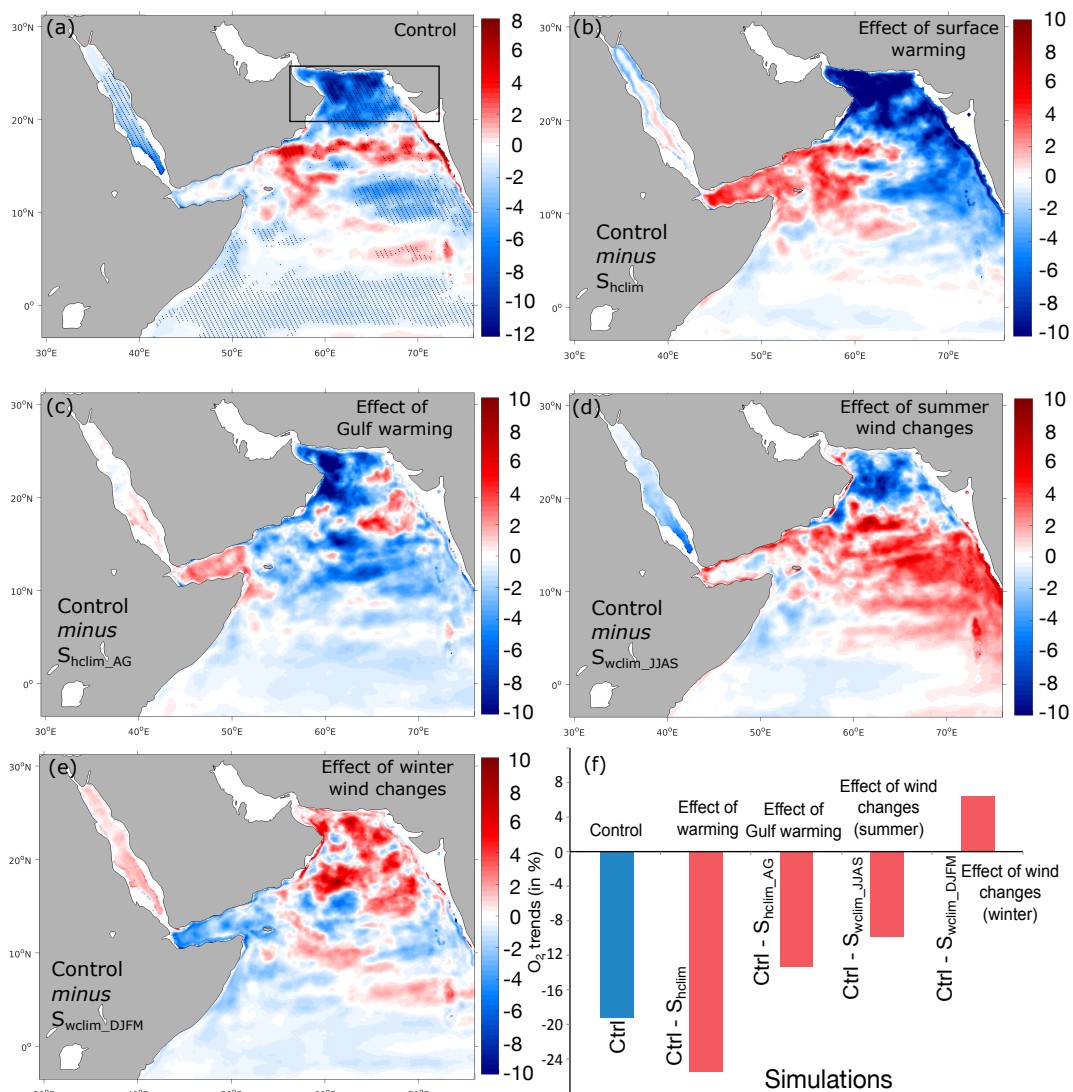

**Figure 8. Effects of different atmospheric forcing perturbations on deoxygenation rates.** (a) Linear trends in $O_2$ inventories in the 100-1000 m layer in the control run (in % decade$^{-1}$). (b-e) Difference in $O_2$ inventory trends between the control run and the $S_{hclim}$ (i.e., effect of warming), $S_{hclim\_AG}$ (i.e., effect of Gulf warming), $S_{wclim\_JJAS}$ (i.e., effect of summer wind intensification) and $S_{wclim\_DJFM}$ (i.e., effect of winter wind changes) sensitivity simulations in the 100-1000 m layer (in % decade$^{-1}$). (f) Difference in $O_2$ inventory trends in the northern AS box between the control and the different sensitivity simulations (in % of the mean $O_2$ inventory over the 1982-2010 period).

trends in the $S_{wclim\_JJAS}$ and $S_{wclim\_DJFM}$ simulations from the control trends we are able to quantify the effects of summer and winter monsoon wind changes, respectively, on the simulated deoxygenation.

This analysis reveals that surface warming causes a substantial decline in the northern AS oxygen inventory of nearly 25% in the 100-1000m layer between 1982 and 2010 relative to the no warming case (Fig. 8). Indeed, oxygen inventory increased

north of 20°N by over 6% between 1982 and 2010 under climatological heat fluxes as opposed to a decrease of over 18% in the control run during the same period (Fig S20, SI). Contrasting oxygen trends in the no-Gulf warming simulation to those simulated in the control run suggests that the fast warming of the Gulf has significantly contributed to northern AS deoxygenation. Concretely, the Gulf warming is associated with a decrease of the northern AS oxygen inventory of around

12% in the 100-1000m layer between 1982 and 2010 relative to the no-Gulf warming case (Fig. 8). Indeed, in the absence of Gulf warming the oxygen inventory decreased north of 20°N by only around 6% between 1982 and 2010, a rate that is twice as weak as that simulated in the control run during the same period (Fig S20, SI). The intensification of summer monsoon winds also appears to contribute to simulated deoxygenation in the northern AS, although to a lesser extent (Fig 8). Indeed, summer wind changes are associated with a decrease of the northern AS oxygen inventory of around 9% in the 100-1000m

layer between 1982 and 2010 relative to the no summer wind change scenario (Fig 8). Indeed, oxygen decreases in the northern AS by around 9% between 1982 and 2010 under climatological summer winds, a rate that is nearly 50% weaker than in the control run during the same period (Fig S20, SI). Finally, contrary to surface warming and summer monsoon intensification, changes in winter monsoon winds do not contribute to deoxygenation trends in the northern Arabian Sea. Indeed, in the absence of winter wind changes, deoxygenation in the northern AS is maintained and is even slightly stronger relative to the control

run (Fig 8).

     In summary, we conclude that deoxygenation in the northern AS is essentially caused by surface warming and that the fast warming of the Gulf plays an important role in this trend. While summer monsoon intensification - to a lesser extent- contributes to deoxygenation in the northern AS, changes in winter monsoon winds have a smaller effect on northern AS oxygen and tends to oppose deoxygenation trends there. Next, we explore the mechanisms through which surface warming and

enhanced summer monsoon winds reduce thermocline ventilation, causing upper ocean deoxygenation in the northern AS.

### 3.4    Mechanisms of ventilation reduction

The ventilation of the northern AS upper thermocline is predominantly sensitive to: (i) the intensity of vertical mixing particularly associated with winter convection (McCreary et al., 2013; Resplandy et al., 2012) , (ii) the magnitude of export and subduction of dense Gulf water (Lachkar et al., 2019) and (iii) the vertical displacement of the thermocline causing changes

in the oxycline depth (Vallivattathillam et al., 2017). Here, we explore how these ventilation mechanisms have responded to changes in atmospheric forcing.

     The ocean surface warming results in an increase in vertical stratification (here estimated using the local vertical density gradient at $100\,m$) that is particularly important in the northern AS with stratification at $100\,m$ increasing on average by nearly 4% per decade north of 20°N (Fig 6 and Fig 9). This positive trend is mostly induced by rapid stratification increases in the

1980s and 1990s with little change in the 2000s, mirroring the evolution of $O_2$ (Fig 6). The stratification does not increase (even slightly decreases) in the northern AS when the heat fluxes are set to be climatological (Fig S21, SI). This confirms that surface warming is responsible for the stratification increase that inhibits vertical mixing and contributes to the reduction of ventilation revealed by the $O_2$ budget analysis (Fig 7). Additional evidence for the reduced vertical mixing can be seen from

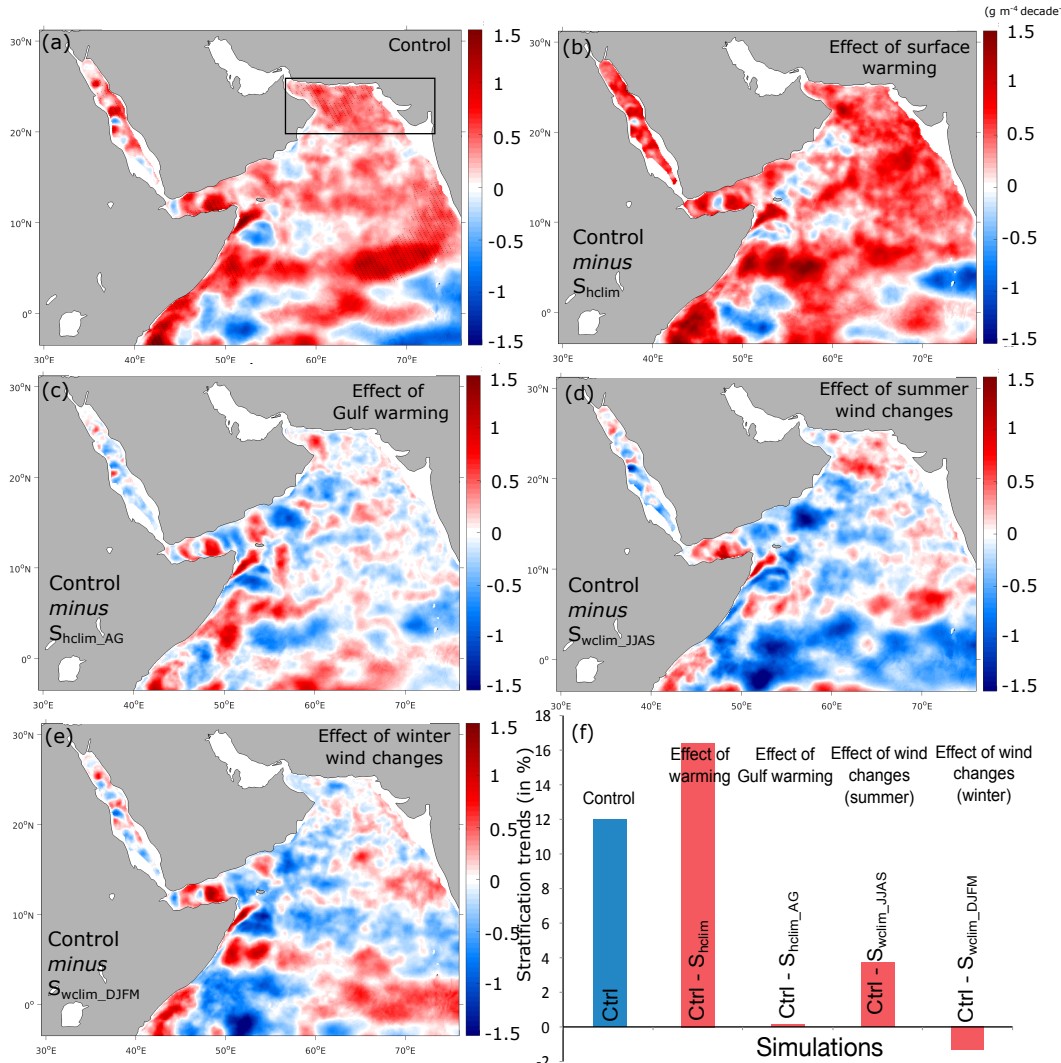

**Figure 9. Effects of different atmospheric forcing perturbations on upper ocean vertical stratification.** (a) Linear trends in vertical stratification at $100\,\mathrm{m}$ in the control run (in $\mathrm{g\,m^{-4}}$ decade$^{-1}$). (b-e) Difference in trends in stratification at $100\,\mathrm{m}$ between the control run and the $S_{hclim}$ (i.e., effect of warming), $S_{hclim\_AG}$ (i.e., effect of Gulf warming), $S_{wclim\_JJAS}$ (i.e., effect of summer wind intensification) and $S_{wclim\_DJFM}$ (i.e., effect of winter wind changes) sensitivity simulations (in $\mathrm{g\,m^{-4}}$ decade$^{-1}$). (f) Difference in stratification trends in the northern AS box between the control and the different sensitivity simulations (in % of the mean stratification over the 1982-2010 period).

the analysis of trends in winter mixed layer depth (MLD) that shows a winter MLD shoaling by over $5\,\mathrm{m}$ in the northern AS between 1982 and 2010 (Fig S22, SI).

While both the warming of the Gulf and summer monsoon intensification contribute to deoxygenation in the northern AS, vertical stratification changes little in response to these two perturbations (Fig 9 and Fig S21). This suggests that other

mechanisms contribute to northern AS ventilation reduction besides vertical stratification enhancement. Lachkar et al. (2019) have shown that the AS OMZ can intensify in response to strong warming of the Gulf causing a reduced outflow of the Gulf water in the northern AS. Here, we find the depth of the Gulf water has shoaled locally by up to 20 m in the Gulf of Oman in the control run relative to the $S_{hclim\_AG}$ run (Fig S23, SI). This indicates an increase in the Gulf water buoyancy and a decline in its subduction to intermediate depths in the northern AS in agreement with the mechanisms described in Lachkar et al. (2019).

Finally, changes in monsoon winds are likely to affect oxygen levels in the upper ocean through their impact on the thermocline depth, and hence the depth of the oxycline. Here we analyze long-term trends in thermocline depth, noted D20 and represented by the depth of isotherm 20°C following previous studies (e.g. Schott et al., 2009). This analysis reveals a shoaling of this interface by over 3% (around 6 m) in the northern AS between 1982 and 2010 (Fig 10). Contrasting these trends to those simulated under climatological summer winds shows a strong sensitivity of this parameter to summer monsoon wind intensification. Indeed, under climatological summer monsoon winds the thermocline depth D20 shoals almost everywhere in the AS except in the northern region where a deepening is simulated (Fig S21, SI). This suggests that summer monsoon wind intensification causes the thermocline depth to rise in the northern AS and deepen elsewhere (Fig 10). This is likely due to enhanced open ocean upwelling (Ekman suction) in the north and downwelling (Ekman pumping) in the south (Fig S24). The shoaling of the thermocline depth contributes to lowering $O_2$ levels in the northern AS upper thermocline whereas its deepening south of 20°N contributes to oxygenate the upper ocean there. This analysis also shows that changes in winter monsoon winds are associated with a deepening of the thermocline in the northern AS (potentially linked to enhanced downwelling along the west coast of India) that contributes to oxygenate the upper thermocline (Fig 2, Fig 10 and Fig S17, SI).

In summary, the analysis of the sensitivity simulations suggests that recent deoxygenation in the northern AS has essentially been caused by surface warming, increasing stratification and inhibiting convective vertical mixing as well as increasing the buoyancy of the Gulf water, thus inhibiting its subduction to intermediate depths. The concomitant shoaling of the thermocline in the northern AS driven by summer monsoon wind intensification further contributes to lower $O_2$ in the upper ocean there (See Fig 11 for a visual depiction of the key mechanisms involved in northern AS recent deoxygenation).

## 4 Discussion

### 4.1 Role of biology

The relatively limited role of biology in the northern AS deoxygenation is a direct consequence of the minimal change the biological productivity has experienced in the region over the study period as well as due to the negative feedback of enhanced denitrification on $O_2$ consumption at depth. Indeed, the biological productivity has changed little in the northern AS as enhanced stratification on the one hand and increased summer upwelling on the other hand have opposing effects on nutrient supply to the euphotic zone (Fig 12). For instance, productivity increases substantially in the northern AS and across the domain in the absence of surface warming (Fig S25, SI). Conversely, under climatological summer winds productivity tends to decrease in most of the northern and western AS (Fig S25, SI). Finally, the enhanced denitrification associated with the deoxygenation trends reduces aerobic $O_2$ consumption and hence opposes deoxygenation in the northern AS (Fig S27, SI), a process

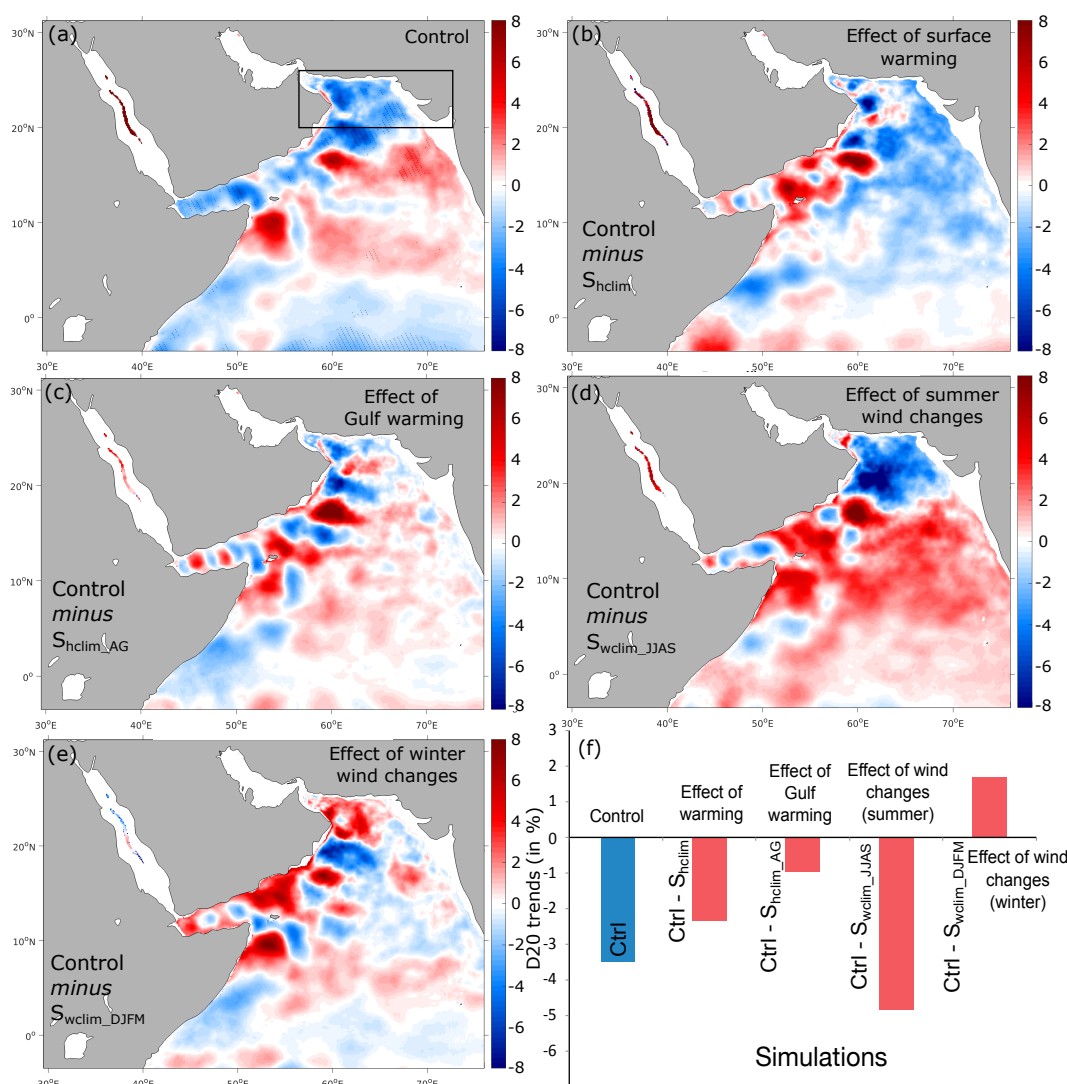

**Figure 10. Effects of different atmospheric forcing perturbations on thermocline depth.** (a) Linear trends in the depth of isotherm 20°C (D20) in the control run (in m decade$^{-1}$). (b-e) Difference in D20 trends between the control run and the S$_{hclim}$ (i.e., effect of warming), S$_{hclim\_AG}$ (i.e., effect of Gulf warming), S$_{wclim\_JJAS}$ (i.e., effect of summer wind intensification) and S$_{wclim\_DJFM}$ (i.e., effect of winter wind changes) sensitivity simulations (in m decade$^{-1}$). (f) Difference in D20 trends in the northern AS box between the control and the different sensitivity simulations (in % of the mean thermocline depth over the 1982-2010 period)..

previously shown to be important for the oxygen budget on long timescales (Oschlies et al., 2019). However, biology appears to play a more important role in the central and western AS as increased summer monsoon winds increase the productivity there (Fig 12 and Fig S25, SI). This productivity enhancement results in an increase in O$_2$ consumption that contributes to the simulated deoxygenation trends in these regions (Fig S25 and Fig S26, SI).

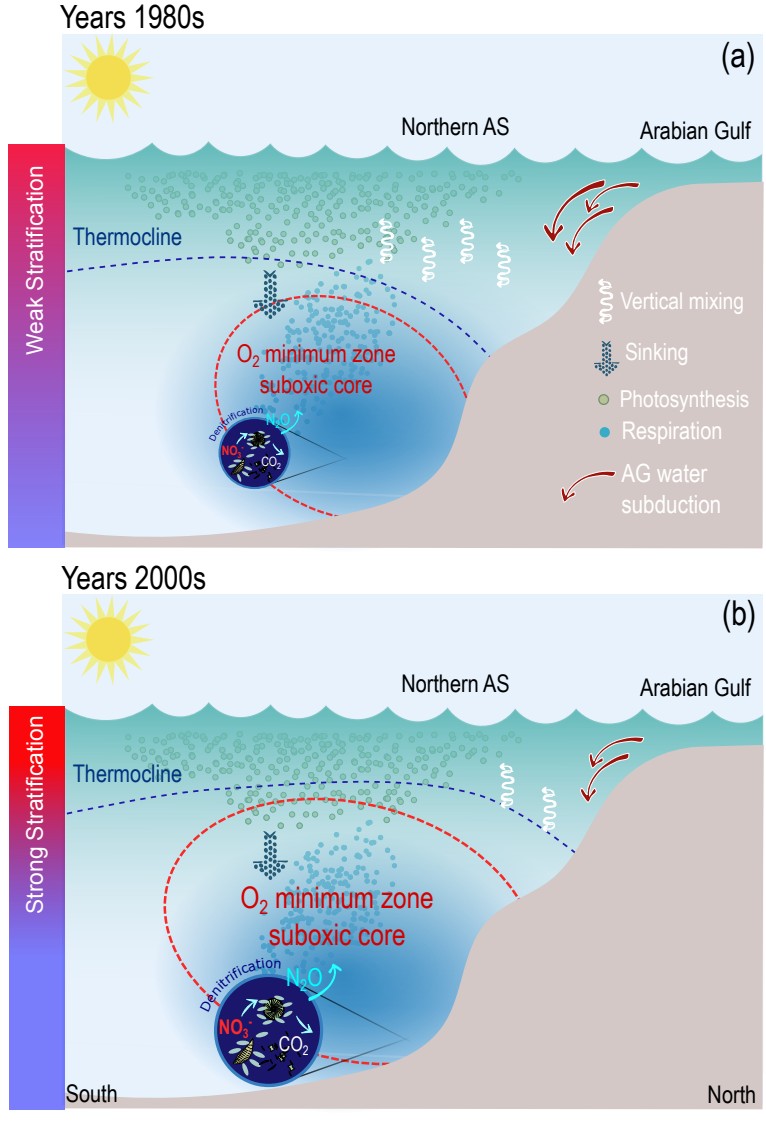

**Figure 11. A schematic summarizing the main processes responsible for the $O_2$ changes in the northern AS.** (a) Conditions in the early 1980s: cool conditions favor weak stratification, enhanced vertical mixing as well as subduction of high-density Gulf water in the northern Arabian Sea, resulting in a deeper and smaller OMZ suboxic core. (b) Conditions in the early 2000s: warm conditions enhance vertical stratification and cause a weaker vertical mixing and a reduced subduction of the Gulf water in the northern Arabian, resulting in an expansion of the suboxic core of the OMZ and an increase in denitrification. Stronger upwelling favorable winds also contribute to raise the thermocline depth and hence bring $O_2$-depleted waters upwards further near the surface.

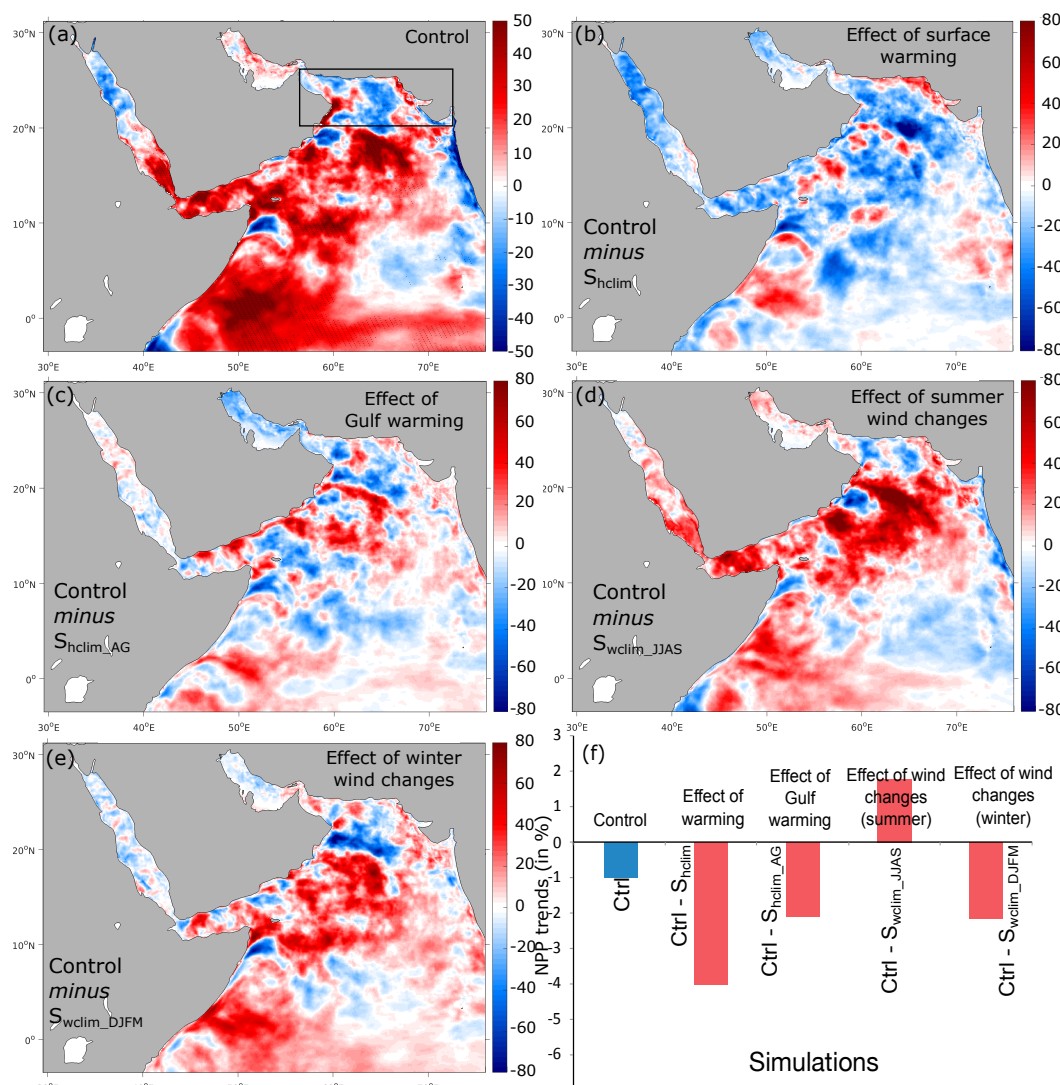

**Figure 12. Effects of different atmospheric forcing perturbations on net primary production (NPP).** (a) Linear trends in NPP in the control run (in mol $m^{-2}$ $yr^{-1}$ decade$^{-1}$). (b-e) Difference in NPP trends between the control run and the $S_{hclim}$ (i.e., effect of warming), $S_{hclim\_AG}$ (i.e., effect of Gulf warming), $S_{wclim\_JJAS}$ (i.e., effect of summer wind intensification) and $S_{wclim\_DJFM}$ (i.e., effect of winter wind changes) sensitivity simulations (in mol $m^{-2}$ $yr^{-1}$ decade$^{-1}$). (f) Difference in NPP trends in the northern AS box between the control and the different sensitivity simulations (in % of the mean NPP over the 1982-2010 period)..

## 4.2 Comparison with previous works

Roxy et al. (2016) have reported a decline in surface chlorophyll-a over the northern and western AS between 1998 and 2013. This appears to be inconsistent with results from the present study that show very weak NPP trends in the northern AS and even

a statistically significant increase in the western Arabian Sea off the coast of Somalia. There are two main differences between our study and the work of Roxy et al. (2016) that may explain this apparent inconsistency. First, in their analysis Roxy and colleagues have considered surface chlorophyll while we are analyzing trends in vertically integrated biological productivity. While these two quantities are usually strongly correlated, they are not identical, especially in tropical systems where deep chlorophyll maxima are commonly observed. Indeed, the trends in surface chlorophyll in our simulation are quite different and are much weaker in the open ocean in comparison to trends in NPP (Fig S28, SI). The second main difference concerns the study periods. Indeed, the satellite chlorophyll data presented in Roxy et al. (2016) is based on a different and shorter period (1998-2013) than in the present study (1982-2010). Interestingly, our model also simulates a decline of surface chlorophyll in the western Arabian Sea when the analysis is restricted to the period between 1998 and 2010 (Fig S28, SI). We believe the high sensitivity of the trends to the considered period of analysis is an indicator of the strong interannual and decadal variability in the region. This can also be seen in chlorophyll trends based on CMIP models presented in Roxy et al. (2016). Indeed, covering a longer period (1950-2005) these time series reveal strong decadal variability with a decline in surface chlorophyll from the 1950s to the late 1970s followed by no significant trend or even a slight increase over the period from 1980 and 2005.

Our finding that the suboxic volume and denitrification are highly sensitive to deoxygenation is consistent with previous studies that suggest high vulnerability of suboxic zones to small changes in the ocean's $O_2$ content. For instance, the world's largest suboxic zone in the Pacific Ocean was shown to vary in size by a factor two in model reconstructions of historical oxygen changes (Deutsch et al., 2011). Finally, the weaker sensitivity of the volume of hypoxia to deoxygenation in the AS is consistent with previous studies suggesting a generally weaker sensitivity of the hypoxic volume to deoxygenation. For instance, it was estimated that a decrease of the upper ocean $O_2$ concentration by 5 mmol m$^{-3}$ could lead to a tripling of the suboxic volume and only 10% increase in the hypoxic volume (Deutsch et al., 2011).

## 4.3  Future deoxygenation in the northern AS

As the AS and the Gulf continue to warm under future climate change, deoxygenation may continue in the northern AS region. Indeed, modeling studies suggest further future ocean deoxygenation in the region. For instance, CMIP5 models project a drop of $O_2$ by up to 20 mmol m$^{-3}$ in the Sea of Oman in the layer between 200 m and 600 m by the end of the century under the RCP8.5 emission scenario (Bopp et al., 2013). The CMIP6 multi-model ensemble average indicates an even stronger oxygen decline of more than 30 mmol m$^{-3}$ by 2100, in the northern AS between 100 m and 600 m, essentially driven by an increase in the AOU (Kwiatkowski et al., 2020). These changes can have dramatic impacts on $O_2$-sensitive species and nitrogen and carbon cycling in the region. Furthermore, the impacts of these $O_2$ changes on marine ecosystems are likely to be exacerbated by concurrent stressors such as warming, declining productivity and ocean acidification (Levin, 2018; Deutsch et al., 2015; Gobler and Baumann, 2016; Miller et al., 2016; Bianchi et al., 2013). In the central and southern AS, however, global model projections show no consistent trends or even a slight oxygenation (Bopp et al., 2013; Kwiatkowski et al., 2020). This may be due to the relatively important future productivity decline these models predict for the western and central AS that may reduce $O_2$ consumption there and compensate for the effect of reduced ventilation.

## 4.4 Caveats and limitations

Our study has a couple of caveats and limitations. Among the study's main limitations is the relatively short simulation period that precludes the attribution of the documented $O_2$ changes to climate change vs. natural variability. Previous observations suggest that the natural variability in $O_2$, dominated by interannual and decadal oscillations, can locally be stronger than the long-term trends associated with climate warming (Whitney et al., 2007; Cummins and Ross, 2020). Strong modulation of interrannual variability in hypoxic and suboxic volumes by decadal oscillations has been documented in previous studies (e.g., Deutsch et al., 2011, 2014). This complicates the detection and attribution of long-term responses to climate change (Bindoff et al., 2019). Therefore, it is possible that an important fraction of the trends simulated here is associated with natural variability as it has been shown that the emergence of the climate change signal among the internal variability range is generally slow for oxygen, with only a small fraction of the ocean experiencing emergence before the end of the century (Frölicher et al., 2016). According to the same study, the earliest emergence of the $O_2$ signal in the AS is expected to occur only by the middle of the current century, although this may also happen significantly earlier according to other studies (Long et al., 2016; Hameau et al., 2019).

An additional caveat of the study is related to the uncertainty around the recent monsoon wind changes. Indeed, no clear consensus emerges from the different studies that have explored the recent and future wind changes in the region, as some point to an intensification (e.g., Wang et al., 2013) whereas others suggest a weakening (e.g., Swapna et al., 2017) or a poleward shift (e.g., Sandeep and Ajayamohan, 2015). Some of this uncertainty may be associated with the strong decadal variability that modulates the surface winds in the region and may lead to aliasing of long-term trends. An additional source of uncertainty is the lack of observations in the AS region that results in not very well constrained reanalyses. To test this, we contrast the trends in surface winds estimated over the study period (1982-2010) from three reanalysis products: ERA-Interim (here used to force the model), the National Centers for Environmental Prediction reanalysis II (NCEP-2) and the Japanese 55-year Reanalysis Project (JRA-55). This comparison reveals important discrepancies among the three products (Fig S29, SI). Indeed, while the NCEP2 winds show a modest increase in upwelling-favorable winds in the western Arabian Sea, there seems to be no such an increase (and even a slight weakening) in the JRA55 winds. While we acknowledge this uncertainty as one of the caveats of the study, we also believe this has likely limited implications for the conclusions of the study as we demonstrate here that surface warming is the dominant factor in the northern Arabian Sea deoxygenation, with the surface wind changes playing only a secondary role.

Other limitations of the study pertain to the biogeochemical model assumptions and model forcing. For instance, the lack of a representation of some major limiting nutrients such as iron, silicate and phosphate can potentially cause biases in regions where these nutrients contribute to limit biological production (e.g., off Somalia). However, previous studies suggest nitrogen is the main limiting nutrient in the Indian Ocean at larger scales (Koné et al., 2009). Other major model-related limitations concern the lack of representation of important biogeochemical processes such as $N_2$ fixation and the crude representation of microbial respiration in the model. Yet, on the one hand recent studies suggest that $N_2$ fixation has a limited effect on the AS OMZ as it constitutes only a negligible proportion of new nitrogen there (Guieu et al., 2019). On the other hand, simple

representations of microbial respiration that miss potentially important biogeochemical feedback are a common problem in most existing biogeochemical models (Oschlies et al., 2018; Robinson, 2019). Additional work that combines expanding the current oxygen-measurement system and improving the complexity of microbial respiration in numerical models is needed to reduce biases in model estimates of deoxygenation (Oschlies et al., 2018). Finally, as the model lateral boundary conditions for nitrate and oxygen were set to be climatological, the effects of potential changes in either $O_2$ or $NO_3^-$ at the domain southern boundary at 31°S are not taken into account. Previous studies (e.g., Keller et al., 2016; Fu et al., 2018) have shown that remote biological processes in the Southern Ocean can significantly affect oxygen levels and OMZs in the tropics either through trapping of nutrients in the Southern Ocean or through changes in $O_2$ levels of locally formed mode and intermediate waters. However, these remote influences have been shown to affect oxygen in the tropics (including the Arabian Sea OMZ region) only on timescales of several decades to centuries (Keller et al., 2016). Therefore, given the short time period considered in the present study we believe the lack of interannual variability at the domain lateral boundaries to have limited impact on our results.

## 5 Summary and Conclusions

We reconstruct the evolution of dissolved oxygen in the AS from 1982 through 2010 using a series of hindcast simulations performed with an eddy-resolving ocean biogeochemical model forced with ERA-Interim atmospheric reanalysis. We find a significant thermocline deoxygenation in the northern AS, with the ocean $O_2$ content dropping by over 6% decade$^{-1}$ in the 100-1000 m layer. These changes are accompanied by a statistically significant increase of the volume of suboxia ($O_2 < 4$ mmol m$^{-3}$) and denitrification by up to 30% and 40% over the study period, respectively. Using a set of sensitivity simulations we demonstrate that deoxygenation in the northern AS has been caused essentially by a widespread warming of the sea surface, in particular in the Gulf, causing a reduction in the ventilation of the subsurface and intermediate layers. Additionally, we show that a concomitant summer monsoon intensification over the study period has enhanced the ventilation and hence the oxygenation of the upper ocean south of 20°N but has contributed to deoxygenation in the northern AS. This is because on the one hand surface warming: (i) increases vertical stratification, thus reducing vertical mixing and (ii) increases the Gulf water buoyancy, inhibiting its subduction to intermediate depths. On the other hand, summer monsoon wind intensification causes the thermocline depth to rise in the northern AS and deepen elsewhere, thus contributing to lowering $O_2$ levels in the upper layers north of 20°N and increasing it in the rest of the AS. Our findings confirm that the AS OMZ is strongly sensitive to upper-ocean warming and concurrent changes in the Indian monsoon winds. Our results also demonstrate that changes in the local climatic forcing play a key role in regional dissolved oxygen changes and hence need to be properly represented in global models to reduce uncertainties in future projections of deoxygenation.

*Code availability.* The model code can be accessed online at http://www.croco-ocean.org/.

*Author contributions.* ZL conceived and designed the study. MM and MA performed the simulations and carried out the model output analysis. ZL wrote the manuscript with contributions from all co-authors.

*Competing interests.* The authors declare that they have no conflict of interest

*Acknowledgements.* Support for this research has come from the Center for Prototype Climate Modeling (CPCM), the New York University Abu Dhabi (NYUAD) Research Institute. Computations were performed at the High Performance cluster (HPC) of NYUAD, Dalma. We thank the NYUAD HPC team for technical support. The authors are also grateful to three anonymous reviewers for their constructive comments that helped improve the paper.

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
