# Peer review of "Fast local warming is the main driver of recent deoxygenation in the northern Arabian Sea"

_Biogeosciences, 2020_

## Short Comment (SC1) · 2 Oct 2020

The findings of the paper are interesting. However, two relevant papers we found missed in citing in this work, as follows.

Sreeush M. G., R. Saran, V. Valsala, S. Pentakota, K.V. S.R. Prasad, R. Murtugudde Variability, trend and controlling factors of Ocean acidification over Western Arabian Sea upwelling region, 2019, Marine Chemistry doi.org/10.1016/j.marchem.2018.12.002

Praveen, V., V. Valsala, R. Ajayamohan, and S. Balasubramanian Oceanic mixing over

northern Arabian Sea in a warming scenario: Tug of war between wind and buoyancy forces. Journal of Physical Oceanography., https://doi.org/10.1175/JPO-D-19-0173.1

In the first one, the paper addresses trends in Arabian Sea acidification over 50 years, which in the present paper, the acidification is mentioned relating to the DO trends. The above paper could be mentioned/cited as that is the first-ever paper available which looks at long-term trends in the Arabian Sea acidification.

In the second paper above, the increase in winds (as mentioned in this paper too) how it impacts the ocean mixing due to counteracting forcing of buoyancy and production is discussed. The wind intensified deoxygenation found north of 20N in this paper is worth pointing out in view of the above paper which exclusively discussed the mechanistic point of view of with exhaustive downscaling modeling of CMIP-5 boundaries. The paper is also worth mentioning/citing too.

---

## Referee Comment (RC1) · Anonymous Referee #1 · 7 Dec 2020

Given their ecological and socio-economical impacts, it is a must to better describe and understand the present and future evolution of the major Oxygen Minimum Zones (OMZ). Despite large uncertainties related to their sparse and inhomogeneous coverage, in-situ observations suggest that suboxic conditions in the Arabian Sea (AS) may have expanded during the past decades. The present study uses a high-resolution ocean regional model forced with an atmospheric reanalysis to assess the mechanisms responsible for the oxygen decline over the AS. Their results point towards a major influence of reduced ventilation through enhanced upper ocean stratification on the northern AS desoxygenation, further strengthen by summer monsoon wind changes. While previous studies recognised that these two physical processes (along with biological ones) may contribute to long-term oxygen evolution in the AS, this study offers for the first time a reliable quantification of the respective influence of each of these processes over the historical period. I hence believe that this is a very interesting and important study that deserves to be published in Biogeosciences. However, there are a number of major issues listed below that need to be addressed before publication.

Major comments: A. Validation: Most of the validation section evaluates the model ability to simulate the seasonal cycle in the Arabian Sea. Because this paper mainly focuses on the oxygen long-term evolution, I would recommend the authors to (1) reduce the validation of the seasonal aspects and (2) expand the validation of the long-term trends as follows: (1) Figure 1 and 2: I would move these Figures in the SI. Given that the authors use a surface restoring the observed salinity and temperature, this is not surprizing that the seasonal cycle of the model SSS and SST agree well with observations. In addition, I feel that Figure 2 is not central to the present study and should not be included in the core of the paper. (2) Figure 3: Keep at least the lower panels (Chl validation) but improve the color scale chosen to emphasize the regional contrasts. (3) Figure 4: Add meridional sections of O2 concentration along with hypoxic/suboxic volume to evaluate the O2 structure at depth in both model and observations. (4) Evaluation of long-term evolution (extra analyses): Given that the reliability of the results presented here relies on the strong assumption that the model reproduces accurately the long-term evolution in the AS, the model ability to simulate this evolution should definitely be assessed in greater details. For instance, this paper should include an evaluation of the long-term evolution of AS maps and yearly time series in the northern/western AS for model and observations of (1) Chl over the common period (1997-2010) where observations and model data are available (using OC-CCI product for instance), (2) SST over the entire period (using ERA5 or HadISST for instance), (3) static stability index (as in Roxy et al. 2016) over the entire period (using an ocean reanalysis such as SODA or ORAS4/5) and (4) wind trends from several .wind products. Finally, I would also like to see an attempt to evaluate the model O2 evolution over wide boxes with in-situ observations (from Ito or Schmidtko et al. 2017 for instance).

This would allow strengthening the reliability of the model results regarding the oxygen decline in the AS and the related mechanisms discussed in the paper. B. Choice of the sensitivity experiments: It is not obvious to be why the authors decided only to investigate the role of the summer monsoon wind changes on the O2 concentration. Looking at Figure S10, it appears to me that winter monsoon wind changes are also significant, with a strong wind decrease in the northern AS. This decrease is expected to reduce convective mixing and hence ventilation. I would recommend the authors to perform an additional experiment to evaluate this impact and report it in the manuscript. C. Summary of the main results: I also believe that the authors should definitely include a schematic summarizing the main processes responsible for the long-term O2 evolution in the AS. This would really be helpful for the reader to grasp the salient results of the present study.

Detailed comments: Abstract: P1 L2-3: Given the strong observational uncertainties, I would recommend the authors to lower down their claim about a decline in AS O2 over the past decades. Use for instance "suggest" instead of "show". More generally, I would recommend the authors to refer to the latest SROCC report on the ocean changes (Bindoff et al. 2019) to refine their statements on O2 trends in observations. This report indeed concludes that: ". . . the challenges of data sparsity, regional differences and the relatively large uncertainties on the oxygen changes across different studies, but also recognising that oxygen declines are significantly different to zero, leads to medium confidence in the observed oxygen decline. Âż In addition, this report does not mention the AS as one of the regions where the O2 decline is the strongest and best observed (they rather mention Southern Ocean, equatorial regions, North Pacific and South Atlantic). P1 L5: "while summer monsoon winds have intensified" : While this is true in the atmospheric reanalysis used in the present study, this intensification of summer monsoon winds may not be evident in all wind products. Indeed, several studies suggest that decadal wind variations are not well constrained in reanalyses, which may lead to strong uncertainties in long-term wind trends over the AS. In addition, most studies rather report a decrease of the Indian summer monsoon circulation over the historical period (e.g. Swapna et al. 2017). This uncertainty should be discussed in the manuscript. P1 L5 : "reconstruct" I don't like the use of this term here, as their is not assimilation in the model. I would rather use "simulate". P1 L7 : Replace Âń observation-based Âż by Âń forced by an atmospheric reanalysis Âż P1 L12 : Âń in particular in the Gulf Âż : The gulf warming contributes to 1/3 of the O2 decline through stratification if I get it well from the bottom panels of Figure 11. I would rather use Âń including in the Gulf Âż for not over-emphasizing the role of the Gulf warming.

Introduction : P2 L2-9: I would recommend the authors to also refer here to the latest SROCC report on the ocean changes (Bindoff et al. 2019) to refine their statements on the model projections and related uncertainties. P2 L13: "hypoxic events" Do you mean "anoxic events"? P2 L16-35: Please also refer to SROCC 2019 report about the confidence we have these reported changes. P3 L8: Regarding the enhanced warming in the AS, you should refer to Gopika et al. (2020) that investigate this in details. P3 L12-14: Results from Roxy et al. (2016) which report a Chl decline over the western AS appears to be inconsistent with results from the present study. This inconsistency and its implication on the robustness of the authors conclusions should be discussed in Section 4.2. P3 L17-18: I would recommend the authors to remove the reference to Goes et al. (2005) as the period covered by this study is very short (and strongly aliased by the 1997 El Niño) and their analysis very local. Most studies rather report a decrease of the Indian summer monsoon circulation over the historical period (e.g. Swapna et al. 2017) as well as in the future (e.g. Sooraj et al. 2015). Our current knowledge on the summer monsoon long-term trends should be better summarized here. P3 L23: Remove "largely" P3 L29: Change ", in particular in the Gulf" into ", with a significant contribution from the Gulf warming"

Methods: P4 L17: I am wondering why the authors did not extend their simulation until present days as ERA-I forcing is available until 2019. This would have allowed the authors to extend their validation over a period with more observations and monitor natural and anthropogenic decadal variations in more details. I would like the authors

to clearly state why they did restrict their simulation to 2010 only. P6 L7-P8 L5: Given
the restoring to observed SST and SSS, the good agreement between observations
and model is far from surprizing here. A dedicated Figure evaluating the surface circu-
lation is not mandatory either given the scope of the paper. I would move these Figures
in the SI to allow more space to validate the long-term evolution of key variables in the
model. P8 L14-15: I would add two meridional sections to Figure 4 showing the ability
of the model to simulate the oxygen vertical structure in the AS. P9 L5-12: While this
paper discusses the mechanisms responsible for the long-term oxygen changes in the
AS, this small paragraph is the only place where the authors evaluate the ability of
their model to simulate the AS long-term changes. I would recommend the authors to
reduce their seasonal validation and expand the validation of the simulated long-term
evolution in the AS. First, I find the authors methodology to derive long-term trends very
rough (difference between first and last 5 years of the period considered). Why not try-
ing to evaluate a proper linear trend with some significance level? In addition, why not
trying to build an average evolution in observations in wide boxes (northern vs/southern
AS) rather than a point-wise comparison? The authors also argue that "Contrasting the
long-term temperature changes in the top 200 m in the model and in the observations
reveals an overall good agreement in terms of the magnitude and the patterns of upper
ocean warming in both summer and winter seasons (Fig S7 and Fig S8, SI). Âż I am
not sure to agree with this statement as Fig. S7 and S8 do not show any colorbars and
are very patchy. I believe the authors can considerably improve this paragraph and
strengthen the reliability of their simulation by extending the validation of the model
long-term evolution to other parameters, for which observations or reconstructed prod-
ucts are available. Maps of trends and yearly time-series of key parameters could be
compared for model and observations for (1) SST, (2) upper ocean stratification, (3)
winds (for several products as these products are not well contrained) and (4) maybe
Chl at least over the common period where model and observational data from OC-CCI
are available (to be able to compare results with Roxy et al. 2016). It would also be
worth trying to compare the oxygen trends over key regions (northern/southern AS) in

model and observational products such as those used in Ito et al. and Schmidtko et al. (2017). Indeed, while I find their analyses on the mechanisms driving the oxygen decline in the model very convincing, these results are not supported by observational evidences. This should be done whenever possible.

Results: P9 L26-27 L30-31: How these numbers do compare with observed estimates globally and regionally? Figure 8bc: Plot a frame on Figure 6 to indicate the averaging region. It would also be interesting to add a time series on these panels showing the upper ocean stratification changes (for instance the static stability index). If ventilation dominates, we should expect this physical index to mirror the oxygen changes at interannual and longer timescales. This model stratification index could be further compared with observational estimates. Figure 9: Replace x-time axis by years (as in Figure 8) instead of months as currently done. Indicate over which region the analysis is performed and refer to Figure 6 to show this region. P14 L10-13 and Fig S9: Why does vertical and lateral components evolution mirror each other? Total advection term is actually a small residual between these two large terms. Can you explain? P14 L15-17: This SST trend in the model could be compared with observations (even if we expect a good agreement because of the relaxation term used) P14 L17-18: The authors argue here that wind changes are particularly prominent during the summer monsoon. However, Figure S10 suggests that winter wind changes also strongly contribute to the annual wind changes shown on Figure 10b. First, I don't understand why the authors decided from that point to focus on the impact of summer wind changes and completely disregard winter wind changes. As previously noted, circulation weakening during the winter monsoon should lead to reduce convective mixing and contribute to reduce the ventilation in the northern AS. I recommend the authors to investigate this potential mechanism futher by performing an additional sensitivity experiment. Second, because of the scarcity of atmospheric observations in the Indian Ocean, decadal to multi-decadal wind signals in this region are not well constrained in atmospheric reanalyses. I would suggest the authors to compare summer and winter wind trends shown on Figure S10 to other reanalyses products (JRA55 or NCEP2 for instance) to

ascertain if the reported wind changes are a robust feature in all products. Figure 11: As they are shown at different places in the manuscript, it is not easy to compare results from the sensitivity experiments (shown in Figure 11) with those of the control simulation (which are shown earlier in the manuscript, i.e. Figure 6). I would recommend the authors to either re-add Figure 6a to Figure 11 to ease comparison or to directly show the difference between the sensitivity experiments and the control simulation. This would ease the interpretation of the results derived fro mthe sensitivity experiments. P16 L5-8 and Figure 12: In addition to Figure 12a, I would here show yearly time series of the stratification index against O2 evolution in the control simulation. This would allow to give some hints on whether the ventilation mechanism operating at long timescales also operate at interannual and decadal timescales, i.e. a strong anti-correlation between between these two time series would further strengthen the case for a strong control of ventilation on oxygen concentration at different timescales. P19 L17-23: The upward trend in surface Chl simulated in the model is opposite to what observed trends and model projections (see Roxy et al. 2016). This discrepancy and its implication should be addressed in the discussion section.

Discussion: P21 L5-7: Check consistency of wind trends in other atmospheric reanalyses (see above). To me, summer monsoonal winds rather weakened over the past decades in observations. Also discuss also here apparent inconsistency with Roxy et al. (2016) study (decline in PP over recent decades). P21 L18-23: This is true but also mention that these models project a weakening of the summer monsoon circulation (e.g. Sooraj et al. 2015), which is not the case in the present study.

References: Bindoff, N.L., W.W.L. Cheung, J.G. Kairo, J. Arístegui, V.A. Guinder, R. Hallberg, N. Hilmi, N. Jiao, M.S. Karim, L. Levin, S. O'Donoghue, S.R. Purca Cuicapusa, B. Rinkevich, T. Suga, A. Tagliabue, and P. Williamson, 2019: Changing Ocean, Marine Ecosystems, and Dependent Communities. In: IPCC Special Report on the Ocean and Cryosphere in a Changing Climate [H.-O. Pörtner, D.C. Roberts, V. Masson-Delmotte, P. Zhai, M. Tignor, E. Poloczanska, K. Mintenbeck, A. Alegría, M. Nicolai, A.

Okem, J. Petzold, B. Rama, N.M. Weyer (eds.)]. 2019. Gopika, S., Izumo, T., Vialard, J., Lengaigne, M., Suresh, I., & Kumar, M. R. (2020). Aliasing of the Indian Ocean externally-forced warming spatial pattern by internal climate variability. Climate Dynamics, 54(1-2), 1093-1111. Sooraj, K. P., P. Terray, and M. Mujumdar (2015), Global warming and the weakening of the Asian summer Monsoon circulation: assessments from the CMIP5 models, Clim. Dyn. 45, 1–20.
 Swapna, P., Jyoti, J., Krishnan, R., Sandeep, N., & Griffies, S. M. (2017). Multidecadal weakening of Indian summer monsoon circulation induces an increasing northern Indian Ocean sea level. Geophysical Research Letters, 44(20), 10-560.

---

## Short Comment (SC2) · 1 Jan 2021

Comments on "Fast local warming of sea-surface is the main factor of recent deoxygenation in the Arabian Sea" by Lachkar et al.

Authors have used a suit of eddy resolving ocean model simulations to study the decline of O2 in the northern Arabian Sea during recent decades. Authors find that the reduced ventilation caused by the fast warming of ocean surface as the major reason for the decline of O2, with contributions from changes in the summer monsoon winds. Though the topic is of high relevance and the model based study approach is sound, the analysis and presentation are not convincing.

1. The central aspect of this study is reduction in O2 due to reduction in upper ocean ventilation. However, this manuscript do not even offer a basic explanation of the ventilation process and a definition of the ventilation term in the budget calculations. - A 2-3 sentence summary of Fig.1 in Oschiles et al. (2018, paper cited in the manuscript) in the introduction will be ideal. - Please explain how is ventilation defined in the O2 budget. - Why words like "little", "slight" and "dominates" are used instead of actual values of budget terms while discussing about the O2 budget?

2. The connection between changes in the upper ocean mixing process and thermocline depth to the change in ventilation is not presented convincingly. - How is vertical stratification defined? - A basic discussion of how the increased stratification leads to reduced ventilation is required here. This is a serious omission considering this is the major mechanism used to explain decline in O2. - How representative is 20 degree C isotherm as thermocline depth in this region (Fig.12)? Does a temperature gradient/slope based criteria (Fiedler, 2010) offer a better estimate of thermocline depth? - What leads to the shoaling of thermocline in the northern AS with summer monsoon wind intensification? - Is there a compensating effect between the surface warming induced vertical stratification and increased vertical mixing from increase in the wind speed (see reference 'c' in item 4 below)? - The impact of increase in stratification on the reduction of ventilation and the resulting reduction in the O2 concentration is not shown quantitatively.

3. Ambiguous or missing description of parameters/features. - What is the vertical resolution of the ocean model for "typical" (say 3000 m depth) water depth in the northern AS? Does it resolve the upper ocean and thermocline depths well? - 58 year Spinup: Which climatological forcing? Was SST and SSS restored? If so to which dataset? - What version of SODA data has been used and what is its resolution? - What frequency was the analyzed model data? Monthly mean? - What is the point in comparing SST and SSS from the model to the same dataset it has been restored to? Is observations used in WOA2013 (salinity) is also used in SODA reanalysis? -

Fig.2: Drifter measurements are for 15 m depth (examples: Section 1, last paragraph of Lumpkin et al. (2013), Section 2.2 last paragraph in Yu et al. (2019)). What depth are the model fields plotted here? When doing vector plots, please mention at what interval of model grid points (in X and Y directions) the arrows are shown. West India Coastal Current and circulation associated with the Lakshadweep High are very weak in the model compared to that in the drifter data and no explanation has been provided for this. - Fig.3: Large difference (model vs observation) in NO3 at the northern coastal-AS in Winter and at the south-west coast of Indian-Peninsula in Summer are not explained properly. - O2 Budget: Please explain the terms included in the budget (grouped as "biology" and "transport") in detail. Is the budget calculated online or offline? How well does the budget balance? - Layers of upper 200 m and below 200 m (Fig.6,7,8 and throughout the manuscript). How can authors justify analyzing data for a selected depth (like 100 m) to describe the properties in a "layer" (like 0-200m, Fig.6a)?. Isn't is appropriate to use a layer averaged (like in Fig.4) or integrated fields (like in Fig.9) instead? Please clearly mention how the data is processed in each of the figures mentioned above and justify the choice. What makes the 200 m depth as a layer separation depth? - Fig.6 panels a and b shows trends in percentage/decade but panel c shows actual values in nmol/m3/decade. It is very difficult to understand the context without visualizing the spatial pattern of actual trend in nmol/m3/decade. - Fig.7: How is the anomaly defined? Difference between monthly-varying model data, and mean of the time series? According to Fig.4 and definition, hypoxic (O2 < 60 nmol/m3) region is bigger than suboxic (O2 < 4 nmol/m3) region in an average (250-700 m) sense. So, naturally one expect the anomaly in hypoxic should be higher than that in the suboxic case but it is not true according to Fig.7 panels a and b. Authors, explain that the small patches of oxygenation in the eastern/central/southern region can make the volume of hypoxic region nearly constant compared to that of suboxic region to the north. But this is very difficult to comprehend from the percentage based trend shown in the panels 6a and 6b. Using percentage trend at two representative depth to explain the anomaly (in actual value/units) over a volume of water is very difficult for

a reader to follow and interpret. Why not use the vertically integrated O2 for 0-200 m range and 200-700 m range in Fig.6. - Sensitivity Experiments: Did any smoothing has been applied while modifying the forcing for a particular region (eg. Gulf) or time (eg. wclim_JJAS). If not, how strong was the impact of sudden jumps in the forcing field resulted from these modifications on the model solution? Labelling of experiments with 1986 fields as "clim" is misleading, instead use either "1986" or "normal" to indicate it is from a specific year. Also explicitly mention what is meant by the "control" run. - SST Warming: What is the mechanism behind widespread SST warming in the AS? Just citing few past studies is not sufficient since this is one of the core process which contributes to the O2 reduction. Please summarize major reasons for this warming where it is first discussed.

4. Missing citations of relevant literature. - Some of the recent studies which are highly relevant for the topics discussed/addressed in this manuscript (a: changes in monsoon winds, b: oceanic impacts of changes in monsoon winds, c:ocean mixing energetics of changes in monsoon winds, d&e: Oxygen minimum zone in the northern AS ) are not cited. a) Sandeep, S., and Ajayamohan, R.S., 2015: Poleward shift in Indian summer monsoon low level jetstream under global warming, Clim. Dyn. 45, 337-351; doi:10.1007/s00382-014-2261-y

b) Praveen, V., Ajayamohan, R.S., and Valsala, V., 2016: Intensification of upwelling along Oman coast in a warming scenario, Geophys. Res. Lett., doi:10.1002/2016GL069638

c) Praveen, V., Valsala, V., Ajayamohan, R.S., and Balasubramanian, S., 2020: Oceanic mixing over northern Arabian Sea in a warming scenario: Tug of war between wind and buoyancy forces, J. Phys. Oceanogr., 50(4), doi:10.1175/JPO-D-19-0173.1

d) Shenoy, D. M. et al., 2020: Variability of dissolved oxygen in the Arabian Sea Oxygen Minimum Zone and its driving mechanisms, J. Marine Sys., 204, https://doi.org/10.1016/j.jmarsys.2020.103310

e) Sarma et al 2020: Potential mechanisms responsible for occurrence of core oxygen minimum zone in the north-eastern Arabian Sea, Deep Sea Res. PartI, 165, https://doi.org/10.1016/j.dsr.2020.103393

- Add a citation or a statement in the acknowledgements for the SODA data (as shown in the link given below) https://climatedataguide.ucar.edu/climate-data/soda-simple-ocean-data-assimilation Carton, J.A. and B. Giese, 2008: A Reanalysis of Ocean Climate Using Simple Ocean Data Assimilation (SODA). Mon. Weath. Rev. , 136, 2999-3017.

As detailed above, this manuscript needs to be rewritten focusing on the details of analysis and presentation in order to make it a publishable one.

References: ————- Fiedler, P., 2010: Comparison of objective descriptions of the thermocline, Limnology and Oceanogr., https://doi.org/10.4319/lom.2010.8.313 Lumpkin, R., Grodsky, S. A., Centurioni, L., Rio M-H, Carton, J. A., and Lee D., 2013: Removing spurious low-frequency variability in drifter velocities. J. Atmos. Oceanic Technol., 30(2), 353-360, https://doi.org/10.1175/JTECH-D-12-00139.1 Yu, X., Ponte, A. L., Elipot, S., Menemenlis, D., Zaron, E. D., and Abernathey D, 2019: Surface kinetic energy distributions in the global oceans from high-resolution numerical models and surface drifter observations, Geophys. Res. Letters, 46(16), 9757-9766, https://doi.org/10.1029/2019GL083074

---

## Referee Comment (RC2) · Anonymous Referee #2 · 4 Jan 2021

General comments

The Arabian Sea (AS) Oxygen Minimum Zone (OMZ) has profound consequences on the ecosystem and climate, making it important to understand the evolution of oxygen in the AS. The present study offers to investigate the mechanisms driving the oxygen evolution in the AS using a set of sensitivity experiments performed with an eddy-resolving model. They conclude that the deoxygenation in the northern AS is primarily caused by the reduced ventilation associated with the recent fast warming, particularly in the Gulf, while the summer monsoon winds intensification caused oxygenation in the rest of the AS.

While agreeing that the topic fits well within the scope of the journal, there are several lacunae both in terms of analyses and presentation that do not allow me to recommend the manuscript for acceptance in its present form. Below are my detailed review comments on the manuscript.

Specific comments

1. The title is somewhat overstated/misleading: "fast" warming is still debated in the observations (see for e.g. Gopika et al., 2020); the "deoxygenation" occurs mainly in the northern AS (as their results suggest), unlike what is stated in the title. 2. What is the focus region – AS/northern AS/AS OMZ? The entire paper, including the abstract, switches its discussion between those regions, making it difficult for the reader to comprehend. 3. The importance of Gulf warming for the AS OMZ has already been addressed by the authors in Lachkar et al., GRL, 2019, which is partly the focus of the current manuscript. The authors need to clarify or discuss this in detail. 4. Schmidtko et al. (2017) did not specifically discuss a decline in oxygen in AS nor on the west coast of India. However, this paper is explicitly referenced in the introduction for claiming deoxygenation in the AS. Similarly, the study by do Rośorio Gomes et al. (2014) is largely debated. On the other hand, many relevant references, for e.g. Sandeep and Ajayamohan, 2015, are not cited. 5. The manuscript does not provide important details. For e.g., how the trends are computed? What is O2sat./AOU and how are they estimated? No details provided on O2 budget, ventilation and biological consumption terms, in the methods section. 6. The manuscript is badly written, with careless handling in many places. Almost every paragraph contains sentences which are not easy to follow. I will mention some of them below. There are many typos too. For e.g. E or W in longitudes (P4 L25 and other places). Figures in supplementary do not have longitude labels, figures with vectors have no reference scale vector, and captions do not provide complete details (i.e. the region for average, Fig 6: what is the reference for % computation, etc.). I strongly suggest the authors to improve the presentation, including the abstract. 7. Avoid methodological details in the results. For e.g. P9 L20:

[Figure]

the first two sentences of section 3.1 belongs to methods section. P14 L20: details of experiments should be moved to methods. 8. P4 L22: "major rivers in the northern Indian Ocean". Provide details as to which rivers are considered, particularly in the Arabian Sea. 9. The authors provide validation of their model at the surface level and at the seasonal scale. The model shows a good performance as the fields are restored at the surface. However, as the paper focusses on the subsurface level, it is necessary to validate the model at the subsurface, for e.g. with Argo observations, both in terms of vertical profiles/sections. It is also important to demonstrate that there is no drift in the model at the deeper levels (e.g. ventilation). 10. P9 L28: How do you ascertain that 14% increase in denitrification is due to 10% decline in oxygen? Are these anomalies correlated? 11. The manuscript discusses many processes and the link/flow is somehow missing. I suggest to first provide a discussion of possible causes of oxygen variations and then address each of them. Section 3.2 P10. It may not be clear to the reader the link between AOU and ventilation. The authors should provide substantial details of these parameters in the methods. 12. P10 L5: O2 decline vs O2 saturation. How do you ascertain their link? What about AOU at this depth? 13. P11 Fig. 6: Panel (c) shows O2 section at 65E. Different color scales make it difficult to compare. There seems to be some inconsistency in the deeper levels (below 200 m) when compared to panel (b), especially between 18 and 20N. Moreover, there appears to some discontinuity in the vertical layers (jumps in color shades) in panel c. Does it arise from some mistake from sigma to depth level conversion? Lastly, provide the details of how the % change is calculated in the caption. 14. P12 Fig. 7: What is the region (lat, lon, depth) over which the volume calculation is performed? The O2 trends are different in northern and southern AS, if you considered the entire AS. This figure suggests decadal signals that may alias the linear trend computation. Are these trend estimates hold good even after removing decadal signals? 15. P13 Fig. 8: How is O2sat computed? No mention of region in the caption. As you discuss AOU in the text, why is that AOU not shown on these panels. 16. P14 L12. Inconsistency between figures 9 and S9. Fig S9 presents the vertical and horizontal components of ventilation

term shown in blue in Fig 9. It is hard to see that they add up to the total ventilation, including their scales. 17. P14 Section 3.3: The details of sensitivity experiments may be shifted to a new section under methods. 18. How sensitive are the results to a different forcing set other than the ERA-Interim? How well the oxygen trends are constrained when using different forcing field? Are the decadal signals (say in winds) consistent among different observations? 19. P14: The sensitivity experiments are conducted by repeating the cycles of 1986 to remove interannual variations and is considered to vary climatologically. I am wondering why 1986 repeated cycles are used for forcing or why this particular choice is made. A cleaner approach would be to compute the climatology (of heat fluxes in AS/Gulf or winds) and use it to force the model. 20. I have a major concern regarding the sensitivity experiments. First of all, the authors do not provide much details on the region over which the fields are allowed to vary climatologically (1986 repeated cycles). Moreover, there is an underlying assumption here. How do you ensure that warming results primarily from the heat fluxes. For e.g., by suppressing the heat flux variations in the Gulf, are you able to remove the Gulf warming? Did you check whether the AS heat fluxes affect the Gulf warming? Similarly, does the AS warming results mainly from the heat fluxes over the AS? In the absence of a figure showing warming trends (somewhat like Figure 10, or time series averaged over Gulf/AS for sensitivity experiments), it is not justified to presume that the experiments indeed isolate the respective processes. 21. P15 Figure 9: There appears to be decadal signals, which may potentially alter the trend estimates. No mention of the region in the caption. Panel numbering has gone wrong. The terms have to be described in the methods. 22. P15 L6: "summer upwelling intensification . . ." How do you conclude that there is an upwelling intensification as there is no figure to support this claim? And how upwelling region contributes to the rest of the basin? These claims are mere speculations without any justification. 23. Fig 11 vs S11. The authors show the maps from the sensitivity experiments, but have chosen to move the maps that demonstrate the effect of each process to supplementary figures. For instance, to know the effect of heat fluxes, one has to refer to the supplementary figure (say S11). I

would choose the other way, as it is difficult to follow the discussion in the text. I guess, the text can be considerably improved if the authors stick to discuss only the effect of processes, instead of describing the results of sensitivity experiments. 24. P16 Fig. 10. There is no reference vector. I find some inconsistency between background color shade (speed) and the vectors. Negative shades on the west coast of India and the direction of overlaid winds do not agree. Further, the low magnitude vectors south of equator & to the east of 65E correspond to high wind speeds (red shade). I expect the trends in wind stresses and wind speeds are not entirely different. 25. P17 Figure 11: Panels g & h are not consistent with the rest of the panels, when visually averaged over the northern AS. As emphasized before, I would suggest to interchange Fig. 11 with S11 to ease the discussion in the text. 26. P19 L9-10: "... potentially impacting O2 supply to northern AS". How? 27. P19 L15-16: "the shoaling of thermocline ... top 200 m". Not really justified. How well thermocline anomalies are correlated with O2 anomalies? 28. P19 L29-30: The authors point to an increased summer upwelling on the western AS. However, there is no clear evidence for its contribution to northern AS. On the other hand, the northern AS productivity is mainly due to convective mixing during winter. The influence of winter monsoon has not been discussed in the manuscript. The winter monsoonal winds directly contribute to the ventilation as well as through productivity changes to the oxygen in the northern AS. 29. P19 L33-34: The negative feedback of denitrification is merely speculated and not really demonstrated. Thus, its effect is overstated. 30. Figure S10: It is not convincing that there are no strong SST trends in the upwelling region of the western AS during JJAS. 31. Figure S11: The 4% increase marked in panel a is for entire AS, not for northern AS as mentioned in the text (Section 3.4; P16 L6). Stratification shows a clear decadal signal.

Technical corrections (not all)

1. P19 L2-4: "This suggests ... at play". Not obvious to the reader. The entire first paragraph in this page is not readable. 2. Many panels and many of the supplementary figures are not properly referenced in the manuscript. 3. Figures S7 & S8: Not easy

for me to understand. No color bar. Repeating statements in caption.

References Gopika, S., Izumo, T., Vialard, J. et al. Aliasing of the Indian Ocean externally-forced warming spatial pattern by internal climate variability. Clim Dyn 54, 1093–1111 (2020). https://doi.org/10.1007/s00382-019-05049-9

Sandeep, S., Ajayamohan, R.S. Poleward shift in Indian summer monsoon low level jetstream under global warming. Clim Dyn 45, 337–351 (2015). https://doi.org/10.1007/s00382-014-2261-y

Please also note the supplement to this comment:
https://bg.copernicus.org/preprints/bg-2020-325/bg-2020-325-RC2-supplement.pdf

---

## Referee Comment (RC3) · Anonymous Referee #3 · 5 Jan 2021

I found this paper very interesting and worthy of being published in BG because it describes with great clarity and yet simply how the OMZ of the Arabian Sea is sensitive to upper-ocean warming and monsoonal changes. It should be of great utility especially to non-modelers especially observational researchers who want to understand how their data falls into the big picture.

ABSTRACT:

Although I finally did understand this sentence "This is because surface warming enhances vertical stratification, thus limiting ventilation of the  intermediate ocean, while summer monsoon wind intensification causes the thermocline depth to rise in the northern AS and deepen elsewhere, thus contributing to lowering O2 levels in the upper 200 m in the northern AS and increasing it in the rest of the  AS' it should be improved as its part of the Abstract which is what most researchers will read.

INTRODUCTION: Mechanisms for deoxygenation should be described better.
The authors should clarify where oxygenation vs deoxygenation takes place. The biology of the Arabian Sea is vastly different in the north vs the south of the Arabian Sea.
Also page 3 and the two paras comprising lines 23-30 could be better organized

RESULTS: The authors contend that biological productivity contributes only minimally to deoxygenation, because stratification decreases input of nutrients and so reduces primary productivity. Conversely, enhanced summer winds will increase biological productivity. However, Goes et al. (2020) has shown that in spite of warmer, winter monsoonal winds, decreased convective winds and consequent increased stratification, winter Chlorophyll a has seen an increase (Fig. 1c) because of the rise of *Noctiluca scintillans* blooms which have high biomass on account of the large populations of photosynthetic endosymbionts that they harbor and their tight nutrient recycling mechanisms that allow them to survive in spite of lowered nutrient concentrations. It would be interesting to see how this recent change in biodiversity of winter blooms plays into the role of deoxygenation.

 Goes, J.I., Tian, H., Gomes, H.d.R., Anderson, O.R., Al-Hashmi, K., deRada, S., Luo, H., Al-Kharusi, L., Al-Azri, A., and Martinson, D.G. (2020). Ecosystem state change in the Arabian Sea fuelled by the recent loss of snow over the Himalayan-Tibetan Plateau region. Scientific Reports 10, 7422.

---

## Author Comment (AC1) · 2 Apr 2021

We are grateful for the first referee's efforts in reviewing the manuscript and for the constructive and valuable comments. We believe our paper will significantly improve as a result of his/her comments. We highlight our responses to the general and specific comments and explain the revisions we will make to the paper accordingly. The reviewer's comments are shown below in italics writing while our response is marked in red.

**Anonymous Referee #1 Received and published: 7 December 2020**

Given their ecological and socio-economical impacts, it is a must to better describe and understand the present and future evolution of the major Oxygen Minimum Zones (OMZ). Despite large uncertainties related to their sparse and inhomogeneous coverage, in-situ observations suggest that suboxic conditions in the Arabian Sea (AS) may have expanded during the past decades. The present study uses a high-resolution ocean regional model forced with an atmospheric reanalysis to assess the mechanisms responsible for the oxygen decline over the AS. Their results point towards a major influence of reduced ventilation through enhanced upper ocean stratification on the northern AS desoxygenation, further strengthen by summer monsoon wind changes. While previous studies recognised that these two physical processes (along with biological ones) may contribute to long-term oxygen evolution in the AS, this study offers for the first time a reliable quantification of the respective influence of each of these processes over the historical period. I hence believe that this is a very interesting and important study that deserves to be published in Biogeosciences.

We thank the reviewer for the positive comment and the encouraging assessment of the paper.

However, there are a number of major issues listed below that need to be addressed before publication.

**Major comments:**

A. Validation: Most of the validation section evaluates the model ability to simulate the seasonal cycle in the Arabian Sea. Because this paper mainly focuses on the oxygen long-term evolution, I would recommend the authors to (1) reduce the validation of the seasonal aspects and (2) expand the validation of the long-term trends as follows: (1) Figure 1 and 2: I would move these Figures in the SI. Given that the authors use a surface restoring the observed salinity and temperature, this is not surprizing that the seasonal cycle of the model SSS and SST agree well with observations. In addition, I feel that Figure 2 is not central to the present study and should

not be included in the core of the paper. (2) Figure 3: Keep at least the lower panels (Chl validation) but improve the color scale chosen to emphasize the regional contrasts. (3) Figure 4: Add meridional sections of O2 concentration along with hypoxic/suboxic volume to evaluate the O2 structure at depth in both model and observations.

As we agree with the reviewer assessment, we will follow his/her suggestion and implement those requested changes. More specifically, we will move Figure 1 and Figure 2 to SI and improve the color scale used in Figure 3 (for surface chlorophyll). We will also add meridional and zonal sections of O2 concentration to evaluate the O2 vertical structure in both model and observations (please see our response to comment #17).

**(4) Evaluation of long-term evolution**

(extra analyses): Given that the reliability of the results presented here relies on the strong assumption that the model reproduces accurately the long-term evolution in the AS, the model ability to simulate this evolution should definitely be assessed in greater details. For instance, this paper should include an evaluation of the long-term evolution of AS maps and yearly time series in the northern/western AS for model and observations of (1) Chl over the common period (1997-2010) where observations and model data are available (using OC-CCI product for instance), (2) SST over the entire period (using ERA5 or HadISST for instance), (3) static stability index (as in Roxy et al. 2016) over the entire period (using an ocean reanalysis such as SODA or ORAS4/5) and (4) wind trends from several .wind products. Finally, I would also like to see an attempt to evaluate the model O2 evolution over wide boxes with in-situ observations (from Ito or Schmidtko et al. 2017 for instance). This would allow strengthening the reliability of the model results regarding the oxygen decline in the AS and the related mechanisms discussed in the paper.

Following the reviewer suggestion, we will substantially expand the evaluation of modeled longterm evolution of key properties. In particular we will evaluate long-term evolution of temperature and salinity at multiple depths as well as upper ocean static stability and surface chlorophyll-a concentration. More specifically, we contrast trends in SST from four SST products: AVHRR (used to force the model), ERA5, HadISST and the NOAA OISST products (see Fig 1). This comparison shows that all SST products agree on the important warming the Arabian Sea has undergone over the study period, despite differences in the magnitude of warming, with the ERA5 (resp. HadISST) displaying the strongest (the weakest) rates of warming.

---

## Author Comment (AC2) · 2 Apr 2021

We would like to thank the second reviewer for the comments that we hope will significantly improve our manuscript. We highlight our responses, point by point, to the reviewer's general and specific comments and indicate the revisions we will make to the paper accordingly. The reviewer's comments are shown below in italics writing while our response is marked in red.

**Anonymous Referee #2**

Received and published: 4 January 2021

**General comments** The Arabian Sea (AS) Oxygen Minimum Zone (OMZ) has profound consequences on the ecosystem and climate, making it important to understand the evolution of oxygen in the AS. The present study offers to investigate the mechanisms driving the oxygen evolution in the AS using a set of sensitivity experiments performed with an eddy resolving model. They conclude that the deoxygenation in the northern AS is primarily caused by the reduced ventilation associated with the recent fast warming, particularly in the Gulf, while the summer monsoon winds intensification caused oxygenation in the rest of the AS.

While agreeing that the topic fits well within the scope of the journal, there are several lacunae both in terms of analyses and presentation that do not allow me to recommend the manuscript for acceptance in its present form. Below are my detailed review comments on the manuscript.

**Specific comments**

1. The title is somewhat overstated/misleading: "fast" warming is still debated in the observations (see for e.g. Gopika et al., 2020); the "deoxygenation" occurs mainly in the northern AS (as their results suggest), unlike what is stated in the title.

The focus of the Gopika et al 2020 paper is on the period (1871-2016), which is different from our study period (1982-2010). In particular most of the uncertainties in SST trends referred to in that paper have to do with the quality and the gaps in data prior to the 1960s. Moreover, the uncertainties debated in that paper relate to the large-scale east-west and north-south patterns of the warming Indian Ocean and less about the average Arabian Sea warming itself. Finally, the warming in the Arabian Sea over the study period has been faster than the global average (e.g., Beal et al., 2020).

However, we do agree that most of the deoxygenation we report (and analyze) in this study is located in the northern Arabian Sea. Therefore, we will change the title of the paper to: "Fast local warming of sea-surface is the main factor of recent deoxygenation in the northern Arabian Sea".

2. What is the focus region – AS/northern AS/AS OMZ? The entire paper, including the abstract, switches its discussion between those regions, making it difficult for the reader to comprehend.

The focus region is the Arabian Sea OMZ. As deoxygenation (and its implications for the OMZ) is largest in the northern part of the AS, we devote particular attention to the northern AS. Please note that we will change the title to "Fast local warming of sea-surface is the main factor of recent deoxygenation in the northern Arabian Sea" to make this clearer. We will also make this clearer in the abstract and introduction and in the discussion of our figures in the revised manuscript.

3. The importance of Gulf warming for the AS OMZ has already been addressed by the authors in Lachkar et al., GRL, 2019, which is partly the focus of the current manuscript. The authors need to clarify or discuss this in detail.

It is true that the importance of the Gulf warming for the AS OMZ has been explored in Lachkar et al. 2019 using a set of idealized future warming scenarios of the Gulf. In the present study, however, we explore the drivers of recent changes in O2 and demonstrate that the recent warming in the Gulf contributes to those trends, but is far from entirely explaining them. Indeed, significant deoxygenation is simulated when the AS warms up, even when the Gulf temperature does not change (see Fig11 of the paper and Fig 2 in this document). Therefore, the present study is consistent with the findings of Lachkar et al (2019) but goes beyond in terms of the processes taken into consideration (warming and wind changes over the entire Arabian Sea) and the nature of the perturbation (real observed changes vs. idealistic future scenarios).

4. Schmidtko et al. (2017) did not specifically discuss a decline in oxygen in AS nor on the west coast of India. However, this paper is explicitly referenced in the introduction for claiming deoxygenation in the AS. Similarly, the study by do Rosorio Gomes et al. (2014) ´ is largely debated. On the other hand, many relevant references, for e.g. Sandeep and Ajayamohan, 2015, are not cited.

Although the focus of the Schmidtko et al. (2017) is global, local O2 trends are shown regionally including in the Indian Ocean and the Arabian Sea. Indeed, in their extended Data Figure 1, panel a, O2 trends (in units of percentage of local dissolved oxygen) are shown in the AS. This figure indicates a drop of O2 in the western and northern AS as well as along the west coast of India between 1960 and 2010 that we refer to in our introduction.

This study is consistent with many other observational studies that have reported or suggested deoxygenation in the Arabian Sea and that we cite. Namely: Ito et al. (2017), Laffoley and Baxter, 2019, Piontkovski and Al-Oufi (2015) Banse et al. (2014) Queste et al. (2018) Al-Ansari et al., 2015; Al-Yamani and Nagvi, 2019 with a focus on the Arabian/Persian Gulf. We also cite do Rosário Gomes et al. (2014) because it points to an important potential link between recent oxygen trends in the region and observed changes in the plankton community composition.

As for the study by Sandeep and Ajayamohan (2015) referred to by the reviewer, it is about potential changes in Indian monsoon atmospheric circulation and is not directly linked to O2 or deoxygenation in the Arabian Sea. However, we will cite this paper in the revised manuscript when we discuss potential changes in regional monsoon winds.

5. The manuscript does not provide important details. For e.g., how the trends are computed? What is O2sat./AOU and how are they estimated? No details provided on O2 budget, ventilation and biological consumption terms, in the methods section.

**O2 trends:**

In the original manuscript we explain the calculation method:

"To investigate long-term changes in O2 time series, data was deseasonalized by removing monthly climatologies from the original time series.

Furthermore, to identify significant trends in O2 we used the non-parametric Mann-Kendall (MK) test (Mann, 1945; Kendall, 1948) that does not assume normality of data distribution and hence is less sensitive to outliers and skewed distributions."

For more clarity, we will add:

"Linear trends were computed from the slope of the least squares regression line."

**O2sat/AOU:**

We agree with the reviewer that while oxygen saturation (O2 sat) and the apparent oxygen utilization (AOU) may be familiar concepts for marine biogeochemists and the oxygen community, they may indeed need to be explicitly defined for a larger audience not necessarily familiar with these notions. Therefore for more clarity we will add an explicit definition of AOU and O2.

**We will include:**

"O2sat is the oxygen saturation concentration (in mmol/m3) computed at 1atm pressure from temperature and salinity following Garcia and Gordon (1992). It corresponds to the maximum O2 concentration in seawater at equilibrium, for a given temperature and salinity."

"AOU (apparent oxygen utilization) is the difference between oxygen saturation O2sat and the actual O2 concentration and can be estimated as:

AOU=O2sat-O2

It is a measure of O2 utilization through biological activity since the water parcel was last at equilibrium in contact with the atmosphere. Therefore it is sensitive to both biological productivity and circulation (ventilation)"

**O2 budget:**

The O2 sources and sinks equations were included in Lachkar et al (2016) that we refer to in the present manuscript. However, for more clarity we will include them in the revised manuscript.

6. The manuscript is badly written, with careless handling in many places. Almost every paragraph contains sentences which are not easy to follow. I will mention some of them below. There are many typos too. For e.g. E or W in longitudes (P4 L25 and other places). Figures in supplementary do not have longitude labels, figures with vectors have no reference scale vector, and captions do not provide complete details (i.e. the region for average, Fig 6: what is the reference for % computation, etc.). I strongly suggest the authors to improve the presentation, including the abstract.

We thank the reviewer for pointing out those typos that will be corrected.

P4 L25: typo will be corrected (78W changed to 78E).

Missing longitude labels in some supplementary figures will be added.

A reference scale vector will be added to Fig 2, Fig 10 and Fig S10.

The area in the northern Arabian Sea (north of 20N) that we average over will be shown explicitly.

The inventories which are vertically integrated O2 concentrations are shown at each grid point. As for change expressed in % units, it is the amount of change as percentage of local dissolved oxygen inventory as routinely presented in many deoxygenation studies (e.g., Schmidtko et al, 2017). We will make this clearer in the revised manuscript.

7. Avoid methodological details in the results. For e.g. P9 L20: the first two sentences of section 3.1 belongs to methods section. P14 L20: details of experiments should be moved to methods.

This will be fixed. We will move these two statements to the methods section.

8. P4 L22: "major rivers in the northern Indian Ocean". Provide details as to which rivers are considered, particularly in the Arabian Sea.

**Will be done.**

9. The authors provide validation of their model at the surface level and at the seasonal scale. The model shows a good performance as the fields are restored at the surface. However, as the paper focusses on the subsurface level, it is necessary to validate the model at the subsurface, for e.g. with Argo observations, both in terms of vertical profiles/sections. It is also important to demonstrate that there is no drift in the model at the deeper levels (e.g. ventilation).

**Validation of the model at the subsurface:**

Beyond the validation of surface properties (including variables that are not restored such as currents in Fig2, and NO3 and Chla in Fig3) we have validated the model at the subsurface:

- 1) O2 and NO3 distributions between 250 and 700m from the model are compared to observations (Fig 4 and Fig S6).
- Additionally, we have shown a quantitative comparison of O2 and NO3 profiles in the top 400m between the model and the observations as a part of the Taylor diagrams presented in Fig 5.

Additionally, in the revised manuscript we will add a thorough validation of long-term changes, including changes in temperature and salinity in the subsurface as well as stratification in the upper ocean. Please see our detailed response to General Comment A by referee #1.

**Model drift:**

This has already been addressed in the original manuscript. Please see the analysis of model drift in page 1 of the Supplementary. Please also see salinity and oxygen long-term trends shown in Figs S1, S2, S3, S4 and S5. Please note that the model drift is analyzed in two vertical layers: 0-200m and the 200-1000m as clearly stated in these figures captions.

**10. P9 L28: How do you ascertain that 14% increase in denitrification is due to 10% decline in oxygen? Are these anomalies correlated?**

We would like to stress that we are not referring to "10% decline in O2" in this statement as stated by the reviewer but instead a 10 % increase in the volume of suboxia (O2

Figure 1: Interannual anomalies in the volume of (top left) suboxia (top right) hypoxia and (bottom) water column denitrification over the study period. The dashed lines indicate the trend lines.

Finally, we do acknowledge that decadal variability is potentially playing an important role in the identified linear trends. However, due to the relatively short simulation period (29 years), it is unfortunately not possible to rigorously filter out that (decadal variability) signal in a statistical sense. We will acknowledge this in the list of the study caveats.

15. P13 Fig. 8: How is O2sat computed? No mention of region in the caption. As you discuss AOU in the text, why is that AOU not shown on these panels.

AOU is the difference between O2 and O2sat. We didn't include AOU in the plots because O2 and O2sat are already shown. Please see our responses to previous comments #5 and #12 for more details regarding the computation of O2sat and AOU.

16. P14 L12. Inconsistency between figures 9 and S9. Fig S9 presents the vertical and horizontal components of ventilation term shown in blue in Fig 9. It is hard to see that they add up to the total ventilation, including their scales.

The difference between Fig 9 and Fig S9 is that the ventilation in Fig S9 was not shown on the same scale (the changes in total ventilation are much smaller than the changes in the individual horizontal and vertical components taken separately).

In the revised manuscript we will split total ventilation into advection and vertical mixing components. Please see Fig 16 in our response to comment #24 by referee #1.

17. P14 Section 3.3: The details of sensitivity experiments may be shifted to a new section under methods.

**This will be done.**

18. How sensitive are the results to a different forcing set other than the ERA-Interim? How well the oxygen trends are constrained when using different forcing field? Are the decadal signals (say in winds) consistent among different observations?

**We discuss this in our response to Reviewer 1 comment #3.**

19. P14: The sensitivity experiments are conducted by repeating the cycles of 1986 to remove interannual variations and is considered to vary climatologically. I am wondering why 1986 repeated cycles are used for forcing or why this particular choice is made. A cleaner approach would be to compute the climatology (of heat fluxes in AS/Gulf or winds) and use it to force the model.

The reason we use the repeated normal year approach instead of a monthly climatology (as done routinely to filter out the interannual variability while keeping high-frequency forcing, e.g., Large and Yeager, 2004, Stewart et al., 2020) is that mixing and Ekman circulation are highly sensitive to the high-frequency wind variability (associated with weather systems) as wind stress is estimated using bulk formulas based on wind speed data. As the wind stress is a quadratic function of wind speed, using smoothed wind speed data would result in artificially suppressed Ekman velocities. This approach is commonly used in models using bulk formula forcing. Finally, the choice of 1986 is justified by the fact that this year is neutral with respect to IOD and El Nino/La Nina oscillation. This will be further highlighted in the revised manuscript.

20. I have a major concern regarding the sensitivity experiments. First of all, the authors do not provide much details on the region over which the fields are allowed to vary climatologically (1986 repeated cycles). Moreover, there is an underlying assumption here. How do you ensure that warming results primarily from the heat fluxes. For e.g., by suppressing the heat flux variations in the Gulf, are you able to remove the Gulf warming? Did you check whether the AS heat fluxes affect the Gulf warming? Similarly, does the AS warming results mainly from the heat fluxes over the AS? In the absence of a figure showing warming trends (somewhat like Figure 10, or time series averaged over Gulf/AS for sensitivity experiments), it is not justified to presume that the experiments indeed isolate the respective processes.

We will show SST trends in all sensitivity experiments in the supplementary information (please see Fig 2 below). This analysis confirms that warming signal is entirely driven by the interannual variability in the forcing.

---

## Author Comment (AC3) · 2 Apr 2021

We are grateful for reviewer #3 efforts in reviewing the manuscript and for the constructive and valuable comments. We highlight our responses to the comments and explain the revisions we will make to the paper accordingly. The reviewer's comments are shown below in italics writing while our response is marked in red.

**Anonymous Referee #3**

*I found this paper very interesting and worthy of being published in BG because it describes with great clarity and yet simply how the OMZ of the Arabian Sea is sensitive to upper-ocean warming and monsoonal changes. It should be of great utility especially to non-modelers especially observational researchers who want to understand how their data falls into the big picture.*

We thank the reviewer for the positive comment and the encouraging assessment of the paper.

*ABSTRACT: Although I finally did understand this sentence "This is because surface warming enhances vertical stratification, thus limiting ventilation of the intermediate ocean, while summer monsoon wind intensification causes the thermocline depth to rise in the northern AS and deepen elsewhere, thus contributing to lowering O2 levels in the upper 200 m in the northern AS and increasing it in the rest of the AS' it should be improved as its part of the Abstract which is what most researchers will read.*

This statement will be rewritten for more clarity.

*INTRODUCTION:*
*Mechanisms for deoxygenation should be described better. The authors should clarify where oxygenation vs deoxygenation takes place. The biology of the Arabian Sea is vastly different in the north vs the south of the Arabian Sea. Also page 3 and the two paras comprising lines 23-30 could be better organized*

These two paragraphs will be rewritten for more clarity. We will also expand the discussion of regional deoxygenation in the Arabian Sea and its controlling mechanisms.

*RESULTS: The authors contend that biological productivity contributes only minimally to deoxygenation, because stratification decreases input of nutrients and so reduces primary productivity. Conversely, enhanced summer winds will increase biological productivity. However, Goes et al. (2020) has shown that in spite of warmer, winter monsoonal winds, decreased convective winds and consequent increased stratification, winter Chlorophyll a has seen an increase (Fig. 1c) because of the rise of Noctiluca scintillans blooms which have high biomass on account of the large populations of photosynthetic endosymbionts that they harbor and their tight nutrient recycling mechanisms that allow them to survive in spite of lowered nutrient concentrations. It would be interesting to see how this recent change in biodiversity of winter blooms plays into the role of deoxygenation.*

*Goes, J.I., Tian, H., Gomes, H.d.R., Anderson, O.R., Al-Hashmi, K., deRada, S., Luo, H., Al-Kharusi, L., AlAzri, A., and Martinson, D.G. (2020). Ecosystem state change in the Arabian Sea fuelled*

We thank the reviewer for pointing us to this important recent study also showing evidence of enhanced stratification with potential consequences for the Arabian Sea ecosystem. We will refer to this work in the revised manuscript.

---

## Author Comment (AC4) · 2 Apr 2021

We would like to deeply thank our colleague Yogesh K. Tiwari for his interest in our work and for his passionate comments and feedback. We highlight our responses to the comments below. Our colleague's comments are shown in italics writing while our response is marked in red.

*Authors have used a suit of eddy resolving ocean model simulations to study the decline of O2 in the northern Arabian Sea during recent decades. Authors find that the reduced ventilation caused by the fast warming of ocean surface as the major reason for the decline of O2, with contributions from changes in the summer monsoon winds. Though the topic is of high relevance and the model based study approach is sound, the analysis and presentation are not convincing.*

*C1 1. The central aspect of this study is reduction in O2 due to reduction in upper ocean ventilation. However, this manuscript do not even offer a basic explanation of the ventilation process and a definition of the ventilation term in the budget calculations. A 2-3 sentence summary of Fig.1 in Oschiles et al. (2018, paper cited in the manuscript) in the introduction will be ideal. Please explain how is ventilation defined in the O2 budget. - Why words like "little", "slight" and "dominates" are used instead of actual values of budget terms while discussing about the O2 budget?*

The O2 sources and sinks equations (including the ventilation terms) were described in detail in Lachkar et al (2016) that we refer to here. However, for more clarity we will include them in the revised manuscript.

*2. The connection between changes in the upper ocean mixing process and thermocline depth to the change in ventilation is not presented convincingly. How is vertical stratification defined? - A basic discussion of how the increased stratification leads to reduced ventilation is required here. This is a serious omission considering this is the major mechanism used to explain decline in O2.*

Vertical stratification at a given depth is given by the vertical density gradient at that depth. This will be defined explicitly in the revised manuscript. We will add a statement to explain that enhanced stratification inhibits vertical mixing and hence reduce ventilation.

*- How representative is 20 degree C isotherm as thermocline depth in this region (Fig.12)? Does a temperature gradient/slope based criteria (Fiedler, 2010) offer a better estimate of thermocline depth?*

The thermocline depth is typically represented by the depth of isotherm 20C in the region (e.g. Schott et al., 2009).

*- What leads to the shoaling of thermocline in the northern AS with summer monsoon wind intensification? - Is there a compensating effect between the surface warming induced vertical stratification and increased vertical mixing from increase in the wind speed (see reference 'c' in item 4 below)? - The impact of increase in stratification on the reduction of ventilation and the resulting reduction in the O2 concentration is not shown quantitatively.*

Indeed, there seems to be some partial compensation between the two processes. In the revised manuscript we quantify the impact of changes in vertical mixing on the O2 budget. See our responses to comments by Reviewer #1.

*3. Ambiguous or missing description of parameters/features. - What is the vertical resolution of the ocean model for "typical" (say 3000 m depth) water depth in the northern AS? Does it resolve the upper ocean and thermocline depths well?*

As stated in the manuscript the model has 32 terrain-following layers with refined resolution near the surface ocean of a few meters on average in top 100m. Therefore, it resolves well the upper ocean physics.

*- 58 year Spinup: Which climatological forcing? Was SST and SSS restored? If so to which dataset? - What version of SODA data has been used and what is its resolution? - What frequency was the analyzed model data? Monthly mean? - What is the point in comparing SST and SSS from the model to the same dataset it has been restored to? Is observations used in WOA2013 (salinity) is also used in SODA reanalysis? –*

The model spinup is actually made of 58 years forced with a repeated annual cycle (forced with SODA climatology) in addition to four 29-year (1982-2010) repeated cycles. So, in total the spinup phase is 145 year long. We will add more details about the spinup and the SODA version in the revised manuscript.

*C2 Fig.2: Drifter measurements are for 15 m depth (examples: Section 1, last paragraph of Lumpkin et al. (2013), Section 2.2 last paragraph in Yu et al. (2019)). What depth are the model fields plotted here? When doing vector plots, please mention at what interval of model grid points (in X and Y directions) the arrows are shown. West India Coastal Current and circulation associated with the Lakshadweep High are very weak in the model compared to that in the drifter data and no explanation has been provided for this. - Fig.3: Large difference (model vs observation) in NO3 at the northern coastal-AS in Winter and at the south-west coast of Indian-Peninsula in Summer are not explained properly.*

Velocities are compared between the model and the drifter data at the same level. The goal of this evaluation is to demonstrate the model skill to represent the large-scale patterns of circulation and biology and not to focus on very local details unrelated to the main goal of the study. Finally, model evaluation in the revised manuscript will be expanded to include a more thorough validation of the long-term trends.

*- O2 Budget: Please explain the terms included in the budget (grouped as "biology" and "transport") in detail. Is the budget calculated online or offline? How well does the budget balance? - Layers of upper 200 m and below 200 m (Fig.6,7,8 and throughout the manuscript). How can authors justify analyzing data for a selected depth (like 100 m) to describe the properties in a "layer" (like 0-200m, Fig.6a)?. Isn't is appropriate to use a layer averaged (like in Fig.4) or integrated fields (like in Fig.9) instead? Please clearly mention how the data is processed in each of the figures mentioned above and justify the choice.*

The O2 sources and sinks equations (including the ventilation terms) were described in detail in Lachkar et al (2016) that we refer to here. However, for more clarity we will include them in the revised manuscript. The O2 budgets are shown for two layers (100-200m and 200-1000m) and not at a particular depth as the comment suggests.

*What makes the 200 m depth as a layer separation depth? - Fig.6 panels a and b shows trends in percentage/decade but panel c shows actual values in nmol/m3/decade. It is very difficult to understand the context without visualizing the spatial pattern of actual trend in nmol/m3/decade. - Fig.7: How is the anomaly defined? Difference between monthly-varying model data, and mean of the time series?*

The 200 m depth is usually used to separate the epipelagic zone (0-200m) and the

mesopelagic zone (200-1000m) and also corresponds roughly to the depth of thermocline in the region. Finally, interannual anomalies are computed by deseasonalizing the data.

*According to Fig.4 and definition, hypoxic ($O2 < 60$ nmol/m3) region is bigger than suboxic ($O2 < 4$ nmol/m3) region in an average (250- 700 m) sense. So, naturally one expect the anomaly in hypoxic should be higher than that in the suboxic case but it is not true according to Fig.7 panels a and b.*

Just because the volume of hypoxia is larger than that of suboxia does not imply it should increase more. Actually, it is the opposite that happens (in % terms) as the suboxic waters are mostly concentrated in the northern Arabian Sea where deoxygenation is strongest, whereas hypoxic waters occupy a larger area that experience both oxygenation and deoxygenation trends.

*Authors, explain that the small patches of oxygenation in the eastern/central/southern region can make the volume of hypoxic region nearly constant compared to that of suboxic region to the north. But this is very difficult to comprehend from the percentage based trend shown in the panels 6a and 6b. Using percentage trend at two representative depth to explain the anomaly (in actual value/units) over a volume of water is very difficult for a reader to follow and interpret. Why not use the vertically integrated O2 for 0-200 m range and 200-700 m range in Fig.6.*

We will add the absolute changes in O2 inventories in the revised manuscript (in the SI).

*- Sensitivity Experiments: Did any smoothing has been applied while modifying the forcing for a particular region (eg. Gulf) or time (eg. wclim_JJAS). If not, how strong was the impact of sudden jumps in the forcing field resulted from these modifications on the model solution? Labelling of experiments with 1986 fields as "clim" is misleading, instead use either "1986" or "normal" to indicate it is from a specific year. Also explicitly mention what is meant by the "control" run. - SST Warming: What is the mechanism behind widespread SST warming in the AS? Just citing few past studies is not sufficient since this is one of the core process which contributes to the O2 reduction. Please summarize major reasons for this warming where it is first discussed.*

Temporal interpolation is applied to reduce the discontinuity in time in the wclim_JJAS and the new wclim_DJFM. As for the no-Gulf-warming simulation, only forcing SST is modified in the Gulf.
Regarding the repeated normal year approach, we refer to it as climatological because 1) the forcing is repeated every year, so it has no interannual variability and 2) because it is based on a normal year (neutral with respect to major climate variability modes).
As for what is causing the warming in the Arabian Sea, this is far beyond the scope of our study. We prefer to keep the paper focused on our key question: deoxygenation, instead of going in all directions.

*4. Missing citations of relevant literature. - Some of the recent studies which are highly relevant for the topics discussed/addressed in this manuscript (a: changes in monsoon winds, b: oceanic impacts of changes in monsoon winds, c:ocean mixing energetics of changes in monsoon winds, d&e: Oxygen minimum zone in the northern AS ) are not cited.*

We thank our colleague for compiling this selection of papers. We will cite relevant ones.

*a) Sandeep, S., and Ajayamohan, R.S., 2015: Poleward shift in Indian summer monsoon low level jetstream under global warming, Clim. Dyn. 45, 337-351; doi:10.1007/s00382-014-2261-y b) Praveen, V., Ajayamohan, R.S., and Valsala, V., 2016: Intensification of upwelling along Oman coast in a warming scenario, Geophys. Res. Lett., doi:10.1002/2016GL069638 c) Praveen, V., Valsala, V., Ajayamohan, R.S., and Balasubramanian, S., 2020: Oceanic mixing over northern Arabian Sea in a warming scenario: Tug of war between wind and buoyancy forces, J. Phys. Oceanogr., 50(4), doi:10.1175/JPO-D-19-0173.1 d) Shenoy, D. M. et al., 2020: Variability of dissolved oxygen in the Arabian Sea Oxygen Minimum Zone and its driving mechanisms, J. Marine Sys., 204, https://doi.org/10.1016/j.jmarsys.2020.103310 C4 e) Sarma et al 2020: Potential mechanisms responsible for occurrence of core oxygen minimum zone in the north-eastern Arabian Sea, Deep Sea Res. PartI, 165, https://doi.org/10.1016/j.dsr.2020.103393 - Add a citation or a statement in the acknowledgements for the SODA data (as shown in the link given below) https://climatedataguide.ucar.edu/climate-data/soda-simpleocean-data-assimilation Carton, J.A. and B. Giese, 2008: A Reanalysis of Ocean Climate Using Simple Ocean Data Assimilation (SODA). Mon. Weath. Rev., 136, 2999-3017. As detailed above, this manuscript needs to be rewritten focusing on the details of analysis and presentation in order to make it a publishable one. References: ————- Fiedler, P., 2010: Comparison of objective descriptions of the thermocline, Limnology and Oceanogr., https://doi.org/10.4319/lom.2010.8.313 Lumpkin, R., Grodsky, S. A., Centurioni, L., Rio M-H, Carton, J. A., and Lee D., 2013: Removing spurious low-frequency variability in drifter velocities. J. Atmos. Oceanic Technol., 30(2), 353-360, https://doi.org/10.1175/JTECH-D-12-00139.1 Yu, X., Ponte, A. L., Elipot, S., Menemenlis, D., Zaron, E. D., and Abernathey D, 2019: Surface kinetic energy distributions in the global oceans from high-resolution numerical models and surface drifter observations, Geophys. Res. Letters, 46(16), 9757-9766, https://doi.org/10.1029/2019GL083074*

---

## Author Comment (AC5) · 2 Apr 2021

We thank our colleague V. Valsala for his interest in our work. We will cite the missing relevant papers.

---

## Author Response (AR1)

**Review #1**

We are grateful for the first referee's efforts in reviewing the manuscript and for the constructive and valuable comments. We believe our paper has significantly improved as a result of his/her comments. We highlight our responses to the general and specific comments and explain the revisions we have made to the paper accordingly. The reviewer's comments are shown below in italics writing while our response is marked in red.

**Anonymous Referee #1**

*Given their ecological and socio-economical impacts, it is a must to better describe and understand the present and future evolution of the major Oxygen Minimum Zones (OMZ). Despite large uncertainties related to their sparse and inhomogeneous coverage, in-situ observations suggest that suboxic conditions in the Arabian Sea (AS) may have expanded during the past decades. The present study uses a high-resolution ocean regional model forced with an atmospheric reanalysis to assess the mechanisms responsible for the oxygen decline over the AS. Their results point towards a major influence of reduced ventilation through enhanced upper ocean stratification on the northern AS desoxygenation, further strengthen by summer monsoon wind changes. While previous studies recognised that these two physical processes (along with biological ones) may contribute to long-term oxygen evolution in the AS, this study offers for the first time a reliable quantification of the respective influence of each of these processes over the historical period. I hence believe that this is a very interesting and important study that deserves to be published in Biogeosciences.*

We thank the reviewer for the positive comment and the encouraging assessment of the paper.

*However, there are a number of major issues listed below that need to be addressed before publication.*

***Major comments:***

*A. Validation: Most of the validation section evaluates the model ability to simulate the seasonal cycle in the Arabian Sea. Because this paper mainly focuses on the oxygen long-term evolution, I would recommend the authors to (1) reduce the validation of the seasonal aspects and (2) expand the validation of the long-term trends as follows: (1) Figure 1 and 2: I would move these Figures in the SI. Given that the authors use a surface restoring the observed salinity and temperature, this is not surprising that the seasonal cycle of the model SSS and SST agree well with observations. In addition, I feel that Figure 2 is not central to the present study and should not be included in the core of the paper. (2) Figure 3: Keep at least the lower panels (Chl validation) but improve the color scale chosen to emphasize the regional contrasts. (3) Figure 4:*

*Add meridional sections of O2 concentration along with hypoxic/suboxic volume to evaluate the O2 structure at depth in both model and observations.*

As we agree with the reviewer assessment, we have followed his/her suggestion and implemented those requested changes. More specifically, we moved Figure 1 and Figure 2 to SI (Fig S6 and Fig S7) and improved the color scale used in Figure 3 for surface chlorophyll (now shown as Fig S8 in the SI). We also added meridional and zonal sections of O2 concentration to evaluate the O2 vertical structure in both model and observations (please see Fig 1 of the revised manuscript).

*(4) Evaluation of long-term evolution*
*(extra analyses): Given that the reliability of the results presented here relies on the strong assumption that the model reproduces accurately the long-term evolution in the AS, the model ability to simulate this evolution should definitely be assessed in greater details. For instance, this paper should include an evaluation of the long-term evolution of AS maps and yearly time series in the northern/western AS for model and observations of (1) Chl over the common period (1997-2010) where observations and model data are available (using OC-CCI product for instance), (2) SST over the entire period (using ERA5 or HadISST for instance), (3) static stability index (as in Roxy et al. 2016) over the entire period (using an ocean reanalysis such as SODA or ORAS4/5) and (4) wind trends from several .wind products. Finally, I would also like to see an attempt to evaluate the model O2 evolution over wide boxes with in-situ observations (from Ito or Schmidtko et al. 2017 for instance). This would allow strengthening the reliability of the model results regarding the oxygen decline in the AS and the related mechanisms discussed in the paper.*

Following the reviewer suggestion, we have substantially expanded the evaluation of modeled long-term evolution of key properties. In particular we have evaluated long-term evolution of temperature and salinity at multiple depths as well as upper ocean static stability and surface chlorophyll-a concentration. More specifically, we contrasted trends in SST from four SST products: AVHRR (used to force the model), ERA5, HadISST and the NOAA OISST products (see Fig 2a-b and Fig S11 in the SI). This comparison shows that all SST products agree on the important warming the Arabian Sea has undergone over the study period, despite differences in the magnitude of warming, with the ERA5 (resp. HadISST) displaying the strongest (the weakest) rates of warming.

Furthermore, we also contrast simulated temperature and salinity interannual anomalies from the northern Arabian Sea to those from WOD2018 observations at different depths (Fig S12 and Fig S13, SI). This comparison reveals a very good agreement between the simulated and observed temperature trends in the upper ocean of the northern Arabian Sea. For salinity, the observational coverage is much sparser with most of the observations coming from the last decade of the simulation (see Fig S14 in the SI showing temporal data coverage in the northern Arabian Sea). This does not allow us to use observations to validate long-term changes in salinity in the model.

Additionally, we also evaluate the evolution of vertical stratification and static stability in the Arabian Sea in the model. As salinity observations are very sparse over the region during the study period we contrast simulated static stability to that from reanalysis products such as SODA and ORAS5 (Fig 3 in the revised manuscript and Fig S15 in the SI). This comparison reveals that overall the simulated increase in vertical stratification in the Arabian Sea is comparable to similar trends in the ORAS5 and SODA reanalyses, although with local differences in their magnitude and regional patterns.

For biological variables, we only evaluate long-term changes in surface chlorophyll as O2 (and NO3) observations are extremely limited in the area over the study period (please see Fig S14, SI). We would like to point out that the O2 data analyzed in the Schmidtko et al (2017) study mentioned by the reviewer covers a different and longer period of time (1960-2010).

Finally, as surface chlorophyll satellite data is available only from september 1997, we contrast simulated chlorophyll to observations over the common period from september 1997 to the end of the year 2010. The 13-year period is too short to extract meaningful long-term trends. Yet, this comparison is still useful as it reveals a decent agreement between the model and observations over the study period with a moderate correlation of 0.48 between the modeled and observed interannual anomalies in the northern Arabian Sea (Fig S16, SI).

This evaluation of long term changes is presented in the revised manuscript (page 6, line 11 to page 8, line 25) whereas the detailed evaluation of the mean state and seasonal variability has been moved to the SI (Figs S6-S10).

*B. Choice of the sensitivity experiments: It is not obvious to be why the authors decided only to investigate the role of the summer monsoon wind changes on the O2 concentration. Looking at Figure S10, it appears to me that winter monsoon wind changes are also significant, with a strong wind decrease in the northern AS. This decrease is expected to reduce convective mixing and hence ventilation. I would recommend the authors to perform an additional experiment to evaluate this impact and report it in the manuscript.*

Done.
Initially, we decided to focus on summer wind changes because we believe that the Arabian Sea OMZ is much more sensitive to changes in summer winds than winter winds as was shown in Lachkar et al (2018). Indeed, in this previous work we found that the volume of both hypoxic and suboxic waters in the Arabian Sea is much more responsive to perturbations in summer winds than to comparable perturbations in winter winds.

[Figure]

Figure 1: (from Lachkar et al., 2018) Biogeochemical response to monsoon wind intensity changes. Relative changes in response to monsoon wind intensity perturbations in net primary production (green), denitrification (blue), suboxic volume (red) and hypoxic volume (orange). Open circles (respectively, squares) indicate the results from the simulation where summer monsoon wind stress is increased (respectively, decreased) by 50 % and winter monsoon wind stress is decreased (respectively, increased) by 50 %.

This is likely due to the fact that convective mixing is more sensitive to winter cooling than monsoon wind intensity alone. The dominance of surface buoyancy fluxes over wind mixing in driving convective deepening of the mixed layer during winter monsoon season has been established in a couple of previous observational and modeling studies (e.g., Weller et al., 2002, Prasad 2004).

However, following the reviewer's suggestion we have run an additional simulation where winter winds (DJFM) were set to be climatological. The results from this additional sensitivity experiment (shown in Figs 8, 9, 10, and 12 of the revised manuscript) confirm that changes in winter winds do not contribute to the simulated O2 decline in the northern Arabian Sea. Indeed, in the absence of winter wind changes, deoxygenation in the northern AS is maintained and is only slightly weaker in the western AS. This indicates that changes in winter winds do not contribute to deoxygenation in the northern AS and may only very partially explain deoxygenation in the western AS. Additionally, changes in winter monsoon winds are shown to have a limited effect on stratification (Fig 9 of the revised manuscript), suggesting that the weakening of winter convective mixing in the northern Arabian Sea is more associated with winter warming (see SST changes in winter, Fig S17 in SI) than changes in the intensity of winter winds.

 The results of the new sensitivity experiment are presented and discussed in the sections 3.3 and 3.4 (page 14 to page 16) of the revised manuscript.

*C. Summary of the main results: I also believe that the authors should definitely include a schematic summarizing the main processes responsible for the long-term O2 evolution in the AS. This would really be helpful for the reader to grasp the salient results of the present study.*

We thank the reviewer for this suggestion. We have included a schematic summarizing the main findings in the revised version of the manuscript (please see Fig 11, page 19).

***Detailed comments:***

*Abstract:*
1) *P1 L2-3: Given the strong observational uncertainties, I would recommend the authors to lower down their claim about a decline in AS O2 over the past decades. Use for instance "suggest" instead of "show".*

Done. We have changed "show" to "suggest".

2) *More generally, I would recommend the authors to refer to the latest SROCC report on the ocean changes (Bindoff et al. 2019) to refine their statements on O2 trends in observations. This report indeed concludes that: ". . . the challenges of data sparsity, regional differences and the relatively large uncertainties on the oxygen changes across different studies, but also recognising that oxygen declines are significantly different to zero, leads to medium confidence in the observed oxygen decline. Âz In addition, this report ˙ does not mention the AS as one of the regions where the O2 decline is the strongest and best observed (they rather mention Southern Ocean, equatorial regions, North Pacific and South Atlantic).*

Done. We now cite the SROCC report (Bindoff et al., 2019) at multiple times in the revised manuscript to highlight the uncertainties about observed trends (please see page 2, lines 2, 5, 8, 15, page 11, lines 10-11 and page 22, lines 7-8).

3) *P1 L5: "while summer monsoon winds have intensified" : While this is true in the atmospheric reanalysis used in the present study, this intensification of summer monsoon winds may not be evident in all wind products. Indeed, several studies suggest that decadal wind variations are not well constrained in reanalyses, which may lead to strong uncertainties in long-term wind trends over the AS. In addition, most studies rather report a decrease of the Indian summer monsoon circulation over the historical period (e.g. Swapna et al. 2017). This uncertainty should be discussed in the manuscript.*

Done.

This uncertainty is now highlighted and better discussed in the revised manuscript. More specifically, in the introduction section we cite both works reporting an increase or poleward shift in summer winds (e.g., Sandeep and Ajayamohan 2016) as well as studies

reporting a weakening of summer monsoon circulation (e.g., Swapna et al. 2017). Please see page 3, lines 15-25.

Additionally, we have considered the trends in surface winds from different products following the reviewer suggestion (Fig S28, SI). As pointed out by the referee, there seem to be large discrepancies in terms of the magnitude and even the direction of the change in surface winds as these reanalyses are not very well constrained. Indeed, while the NCEP2 winds show a modest increase in upwelling-favorable winds in the western Arabian Sea, there seems to be no such an increase (and even a slight weakening) in the JRA55 winds.

Therefore, we discuss this uncertainty in detail in the revised version of the manuscript in a new paragraph under section 4.4 describing the caveats of the study (please see page 22, lines 14-27). While we acknowledge this as one of the caveats of the study, we also believe that this has relatively limited implications for the conclusions of the study as we demonstrate here that the dominant factor in deoxygenation in the northern Arabian Sea is surface warming, with the winds playing only a secondary role.

4) *P1 L5 : "reconstruct" I don't like the use of this term here, as their is not assimilation in the model. I would rather use "simulate".*

Done.

5) *P1 L7 : Replace Ân observation-based  ́z by  ̇n forced by an atmospheric reanalysis  ́z*

Done.

6) *P1 L12 : Â ̇n in ́ particular in the Gulf Âz : The gulf warming contributes to 1/3 of the O2 decline through ̇ stratification if I get it well from the bottom panels of Figure 11. I would rather use Ân ́ including in the Gulf Âz for not over-emphasizing the role of the Gulf warming. ̇*

Done.

*Introduction :*

7) *P2 L2-9: I would recommend the authors to also refer here to the latest SROCC report on the ocean changes (Bindoff et al. 2019) to refine their statements on the model projections and related uncertainties.*

Done. Please see our response to previous comment #2.

*8) P2 L13: "hypoxic events" Do you mean "anoxic events"?*

We refer to low O2 events in the hypoxic range (O2<60-80 mmol/m3) not necessarily anoxic events. These have been shown to stress sensitive organisms and cause loss of marine biodiversity and shifts in the food web structure (Rabalais et al., 2002; Vaquer-Sunyer and Duarte, 2008; Laffoley and Baxter, 2019).

*9) P2 L16-35: Please also refer to SROCC 2019 report about the confidence we have these reported changes.*

Done. Please see our response to previous comment #2.

*10) P3 L8: Regarding the enhanced warming in the AS, you should refer to Gopika et al. (2020) that investigate this in details.*

We thank the reviewer for his/her suggestion. Done. We refer to this study in the revised manuscript (please see page 3, line 3).

*11) P3 L12-14: Results from Roxy et al. (2016) which report a Chl decline over the western AS appears to be inconsistent with results from the present study. This inconsistency and its implication on the robustness of the authors conclusions should be discussed in Section 4.2.*

There are two differences between our study and Roxy et al (2016) that may explain this inconsistency. First, in their analysis Roxy and colleagues have considered surface chlorophyll while we are analyzing trends in vertically integrated biological productivity. While these two quantities are usually strongly correlated, they are not always identical, especially in tropical systems where deep chlorophyll maxima are a common feature. Indeed, the trends in surface chlorophyll in our simulation are quite different and are much weaker in the open ocean in comparison to trends in NPP (Fig S27, SI). The second main difference with Roxy et al. (2016) is the study period. Indeed, the satellite chlorophyll data presented in Roxy et al (2016) is based on a different and shorter period (1998-2013) than in the present study (1982-2010). Interestingly, our model also simulates a decline of surface chlorophyll in the western Arabian Sea when the analysis is restricted to the period between 1998 and 2010 (Fig S27, SI). We believe the high sensitivity of the trends to the study period is a consequence of the strong decadal variability in the region. This can also be seen in chlorophyll trends based on CMIP models presented in Roxy et al (2016). These time series that cover a longer period (1950-2005) reveal strong decadal variability with a decline in surface chlorophyll from the 1950s to the late 1970s but an increase from then on. We believe this can also contribute to this apparent inconsistency.

We discuss this point thoroughly in section 4.2 (comparison with previous works) of the revised manuscript as suggested by the reviewer (see page 20 to page 21, lines 1-13).

*12) P3 L17-18: I would recommend the authors to remove the reference to Goes et al. (2005) as the period covered by this study is very short (and strongly aliased by the 1997 El Niño) and their analysis very local. Most studies rather report a decrease of the Indian summer monsoon circulation over the historical period (e.g. Swapna et al. 2017) as well as in the future (e.g. Sooraj et al. 2015). Our current knowledge on the summer monsoon long-term trends should be better summarized here.*

This reference has been removed.

*13) P3 L23: Remove "largely"*

Done.

*14) P3 L29: Change ", in particular in the Gulf" into ", with a significant contribution from the Gulf warming"*

Done.

*Methods:*

*15) P4 L17: I am wondering why the authors did not extend their simulation until present days as ERA-I forcing is available until 2019. This would have allowed the authors to extend their validation over a period with more observations and monitor natural and anthropogenic decadal variations in more details. I would like the authors to clearly state why they did restrict their simulation to 2010 only.*

When this research effort was initiated back in 2017, the SODA reanalysis (version 2.2.4) used for boundary conditions was not available past the year 2010. Therefore we decided to run the simulation until 2010 (despite the fact that ERA forcing was available for more recent years). In 2018, a more recent version of SODA (version 3) was made available with coverage extending to recent years. Unfortunately, this version (and the more recent ones) has among other problems a major issue in the Arabian/Persion Gulf and the northern Arabian Sea (our focus area) with surface salinities unrealistically low (in the 33 psu) nearly 8 psu below the actual observations (Fig 2). Because we restore our surface salinity to SODA, and as the formation of dense water in the Arabian Gulf is extremely important for the ventilation of the Arabian Sea (Lachkar et al., 2019), we decided against extending the simulations using the most recent SODA version (3.x). We plan in the future to switch to an alternative reanalysis (e.g., ORAS5) to force our future simulations.

[Figure]

Figure 2: Sea surface salinity (in psu) from SODA version 2.2.4 (left) and SODA version 3.1 (right) in January 1984.

16) *P6 L7-P8 L5: Given the restoring to observed SST and SSS, the good agreement between observations and model is far from surprizing here. A dedicated Figure evaluating the surface circulation is not mandatory either given the scope of the paper. I would move these Figures in the SI to allow more space to validate the long-term evolution of key variables in the model.*

Done. The evaluation of the seasonal variability has been moved to the SI and a detailed evaluation of long-term changes has been included in the revised manuscript (please see our response to previous comment A).

17) *P8 L14-15: I would add two meridional sections to Figure 4 showing the ability of the model to simulate the oxygen vertical structure in the AS.*

Done. Following the reviewer suggestion, we have added an evaluation of the vertical structure of the simulated OMZ by showing O2 across one north-south (65E) and one east-west (18N) sections (Fig 1 in the revised manuscript).

18) *P9 L5-12: While this paper discusses the mechanisms responsible for the long-term oxygen changes in the AS, this small paragraph is the only place where the authors evaluate the ability of their model to simulate the AS long-term changes. I would recommend the authors to reduce their seasonal validation and expand the validation of the simulated long-term evolution in the AS. First, I find the authors methodology to derive long-term trends very rough (difference between first and last 5 years of the period considered). Why not trying to evaluate a proper linear trend with some significance level? In addition, why not trying to build an average evolution in observations in wide boxes (northern vs/southern AS) rather than a point-wise comparison?*

Done. Following the reviewer suggestion, we have substantially expanded the evaluation of long-term changes in the model. Please see our detailed response to previous general comment A. We have removed Fig S7 and Fig S8 and replaced them by Figs S11, S12, S13, S14, S15 and S16 in the revised SI.

19) The authors also argue that *"Contrasting the long-term temperature changes in the top 200 m in the model and in the observations reveals an overall good agreement in terms of the magnitude and the patterns of upper ocean warming in both summer and winter seasons (Fig S7 and Fig S8, SI). Âz I am ˙ not sure to agree with this statement as Fig. S7 and S8 do not show any colorbars and are very patchy. I believe the authors can considerably improve this paragraph and strengthen the reliability of their simulation by extending the validation of the model long-term evolution to other parameters, for which observations or reconstructed products are available. Maps of trends and yearly time-series of key parameters could be compared for model and observations for (1) SST, (2) upper ocean stratification, (3) winds (for several products as these products are not well contrained) and (4) maybe Chl at least over the common period where model and observational data from OC-CCI are available (to be able to compare results with Roxy et al. 2016).*

We have removed these figures (Fig S7 and S8) and replaced them with figures FigS11 to Fig S16 in the revised SI. Please see our detailed response to General Comment A.

20) *It would also be worth trying to compare the oxygen trends over key regions (northern/southern AS) in model and observational products such as those used in Ito et al. and Schmidtko et al. (2017). Indeed, while I find their analyses on the mechanisms driving the oxygen decline in the model very convincing, these results are not supported by observational evidences. This should be done whenever possible.*

Unfortunately, O2 observations are extremely limited in the area, especially during the 1982-2010 study period (most available observations were collected in 1994/1995 as a part of the JGOFS Arabian Sea Process Study) and hence cannot be used to estimate trends. Please see Fig S14 in the SI and our response to General Comment A.

*Results:*

21) *P9 L26-27 L30-31: How these numbers do compare with observed estimates globally and regionally?*

Done. We now contrast these numbers with published estimates of recent changes in global volume of OMZs as well as the volume of suboxic and anoxic waters based on Bindoff et al.

(2019), Schmidtko et al. (2017) and Deutsch et al. (2011). More concretely, we have included the following statement (lines 9-15, p11):

"To put these numbers in a broader context, observations suggest that the volume of the world ocean OMZs has expanded in a range of 3-8% between 1970 and 2010 (Bindoff et al., 2019). The stronger trends simulated for the suboxic volume relative to the hypoxic volume is also consistent with previous studies. For instance, it has been suggested that the volume of anoxic waters in the global ocean has quadrupled since 1960 (Schmidtko et al., 2017). Moreover, using numerical simulations, Deutsch et al. (2011) have shown
that the amplitude of variations in the volume of low oxygen conditions in the global ocean increases between 1959 and 2005 from 10% for strong hypoxia (O2 < 40 mmolm-3 ) to nearly 100% for suboxia."

22) *Figure 8bc: Plot a frame on Figure 6 to indicate the averaging region. It would also be interesting to add a time series on these panels showing the upper ocean stratification changes (for instance the static stability index). If ventilation dominates, we should expect this physical index to mirror the oxygen changes at interannual and longer timescales. This model stratification index could be further compared with observational estimates.*

Done. The averaging region in the northern AS is shown in Fig 4a (as well as in Figs 8a, 9a, 10a, 12a).

Following the reviewer suggestion, we have also superimposed upper ocean stratification changes (at 100m) on top of O2 changes in the 100-1000m layer in Fig 6 of the revised manuscript. As can be seen in the figure, the vertical stratification mirrors relatively well the O2 changes at interannual and longer timescales (correlation coefficient R= -0.65), confirming the dominance of ventilation control over O2 changes.

23) *Figure 9: Replace x-time axis by years (as in Figure 8) instead of months as currently done. Indicate over which region the analysis is performed and refer to Figure 6 to show this region.*

Done. In the revised manuscript the x-axis shows years (instead of months) and the region where the analysis is performed (the same northern AS box) is shown in Fig 4a and indicated in the caption.

24) *P14 L10-13 and Fig S9: Why does vertical and lateral components evolution mirror each other? Total advection term is actually a small residual between these two large terms. Can you explain?*

This is due to the fact that the net divergence of the transport term is much smaller than the vertical and lateral contributions taken separately (that partially compensate each other).

In the revised manuscript, we have split the total ventilation into total advection and vertical mixing components (see Fig 7 of the revised manuscript). This reveals that most of the decline in ventilation seen in the upper ocean is caused by a reduction in the contribution by vertical mixing. This is consistent with the enhanced vertical stratification suppressing vertical mixing in the upper ocean.

25) *P14 L15-17: This SST trend in the model could be compared with observations (even if we expect a good agreement because of the relaxation term used)*

Done. The agreement as expected is good (see Fig 2 and Figs S11 and S12 in the SI).

26) *P14 L17-18: The authors argue here that wind changes are particularly prominent during the summer monsoon. However, Figure S10 suggests that winter wind changes also strongly contribute to the annual wind changes shown on Figure 10b. First, I don't understand why the authors decided from that point to focus on the impact of summer wind changes and completely disregard winter wind changes. As previously noted, circulation weakening during the winter monsoon should lead to reduce convective mixing and contribute to reduce the ventilation in the northern AS. I recommend the authors to investigate this potential mechanism futher by performing an additional sensitivity experiment.*

Done. Please see our response to previous General Comment B.

27) *Second, because of the scarcity of atmospheric observations in the Indian Ocean, decadal to multi-decadal wind signals in this region are not well constrained in atmospheric reanalyses. I would suggest the authors to compare summer and winter wind trends shown on Figure S10 to other reanalyses products (JRA55 or NCEP2 for instance) to ascertain if the reported wind changes are a robust feature in all products.*

Done. Please see our response to previous comment #3.

28) *Figure 11: As they are shown at different places in the manuscript, it is not easy to compare results from the sensitivity experiments (shown in Figure 11) with those of the control simulation (which are shown earlier in the manuscript, i.e. Figure 6). I would recommend the authors to either re-add Figure 6a to Figure 11 to ease comparison or to directly show the difference between the sensitivity experiments and the control simulation. This would ease the interpretation of the results derived fro mthe sensitivity experiments.*

We now show the differences between the sensitivity experiments and the control (Fig 8) and have moved panels in previous Fig 11 to the SI.

29) *P16 L5-8 and Figure 12: In addition to Figure 12a, I would here show yearly time series of the stratification index against O2 evolution in the control simulation. This would allow to*

give some hints on whether the ventilation mechanism operating at long timescales also operate at interannual and decadal timescales, i.e. a strong anti-correlation between between these two time series would further strengthen the case for a strong control of ventilation on oxygen concentration at different timescales.

Done. Please see also our response to previous comment #22.

30) *P19 L17-23: The upward trend in surface Chl simulated in the model is opposite to what observed trends and model projections (see Roxy et al. 2016). This discrepancy and its implication should be addressed in the discussion section.*

Done. Please see our response to comment #11.

*Discussion:*

31) *P21 L5-7: Check consistency of wind trends in other atmospheric reanalyses (see above). To me, summer monsoonal winds rather weakened over the past decades in observations. Also discuss also here apparent inconsistency with Roxy et al. (2016) study (decline in PP over recent decades).*

Please see our response to comments #3 and #11.

32) *P21 L18-23: This is true but also mention that these models project a weakening of the summer monsoon circulation (e.g. Sooraj et al. 2015), which is not the case in the present study.*

We agree that there is a lot of uncertainty around the future changes in summer monsoon winds. Some models suggest a strengthening while others project a weakening.
We discuss this in the revised manuscript (Introduction, page 3, lines 15-25, and section 4.4, page 22 lines 14-27). Please also see our response to previous comment #3.

*References:*
*Bindoff, N.L., W.W.L. Cheung, J.G. Kairo, J. Arístegui, V.A. Guinder, R. Hallberg, N. Hilmi, N. Jiao, M.S. Karim, L. Levin, S. O'Donoghue, S.R. Purca Cuicapusa, B. Rinkevich, T. Suga, A. Tagliabue, and P. Williamson, 2019: Changing Ocean, Marine Ecosystems, and Dependent Communities. In: IPCC Special Report on the Ocean and Cryosphere in a Changing Climate [H.-O. Pörtner, D.C. Roberts, V. MassonDelmotte, P. Zhai, M. Tignor, E. Poloczanska, K. Mintenbeck, A. Alegría, M. Nicolai, A. Okem, J. Petzold, B. Rama, N.M. Weyer (eds.)]. 2019. Gopika, S., Izumo, T., Vialard, J., Lengaigne, M., Suresh, I., & Kumar, M. R. (2020). Aliasing of the Indian Ocean externally-forced warming spatial pattern by internal climate variability. Climate Dynamics, 54(1-2), 1093-1111. Sooraj, K. P., P. Terray, and M. Mujumdar (2015), Global*

*warming and the weakening of the Asian summer Monsoon circulation: assessments from the CMIP5 models, Clim. Dyn. 45, 1–20.âAˇˊl Swapna, P., Jyoti, J., Krishnan, R., Sandeep, N., & Griffies, S. M. (2017). Multidecadal weakening of Indian summer monsoon circulation induces an increasing northern Indian Ocean sea level. Geophysical Research Letters, 44(20), 10-560.*

References:

Lachkar, Z., Lévy, M., and Smith, S.: Intensification and deepening of the Arabian Sea oxygen minimum zone in response to increase in Indian monsoon wind intensity, Biogeosciences, 15, 159–186, https://doi.org/10.5194/bg-15-159-2018, 2018.

Lachkar, Z., Lévy, M., 5 and Smith, K.: Strong intensification of the Arabian Sea oxygen minimum zone in response to Arabian Gulf warming, Geophysical Research Letters, 46, 5420–5429, 2019.

Laffoley, D. D. and Baxter, J.: Ocean Deoxygenation: Everyone's Problem-Causes, Impacts, Consequences and Solutions, IUCN, 2019.

Prasad, T. G. (2004), A comparison of mixed‐layer dynamics between the Arabian Sea and Bay of Bengal: One‐dimensional model results, *J. Geophys. Res.*, 109, C03035, doi:10.1029/2003JC002000.

Rabalais, N. N., Turner, R. E., and Wiseman Jr, W. J.: Gulf of Mexico hypoxia, aka "The dead zone", Annual Review of ecology and Systematics, 33, 235–263, 2002.

Roxy, M. K., Modi, A., Murtugudde, R., Valsala, V., Panickal, S., Prasanna Kumar, S., Ravichandran, M., Vichi, M., and Lévy, M.: A reduction in marine primary productivity driven by rapid warming over the tropical Indian Ocean, Geophysical Research Letters, 43, 826–833, 2016.

Vaquer-Sunyer, R. and Duarte, C. M.: Thresholds of hypoxia for marine biodiversity, Proceedings of the National Academy of Sciences, 105, 15 452–15 457, 2008.

Weller, R. A., A. S. Fischer, D. L. Rudnick, C. C. Eriksen, T. D. Dickey, J. Marra, C. Fox, and R. Leben (2002), Moored observations of upper‐ocean response to the monsoons in the Arabian Sea during 1994–1995, *Deep Sea Res., Part II*, **49**, 2195– 2230.

We would like to thank the second reviewer for the comments that have significantly improved our manuscript. We highlight our responses, point by point, to the reviewer's general and specific comments and indicate the revisions we have made to the paper accordingly. The reviewer's comments are shown below in italics writing while our response is marked in red.

**Anonymous Referee #2**

*General comments The Arabian Sea (AS) Oxygen Minimum Zone (OMZ) has profound consequences on the ecosystem and climate, making it important to understand the evolution of oxygen in the AS. The present study offers to investigate the mechanisms driving the oxygen evolution in the AS using a set of sensitivity experiments performed with an eddy resolving model. They conclude that the deoxygenation in the northern AS is primarily caused by the reduced ventilation associated with the recent fast warming, particularly in the Gulf, while the summer monsoon winds intensification caused oxygenation in the rest of the AS.*

*While agreeing that the topic fits well within the scope of the journal, there are several lacunae both in terms of analyses and presentation that do not allow me to recommend the manuscript for acceptance in its present form. Below are my detailed review comments on the manuscript.*

*Specific comments*

*1. The title is somewhat overstated/misleading: "fast" warming is still debated in the observations (see for e.g. Gopika et al., 2020); the "deoxygenation" occurs mainly in the northern AS (as their results suggest), unlike what is stated in the title.*

The focus of the Gopika et al 2020 paper is on the period (1871-2016), which is different from our study period (1982-2010). In particular most of the uncertainties in SST trends referred to in that paper have to do with the quality and the gaps in data prior to the 1960s. Moreover, the uncertainties debated in that paper relate to the large-scale east-west and north-south patterns of the warming Indian Ocean and less about the average Arabian Sea warming itself. Finally, the warming in the Arabian Sea over the study period has been faster than the global average (e.g., Beal et al., 2020).
However, we do agree that most of the deoxygenation we report (and analyze) in this study concerns the northern Arabian Sea. Therefore, we have changed the title of the paper to: "Fast local warming is the main driver of recent deoxygenation in the northern Arabian Sea".

*2. What is the focus region – AS/northern AS/AS OMZ? The entire paper, including the abstract, switches its discussion between those regions, making it difficult for the reader to comprehend.*

As deoxygenation (and its implications for the OMZ) is largest in the northern part of the AS, we devote particular attention to the northern AS, which is our focus region.
Please note that we have changed the title to "Fast local warming is the main driver of recent deoxygenation in the northern Arabian Sea" to make this clearer. Please also note that in the revised manuscript we have also made this clearer by systematically referring to deoxygenation in the northern AS in the abstract, the introduction and in the discussion of our results and figures (Figs 5 to 12).

*3. The importance of Gulf warming for the AS OMZ has already been addressed by the authors in Lachkar et al., GRL, 2019, which is partly the focus of the current manuscript. The authors need to clarify or discuss this in detail.*

It is true that the importance of the Gulf warming for the AS OMZ has been explored in Lachkar et al. 2019 using a set of idealized future warming scenarios of the Gulf. In the present study, however, we explore the drivers of recent changes in O2 and demonstrate that the recent warming in the Gulf contributes to those trends, but is far from entirely explaining them. Indeed, significant deoxygenation is simulated when the AS warms up, even when the Gulf temperature does not change (see Fig 8 and FigS20 in the Supp Info). Therefore, the present study is consistent with the findings of Lachkar et al (2019) but goes beyond the previous work in terms of the processes taken into consideration (warming and wind changes over the entire Arabian Sea) and the nature of the perturbation (real observed changes vs. idealistic future scenarios).

*4. Schmidtko et al. (2017) did not specifically discuss a decline in oxygen in AS nor on the west coast of India. However, this paper is explicitly referenced in the introduction for claiming deoxygenation in the AS. Similarly, the study by do Rosorio Gomes et al. (2014) ´ is largely debated. On the other hand, many relevant references, for e.g. Sandeep and Ajayamohan, 2015, are not cited.*

Although the focus of the Schmidtko et al. (2017) is global, local O2 trends are shown regionally including in the Indian Ocean and the Arabian Sea. Indeed, in their extended Data Figure 1, panel a, O2 trends (in units of percentage of local dissolved oxygen) are shown in the AS. This figure indicates a drop of O2 in the western and northern AS as well as along the west coast of India between 1960 and 2010 that we refer to in our introduction.

This study is consistent with many other observational studies that have reported or suggested deoxygenation in the Arabian Sea and that we cite. Namely:
Ito et al. (2017),
Laffoley and Baxter, 2019,
Piontkovski and Al-Oufi (2015)

Banse et al. (2014)

Queste et al. (2018)

Al-Ansari et al., 2015; Al-Yamani and Naqvi, 2019 with a focus on the Arabian/Persian Gulf.

We also cite do Rosário Gomes et al. (2014) because it points to an important potential link between recent oxygen trends in the region and observed changes in the plankton community composition.

As for the study by Sandeep and Ajayamohan (2015) referred to by the reviewer, it is about potential changes in Indian monsoon atmospheric circulation and is not directly linked to O2 or deoxygenation in the Arabian Sea. However, we now cite this paper in the revised manuscript both in the Introduction and the Discussion sections when we discuss recent changes in regional monsoon winds (please see page 3, lines 15-16 and page 22, lines 16-17).

*5. The manuscript does not provide important details. For e.g., how the trends are computed? What is O2sat./AOU and how are they estimated? No details provided on O2 budget, ventilation and biological consumption terms, in the methods section.*

We have added a new section 2.4 in the revised manuscript detailing the calculation of trends and other oxygen diagnostics. In particular, we added the following statement (page 8, lines 27-31): " To investigate long-term changes in O2  time series and other environmental properties (e.g., SST, winds, stratification, etc), data was deseasonalized by removing monthly climatologies from the original time series. Furthermore, to identify significant trends in O2  and other environmental factors we used the non-parametric Mann-Kendall (MK) test (Mann, 1945; Kendall, 1948)  that does not assume normality of data distribution and hence is less sensitive to outliers and skewed distributions. Linear trends were computed from the slope of the least squares regression line."

Additionally, for more clarity we have added an explicit definition of AOU (Apparent Oxygen Utilization) and O2sat (oxygen saturation).

More concretely, we have added the following statement (page 8, lin 31 to page 9,  line 7): "To separate oxygen trends driven by changes in solubility (thermal effect) from those caused by changes in ocean ventilation and respiration of organic matter, we decompose the oxygen anomaly O2  as:

$$O2 =  O2sat  -  AOU$$

where O2sat  is the oxygen saturation concentration (in mmol m□3 ) computed at 1 atm pressure from temperature and salinity following Garcia and Gordon (1992). It corresponds to

the maximum O2  concentration in seawater at equilibrium, for a given temperature and salinity; AOU (apparent oxygen utilization) is the difference between oxygen saturation O2sat  and the actual O2  concentration (AOU=O2sat -O2). It is a measure of O2  utilization through biological activity since the water parcel was last at equilibrium in contact with the atmosphere. Therefore it is sensitive to both biological productivity and circulation (ventilation)."

As for the O2 budget, we have detailed the mass balance equation and described the mathematical representation of the O2 sources and sinks in the model in the revised version of the Supp Info.

*6. The manuscript is badly written, with careless handling in many places. Almost every paragraph contains sentences which are not easy to follow. I will mention some of them below. There are many typos too. For e.g. E or W in longitudes (P4 L25 and other places). Figures in supplementary do not have longitude labels, figures with vectors have no reference scale vector, and captions do not provide complete details (i.e. the region for average, Fig 6: what is the reference for % computation, etc.). I strongly suggest the authors to improve the presentation, including the abstract.*

We thank the reviewer for pointing out these typos that we have now corrected.
P4 L25: typo was corrected.
Missing longitude labels in some supplementary figures have been added.
A reference scale vector has been added to Fig 2 in the revised manuscript and Figs S7 and S28 of the Supp Info.
The area in the northern Arabian Sea (north of 20N) that we average over is shown in Fig 4a (also in Figs 8, 9, 10 and 12).
The inventories which are vertically integrated O2 concentrations are shown at each grid point. As for change expressed in % units, it is the amount of change as percentage of local dissolved oxygen inventory as presented in previous deoxygenation studies (e.g., Schmidtko et al, 2017). We have made this clearer by stating in the caption of Fig 4 that "relative trends (in % per decade) were obtained by dividing the absolute trends by the local mean O2 inventory".

*7. Avoid methodological details in the results. For e.g. P9 L20: the first two sentences of section 3.1 belongs to methods section. P14 L20: details of experiments should be moved to methods.*

Done. We have moved these two statements to the methods section (see new section 2.4 in page 8).

*8. P4 L22: "major rivers in the northern Indian Ocean". Provide details as to which rivers are considered, particularly in the Arabian Sea.*

Done. We now explicitly list the Indus and Narmada rivers flowing into the Arabian Sea (see page 4, lines 24-26).

*9. The authors provide validation of their model at the surface level and at the seasonal scale. The model shows a good performance as the fields are restored at the surface. However, as the paper focusses on the subsurface level, it is necessary to validate the model at the subsurface, for e.g. with Argo observations, both in terms of vertical profiles/sections. It is also important to demonstrate that there is no drift in the model at the deeper levels (e.g. ventilation).*

**Validation of the model at the subsurface:**
Beyond the validation of surface properties (including variables that are not restored such as currents in Fig S7, and NO3 and Chla in Fig S8, Supp Info) we have validated the model at the subsurface:

1) O2 and NO3 distributions between 250 and 700m from the model are compared to observations (Fig 1 of the revised manuscript and Fig S9, Supp Info).
2) Additionally, we have shown a quantitative comparison of O2 and NO3 profiles in the top 400m between the model and the observations as a part of the Taylor diagrams presented in Fig S10, Supp Info.

Additionally, in the revised manuscript we have added a thorough validation of long-term changes, including changes in temperature and salinity in the subsurface as well as stratification in the upper ocean. Please see our detailed response to General Comment A by referee #1.

**Model drift:**
This has already been addressed in the original manuscript. Please see the analysis of model drift in pages 3-4 of the Supp Info. Please also see salinity and oxygen long-term trends shown in Figs S2, S3, S4 and S5. Please note that the model drift is analyzed in two vertical layers: 0-200m and the 200-1000m as clearly stated in these figures captions.

*10. P9 L28: How do you ascertain that 14% increase in denitrification is due to 10% decline in oxygen? Are these anomalies correlated?*

We would like to stress that we are not referring to "10% decline in O2 " in this statement as stated by the reviewer but instead a 10 % increase in the volume of suboxia (O2 < 4mmol m−3 ) per decade. Denitrification in the model occurs only if O2 drops below this threshold. Therefore, an increase in the suboxic volume is naturally expected to lead to an increase in denitrification fluxes. Therefore these anomalies are strongly correlated, but are not necessarily identical in relative terms. This is because denitrification depends also on the amplitude of the organic matter fluxes that enter the suboxic zone, which have not been constant over time as can be seen in NPP trends (Fig 12a). However, in the northern AS where NPP trends tend to be weak,

increases in suboxic volume (+14% per dec) and denitrification (+15% per dec) are very similar (see Fig 5 in the revised manuscript).

*11. The manuscript discusses many processes and the link/flow is somehow missing. I suggest to first provide a discussion of possible causes of oxygen variations and then address each of them. Section 3.2 P10. It may not be clear to the reader the link between AOU and ventilation. The authors should provide substantial details of these parameters in the methods.*

Done. We have added a short paragraph summarizing what can cause O2 to change (solubility vs. AOU) and that AOU is controlled by ventilation and biology. More specifically, we have added the following statement at the beginning of section 3.2 (page 12, lines 1-7): "As oxygen equilibrates relatively rapidly at the air-sea interface relative to ocean circulation timescales, observed surface oxygen concentrations are generally close to their saturation levels (O2sat ). In the ocean interior, oxygen is depleted because of biological respiration. Therefore, changes in dissolved oxygen concentrations in the ocean interior translate either changes in oxygen saturation levels (solubility effect) or changes in the accumulated oxygen deficits associated with respiration. The latter term, measured by AOU (AOU=O2sat -O2 ), depends on both biological activity and ventilation. Here, we explore how these different components contribute to the simulated oxygen changes."

Additionally. We have added a definition of AOU and its link to biology/ventilation in the method section 2.4. (please also see our response to previous comment #5).

*12. P10 L5: O2 decline vs O2 saturation. How do you ascertain their link? What about AOU at this depth?*

O2 declined in the 0-30m layer at a rate of 0.35% per decade. The decline in O2 saturation in the same layer (as can be seen in Fig S19, Supp. Info.) amounts to 0.25% per decade. Therefore the change in O2 saturation explains around 70% of the total trend in O2 (the remainder is driven by AOU). Please note that O2=O2sat - AOU. Please also see our response to previous comment #5.

*13. P11 Fig. 6: Panel (c) shows O2 section at 65E. Different color scales make it difficult to compare. There seems to be some inconsistency in the deeper levels (below 200 m) when compared to panel (b), especially between 18 and 20N. Moreover, there appears to some discontinuity in the vertical layers (jumps in color shades) in panel c. Does it arise from some mistake from sigma to depth level conversion? Lastly, provide the details of how the % change is calculated in the caption.*

We replaced this with the new Fig 4 where we show (a) the trends in O2 inventory in % per decade in the the 100-1000 m layer, and (b) the trends in O2 concentration (not O2 inventory) in the upper 1000 m across 65°E (also in % per decade). As for the discontinuity in colors in (b), it stems from the fact that we show trends computed at each native sigma layer with no vertical interpolation or smoothing.

Finally for more clarity, we define the trends in relative terms (%) in the caption of the figure following the reviewer suggestion. More concretely, we have added the following statement in the figure caption: "Relative trends (in % per decade) were obtained by dividing the absolute trends by the local mean O2 inventory (a) or concentration (b)".

*14. P12 Fig. 7: What is the region (lat, lon, depth) over which the volume calculation is performed? The O2 trends are different in northern and southern AS, if you considered the entire AS. This figure suggests decadal signals that may alias the linear trend computation. Are these trend estimates hold good even after removing decadal signals?*

The volume of hypoxic and suboxic waters is evaluated in the upper 1000m. This is now stated explicitly in the figure caption. In the original manuscript, we integrated the volume horizontally over the entire Arabian Sea. In the revised manuscript, we show these volumes and denitrification rates integrated over the northern Arabian Sea box for consistency with the focus on the northern AS region (Fig 5 in the revised manuscript). However, the evolution of the suboxic volume and denitrification is quite similar to when the integration is done for the entire Arabian Sea (Fig S18, Supp, Info.) because suboxia tends to be located mostly in the northern AS (cyan line in Fig 4). However, the positive trend in the hypoxic volume gets stronger and statistically significant (p<0.05) when we restrict the analysis to the northern AS only. We conclude that in the northern AS, the OMZ not only gets more intense but also expands. This is discussed in the revised manuscript (please see page 11, lines 1-9).

Finally, we do acknowledge that decadal variability is potentially playing an important role in the identified linear trends. However, due to the relatively short simulation period (29 years), it is unfortunately not possible to rigorously filter out that (decadal variability) signal in a statistical sense. We acknowledge this in the list of the study caveats (section 4.4, page 22, lines 1-12). We also acknowledge the importance of interannual and decadal variability in the abstract (lines 11-12) and in section 3.2 (page 13, lines 1-4).

*15. P13 Fig. 8: How is O2sat computed? No mention of region in the caption. As you discuss AOU in the text, why is that AOU not shown on these panels.*

AOU is the difference between O2 and O2sat. We didn't include AOU in the plots because O2 and O2sat are already shown. Please see our responses to previous comments #5 and #12 for more details regarding the computation of O2sat and AOU.

*16. P14 L12. Inconsistency between figures 9 and S9. Fig S9 presents the vertical and horizontal components of ventilation term shown in blue in Fig 9. It is hard to see that they add up to the total ventilation, including their scales.*

The difference between Fig 9 and Fig S9 is that the ventilation in Fig S9 was not shown on the same scale (the changes in total ventilation are much smaller than the changes in the individual horizontal and vertical components taken separately).
In the revised manuscript we split total ventilation into advection and vertical mixing components. Please see our response to comment #24 by referee #1.

*17. P14 Section 3.3: The details of sensitivity experiments may be shifted to a new section under methods.*

Done. Please see new section 2.5 on page 9.

*18. How sensitive are the results to a different forcing set other than the ERA-Interim? How well the oxygen trends are constrained when using different forcing field? Are the decadal signals (say in winds) consistent among different observations?*

We discuss this in our response to Reviewer 1 comment #3.

*19. P14: The sensitivity experiments are conducted by repeating the cycles of 1986 to remove interannual variations and is considered to vary climatologically. I am wondering why 1986 repeated cycles are used for forcing or why this particular choice is made. A cleaner approach would be to compute the climatology (of heat fluxes in AS/Gulf or winds) and use it to force the model.*

The reason we use the repeated normal year approach instead of a monthly climatology (as done routinely to filter out the interannual variability while keeping high-frequency forcing, e.g., Large and Yeager, 2004, Stewart et al., 2020) is that mixing and Ekman circulation are highly sensitive to the high-frequency wind variability (associated with weather systems) as wind stress is estimated using bulk formulas based on wind speed data. As the wind stress is a quadratic function of wind speed, using smoothed wind speed data would result in artificially suppressed Ekman velocities. This approach is commonly used in models using bulk formula forcing. Finally, the choice of 1986 is justified by the fact that this year is neutral with respect to IOD and El Nino/La Nina oscillation. This is highlighted in the revised manuscript (page 9, lines 14-17).

*20. I have a major concern regarding the sensitivity experiments. First of all, the authors do not provide much details on the region over which the fields are allowed to vary climatologically (1986 repeated cycles). Moreover, there is an underlying assumption here. How do you ensure that warming results primarily from the heat fluxes. For e.g., by suppressing the heat flux variations in the Gulf, are you able to remove the Gulf warming? Did you check whether the AS heat fluxes affect the Gulf warming? Similarly, does the AS warming results mainly from the heat*

*fluxes over the AS? In the absence of a figure showing warming trends (somewhat like Figure 10, or time series averaged over Gulf/AS for sensitivity experiments), it is not justified to presume that the experiments indeed isolate the respective processes.*

We show SST trends in all sensitivity experiments below. This confirms that the warming signal is entirely driven by the interannual variability in the forcing.

[Figure]

Linear trends in SST (in C per decade) from the control simulation (top left), the no-summer-wind-changes simulation (top right), the no-warming simulation (bottom left), the no-warming-over-the-Gulf (bottom right).

*21. P15 Figure 9: There appears to be decadal signals, which may potentially alter the trend estimates. No mention of the region in the caption. Panel numbering has gone wrong. The terms have to be described in the methods.*

For the decadal variability see our response to previous comment #14.
The region over which the budget is estimated is north of 20N. This is indicated in the text and now also in the figure caption. Furthermore, we show the location of the averaging box in Fig4a of the revised manuscript.

Typos with the panel numbers are fixed.
The terms of the mass balance equation are defined in the Supp. Info (please also see our response to previous comment #5).

*22. P15 L6: "summer upwelling intensification . . ." How do you conclude that there is an upwelling intensification as there is no figure to support this claim? And how upwelling region contributes to the rest of the basin? These claims are mere speculations without any justification.*

Summer upwelling intensification can be seen in Fig2c that shows an intensification of upwelling favorable winds off the western AS. It can also be seen in the positive trends in Ekman suction velocity shown below.

[Figure]

Figure caption: Linear trends in Ekman suction velocity (upwelling) during summer monsoon season (in m/day per decade).

*23. Fig 11 vs S11. The authors show the maps from the sensitivity experiments, but have chosen to move the maps that demonstrate the effect of each process to supplementary figures. For instance, to know the effect of heat fluxes, one has to refer to the supplementary figure (say S11). I would choose the other way, as it is difficult to follow the discussion in the text. I guess, the text can be considerably improved if the authors stick to discuss only the effect of processes, instead of describing the results of sensitivity experiments.*

Done. We have followed the reviewer's suggestion and moved the figures showing the effect of each process to the main paper (Figs 8, 9, 10, 12), while the results of the sensitivity experiments were moved to the Supp. Info (Figs S20, S21, S24 and S25).

*24. P16 Fig. 10. There is no reference vector. I find some inconsistency between background color shade (speed) and the vectors. Negative shades on the west coast of India and the direction of overlaid winds do not agree. Further, the low magnitude vectors south of equator & to the east of 65E correspond to high wind speeds (red shade). I expect the trends in wind stresses and wind speeds are not entirely different.*

We have added a reference vector for wind stress trends.
Finally, the apparent inconsistency between the trends in the wind stress vector and the trends in the wind speed is a consequence of the fact that wind stress and wind speed have a nonlinear relationship, and hence their linear trends are not necessarily proportional.

*25. P17 Figure 11: Panels g & h are not consistent with the rest of the panels, when visually averaged over the northern AS. As emphasized before, I would suggest to interchange Fig. 11 with S11 to ease the discussion in the text.*

We would like to note that the trends shown in g and h are trends in total inventory of O2 integrated over the northern AS. They are not computed from spatially averaging the local linear trends in the northern AS as the referee question may suggest. Therefore, trends shown in g and h cannot be directly inferred from visually averaging the local trends shown in other panels (which are expressed in % of local O2 inventory). Please also note that this figure was moved to the Supp. Info.

Interchanging Fig 11 with Fig S11: done.

*26. P19 L9-10: ". . . potentially impacting O2 supply to northern AS". How?*

This statement was removed in the revised version of the manuscript.

*27. P19 L15-16: "the shoaling of thermocline . . . top 200 m". Not really justified. How well thermocline anomalies are correlated with O2 anomalies?*

Previous works on Arabian Sea OMZ have shown a strong correlation between thermocline depth and oxycline depth on both seasonal and interannual timescales (e.g., Vallivattathillam et al., 2017). This has been clarified in the revised manuscript (please see page 15, lines 24-26 and page 17, lines 6-15).

*28. P19 L29-30: The authors point to an increased summer upwelling on the western AS. However, there is no clear evidence for its contribution to northern AS. On the other hand, the northern AS productivity is mainly due to convective mixing during winter. The influence of winter monsoon has not been discussed in the manuscript. The winter monsoonal winds directly contribute to the ventilation as well as through productivity changes to the oxygen in the northern AS.*

We have added a new sensitivity experiment to test for the impact of changes in winter monsoon winds and have included a discussion of the role of winter monsoon winds in the revised manuscript. Please see our response to General Comment B by referee #1.

*29. P19 L33-34: The negative feedback of denitrification is merely speculated and not really demonstrated. Thus, its effect is overstated.*

We do not agree with the referee's statement that the denitrification feedback is not demonstrated and just speculation. Fig S26 quantifies this feedback and shows that it is not only statistically significant but also has the same order of magnitude as the changes in biological sources minus sinks of O2 (Fig S25).

*30. Figure S10: It is not convincing that there are no strong SST trends in the upwelling region of the western AS during JJAS.*

The absence of strong warming (and even local cooling) in the western AS during JJAS is a consequence of the concomitant increase in upwelling favorable winds and upwelling intensification over the study period (Fig 2). Please also see trends in summer Ekman suction velocity (shown in response to comment #22 by the same reviewer) confirming these trends.

*31. Figure S11: The 4% increase marked in panel a is for entire AS, not for northern AS as mentioned in the text (Section 3.4; P16 L6). Stratification shows a clear decadal signal. Technical corrections (not all) 1. P19 L2-4: "This suggests . . . at play". Not obvious to the reader. The entire first paragraph in this page is not readable. 2. Many panels and many of the supplementary figures are not properly referenced in the manuscript. 3. Figures S7 & S8: Not easy for me to understand. No color bar. Repeating statements in caption.*

This was a typo in the caption of figure S11 as the stratification is shown for the northern AS as mentioned in the text. This has been corrected in the revised manuscript (please see Fig 6).

In response to the reviewer comment, we have rewritten the first paragraph of section 3.4 to enhance the clarity of the text. In particular, we now start this section with a listing of the potentially important processes that affect thermocline ventilation as also suggested by the referee (please see page 15, lines 22-26 of the revised manuscript).

Finally, Figs S7 and S8 were removed and replaced by new evaluation of long-term changes in temperature and salinity (please see our response to General Comment A by referee #1).

*References*
*Gopika, S., Izumo, T., Vialard, J. et al. Aliasing of the Indian Ocean externally-forced warming spatial pattern by internal climate variability. Clim Dyn 54, 1093–1111 (2020). https://doi.org/10.1007/s00382-019-05049-9 Sandeep, S., Ajayamohan, R.S. Poleward shift in Indian summer monsoon low level jetstream under global warming. Clim Dyn 45, 337–351 (2015).*

*https://doi.org/10.1007/s00382-014-2261-y Please also note the supplement to this comment:*
*https://bg.copernicus.org/preprints/bg-2020-325/bg-2020-325-RC2-supplement.pdf*

References:

Beal, L.M., J. Vialard, M.K. Roxy, J. Li, M. Andres, H. Annamalai, V. Parvathi, A roadmap to IndOOS-2: better observations of the rapidly-warming Indian Ocean, Bull. Am. Meteorol. Soc., 101 (2020).

Lachkar, Z., Smith, S., Lévy, M., and Pauluis, O.: Eddies reduce denitrification and compress habitats in the Arabian Sea, Geophysical Research Letters, 43, 9148–9156, 2016.

Lachkar, Z., Lévy, M., 5 and Smith, K.: Strong intensification of the Arabian Sea oxygen minimum zone in response to Arabian Gulf warming, Geophysical Research Letters, 46, 5420–5429, 2019.

Laffoley, D. D. and Baxter, J.: Ocean Deoxygenation: Everyone's Problem-Causes, Impacts, Consequences and Solutions, IUCN, 2019.

Large W.G., Yeager S.G., Diurnal to Decadal Global Forcing for Ocean and Sea-Ice Models: The Data Sets and Flux Climatologies: NCAR Technical Note NCAR/TN-460+STR (2004), 10.5065/D6KK98Q6

Garcia, H. E.; Gordon, L. I. Oxygen solubility in seawater: better fitting equations. *Limnol. Oceanogr.* 1992, *37*, 1307– 1312, DOI: 10.4319/lo.1992.37.6.1307.

Rixen, T., Cowie, G., Gaye, B., Goes, J., do Rosário Gomes, H., Hood, R. R., Lachkar, Z., Schmidt, H., Segschneider, J., and Singh, A.: Reviews and syntheses: Present, past, and future of the oxygen minimum zone in the northern Indian Ocean, Biogeosciences, 17, 6051–6080, https://doi.org/10.5194/bg-17-6051-2020, 2020.

Stewart, K. D., Kim, W.M. , S. Urakawa, A.Mc C. Hogg, S. Yeager, H. Tsujino, H. Nakano, A.E. Kiss, G. Danabasoglu, JRA55-do-based repeat year forcing datasets for driving ocean–sea-ice models, Ocean Modelling, Volume 147, 2020, 101557, https://doi.org/10.1016/j.ocemod.2019.101557.

Vallivattathillam, P., Iyyappan, S., Lengaigne, M., Ethé, C., Vialard, J., Levy, M., Suresh, N., Aumont, O., Resplandy, L., Naik, H., and Naqvi, W.: Positive Indian Ocean Dipole events prevent anoxia off the west coast of India, Biogeosciences, 14, 1541–1559, https://doi.org/10.5194/bg-14-1541-2017, 2017.

**Review#3**

We are grateful for reviewer #3 efforts in reviewing the manuscript and for constructive and valuable comments. We highlight our responses to the comments and explain the revisions we have made to the paper accordingly. The reviewer's comments are shown below in italics writing while our response is marked in red.

**Anonymous Referee #3**

*I found this paper very interesting and worthy of being published in BG because it describes with great clarity and yet simply how the OMZ of the Arabian Sea is sensitive to upper-ocean warming and monsoonal changes. It should be of great utility especially to non-modelers especially observational researchers who want to understand how their data falls into the big picture.*

We thank the reviewer for the positive comment and the encouraging assessment of the paper.

*ABSTRACT: Although I finally did understand this sentence "This is because surface warming enhances vertical stratification, thus limiting ventilation of the intermediate ocean, while summer monsoon wind intensification causes the thermocline depth to rise in the northern AS and deepen elsewhere, thus contributing to lowering O2 levels in the upper 200 m in the northern AS and increasing it in the rest of the AS' it should be improved as its part of the Abstract which is what most researchers will read.*

This statement has been rewritten for more clarity. Please see the new statement (page 1, lines 15-17):
"This is because, on the one hand, surface warming enhances vertical stratification and increases Gulf water buoyancy, thus inhibiting vertical mixing and ventilation of the thermocline. On the other hand, summer monsoon wind intensification causes a rise of the thermocline depth in the northern AS that lowers O2  levels in the upper ocean."

*INTRODUCTION:*
*Mechanisms for deoxygenation should be described better. The authors should clarify where oxygenation vs deoxygenation takes place. The biology of the Arabian Sea is vastly different in the north vs the south of the Arabian Sea. Also page 3 and the two paras comprising lines 23-30 could be better organized*

Please see lines 19-34, page 2 where deoxygenation in different parts of the Arabian Sea (north vs south vs west) is discussed. We also discuss the contrasts in potential future deoxygenation vs oxygenation in the northern vs. central and southern Arabian Sea in section 4.3 on page 21.

*RESULTS: The authors contend that biological productivity contributes only minimally to deoxygenation, because stratification decreases input of nutrients and so reduces primary productivity. Conversely, enhanced summer winds will increase biological productivity. However, Goes et al. (2020) has shown that in spite of warmer, winter monsoonal winds, decreased convective winds and consequent increased stratification, winter Chlorophyll a has seen an increase (Fig. 1c) because of the rise of Noctiluca scintillans blooms which have high biomass on account of the large populations of photosynthetic endosymbionts that they harbor and their tight nutrient recycling mechanisms that allow them to survive in spite of lowered nutrient concentrations. It would be interesting to see how this recent change in biodiversity of winter blooms plays into the role of deoxygenation.*

*Goes, J.I., Tian, H., Gomes, H.d.R., Anderson, O.R., Al-Hashmi, K., deRada, S., Luo, H., Al-Kharusi, L., AlAzri, A., and Martinson, D.G. (2020). Ecosystem state change in the Arabian Sea fuelled*

We thank the reviewer for bringing this important recent study to our attention also showing evidence of enhanced stratification with potential consequences for the Arabian Sea ecosystem. We now refer to this work in the revised manuscript (page 3, lines 9-12).

---

## Referee Report (RR1)

**Comments on Revised Manuscript titled "Fast local warming is the main driver of recent deoxygenation in the northern Arabian Sea" by Zouhair Lachkar et al.**

**General comments**

I find that the revised manuscript has undergone extensive changes and has significantly improved in its scientific content, with much cleaner analyses and illustrations compared to the original version. The authors have carefully addressed all my comments on the original version. While some improvements are still required in writing (presentation), I would recommend the manuscript be considered for publication after addressing some minor issues listed below.

**Minor comments**

1. P6 L1: It should be S3 instead of S2.
2. Fig S1: Change 76W to 76E.
3. P9 L13: change "three" to "four".
4. P9 L14-24: Mention or mark the region over which the heat fluxes are set to climatological in $S_{hclim}$ and $S_{hclim\_AG}$. I presume (as inferred from your answer to my earlier comment # 3) that Gulf warming is not included in $S_{hclim}$. Similarly, provide precisely the region over which the winds have been modified in $S_{wclim\_JJAS}$ and $S_{wclim\_DJFM}$.
5. What are the 0 contours in Figure 4b in most of the regions? The consistency between contours (last minus first five years) and color shading (trend) is unclear.
6. P15 L10-12: "This is as oxygen…" - Not clear
7. P17 L11-13: I find that summer wind intensification occurs mainly south of 20°N in the AS (Figures 2c, S17d). Then how does the wind intensification drive shoaling of thermocline depth in the northern AS? This needs clarification.
8. P17 L31-32:  Is it referring to Oschiles et al. (2019; Loss of fixed nitrogen causes net oxygen gain in a warmer future ocean)?  May cite the paper.
9. Fig S13: How are the trends computed with gaps in the data?
10. Fig S14: Which box is referred to in the caption?
11. Fig S16 vs S8: Why different (SeaWiFs/OC-CCI) products are considered, though both are available during the period considered?
12. Fig S18: Mention it for the AS.

---

## Author Response (AR2)

Review #2

We are grateful to Referee #2's for his/her substantial efforts in reviewing the revised manuscript and for the constructive and valuable comments. We believe our paper has improved as a result of his/her comments. We highlight our responses to the general and specific comments and explain the revisions we have made to the paper accordingly. The reviewer's comments are shown below in italics writing while our response is marked in red.

**Comments on Revised Manuscript titled "Fast local warming is the main driver of recent deoxygenation in the northern Arabian Sea" by Zouhair Lachkar et al.**

**General comments**

*I find that the revised manuscript has undergone extensive changes and has significantly improved in its scientific content, with much cleaner analyses and illustrations compared to the original version. The authors have carefully addressed all my comments on the original version. While some improvements are still required in writing (presentation), I would recommend the manuscript be considered for publication after addressing some minor issues listed below.*

We thank the reviewer for the encouraging comment and for the positive assessment of the revised paper.

**Minor comments**

*1. P6 L1: It should be S3 instead of S2.*

Corrected.

*2. Fig S1: Change 76W to 76E.*

Corrected.

*3. P9 L13: change "three" to "four".*

Corrected.

*4. P9 L14-24: Mention or mark the region over which the heat fluxes are set to climatological in $S_{hclim}$ and $S_{hclim\_AG}$. I presume (as inferred from your answer to my earlier comment # 3) that Gulf warming is not included in $S_{hclim}$. Similarly, provide precisely the region over which the winds have been modified in $S_{wclim\_JJAS}$ and $S_{wclim\_DJFM}$.*

Done. We now specifically mention that the climatological heat fluxes are prescribed across the entire domain for simulations $S_{hclim}$, $S_{wclim\_JJAS}$ and $S_{wclim\_DJFM}$, whereas for simulation $S_{hclim\_AG}$ (where Gulf warming is not included) the climatological heat fluxes are prescribed over the Arabian Gulf region only.
Please see lines 16-24, page P9, of the revised manuscript.

*5. What are the 0 contours in Figure 4b in most of the regions? The consistency between contours (last minus first five years) and color shading (trend) is unclear.*

The contour lines correspond to changes between the first five years [1982-1986] and the last five years [2006-2010] in O2 (in mmol/m3/decade). This is now explicitly stated in the revised caption of Fig4b. Although they are not exactly identical with the linear trends because of the interdecadal variability in the timeseries and the difference in the units, the general patterns of oxygenation/deoxygenation do correspond relatively well, especially in the northern Arabian Sea.

*6. P15 L10-12: "This is as oxygen…" - Not clear*

To clarify the statement, we changed it to: "Indeed, oxygen decreases in the northern AS by around 9% between 1982 and 2010 under climatological summer winds, a rate that is nearly 50% weaker than in the control run during the same period (Fig S20, SI)." (P15, line 10).

*7. P17 L11-13: I find that summer wind intensification occurs mainly south of 20₀N in the AS (Figures 2c, S17d). Then how does the wind intensification drive shoaling of thermocline depth in the northern AS? This needs clarification.*

Although the summer wind intensification is stronger south of 20N as correctly pointed out by the referee, a strong negative wind stress curl (causing downwelling and thermocline deepening) dominates in the open ocean south of 20N as can be seen in the Ekman velocity and its trends included in our response to previous reviewer's comment #22. In contrast, open ocean upwelling and upwelling trends dominate north of 20N.

For more clarity, we have included the figure showing the average (and linear trends) in the Ekman suction velocity during the summer monsoon season in the Supp Information (new Fig S24).
We have also added the following statement:
"This suggests that summer monsoon wind intensification causes the thermocline depth to rise in the northern AS and deepen elsewhere (Fig 10). This is likely due to enhanced open ocean upwelling (Ekman suction) in the north and downwelling (Ekman pumping) in the south (Fig S24)." (P17, lines-13-14).

*8. P17 L31-32: Is it referring to Oschiles et al. (2019; Loss of fixed nitrogen causes net oxygen gain in a warmer future ocean)? May cite the paper.*

This statement was in reference to Fig S26. We have added the following statement with a reference to Oschlies et al. (2019) in the revised manuscript:
"[..], a process previously shown to be important for the oxygen budget on long timescales (Oschlies et al., 2019)". (P18, line 1).

*9. Fig S13: How are the trends computed with gaps in the data?*

These trends have been shown and included in response to Referee #1 comment. We do acknowledge that the substantial gaps in salinity data precludes extracting statistically significant trends as can be read in the manuscript text (page 8, lines 9-11):
"Yet, the highly sparse observational coverage (most of the observations coming from the last decade of the simulation) precludes extracting meaningful trends from the data to validate the simulated salinity long-term changes (Fig S14, SI)."

*10. Fig S14: Which box is referred to in the caption?*

The box is shown in Fig 4a. The caption was corrected.

*11. Fig S16 vs S8: Why different (SeaWiFs/OC-CCI) products are considered, though*
*both are available during the period considered?*

We initially used the Ocean-Colour Climate Change Initiative (OC-CCI) product to evaluate the interannual variability in simulated chlorophyll in response to an explicit request made by Referee #1.
However, for more consistency with the seasonal chlorophyll variability analysis we have now replaced the OC-CCI dataset with the SeaWiFS product over the same period. This does not change the comparison dramatically and only slightly improves the correlation between simulated and observed interannual chlorophyll fluctuations (R=0.54 instead of R=0.48 with OC-CCI). Please see the new Fig S16.

*12. Fig S18: Mention it for the AS.*

Done.